# Understanding Guidance Scale in Diffusion Models from a Geometric Perspective

**Zhiyuan Zhan**  *zhan@ms.k.u-tokyo.ac.jp*
*The University of Tokyo, Japan*
*RIKEN Center for AIP, Japan*

**Liuzhuozheng Li**  *liuzhuozheng-li@outlook.com*
*The University of Tokyo, Japan*

**Masashi Sugiyama**  *sugi@k.u-tokyo.ac.jp*
*RIKEN Center for AIP, Japan*
*The University of Tokyo, Japan*

**Reviewed on OpenReview:** *https://openreview.net/forum?id=nfHimL6g8G*

## Abstract

Conditional diffusion models have become a leading approach for generating condition-consistent samples, such as class-specific images. In practice, the guidance scale is a key hyperparameter in conditional diffusion models, used to adjust the strength of the guidance term. While empirical studies have demonstrated that appropriately choosing the scale can significantly enhance generation quality, the theoretical understanding of its role remains limited. In this work, we analyze the probabilistic guidance term from a geometric view under the linear manifold assumption and, based on this analysis, construct a geometric guidance model that enables tractable theoretical study. To address regularity issues arising from multi-modal data, we introduce a mollification technique that ensures well-posed dynamics. Our theoretical results show that increasing the guidance scale improves alignment with the target data manifold, thereby enhancing generation performance. We further extend our framework to nonlinear manifolds, and empirical results on real-world datasets validate the effectiveness of the proposed model and are consistent with our theories.

## 1 Introduction

Diffusion models (Ho et al., 2020; Song et al., 2021a) have achieved state-of-the-art performance on generative tasks across various domains, including images (Dhariwal & Nichol, 2021; Rombach et al., 2022), text-to-image synthesis (Saharia et al., 2022), videos (Ho et al., 2022), and audio (Kong et al., 2021). As a result, their empirical success has led to increasing interest in understanding the theoretical foundations of diffusion models (De Bortoli et al., 2021; Lee et al., 2022; Chen et al., 2023c;a; Gao et al., 2025). In particular, under the manifold hypothesis (Bengio et al., 2013), the ability of diffusion models to output high-quality samples in high-dimensional spaces motivates researchers to investigate how these models can generate distributions supported on low-dimensional manifolds in high-dimensional ambient spaces (De Bortoli, 2022; Oko et al., 2023; Li & Yan, 2024; Wan et al., 2025).

Controlling diffusion models to generate conditional distributions is another active area of research. Based on the theoretical framework proposed by Song et al. (2021b), both classifier guidance and classifier-free guidance models (Dhariwal & Nichol, 2021; Ho & Salimans, 2022) apply a probabilistic guidance term—derived from Bayes' rule—to guide the sampling process toward the target conditional distribution. These methods also introduce a scale to adjust the strength of the guidance, and they showed that the performance depends strongly on the choice of the guidance scale and an appropriate value can significantly improve generation

quality. Recent empirical studies further demonstrated the importance of the guidance scale in conditional generation tasks (Dinh et al., 2023; Sadat et al., 2024; 2025). However, the theoretical understanding of how the guidance scale affects the generation remains limited (Chidambaram et al., 2024; Wu et al., 2024).

In this work, we propose a new geometric guidance model to enable the theoretical analysis of the role of the guidance scale in conditional generation. A key challenge in studying the guidance scale in classifier(-free) models is the analytical complexity of the probabilistic guidance term. To address this, we replace the probabilistic guidance with a new geometric guidance term. Specifically, under the linear manifold hypothesis (Chung et al., 2022), we study the geometric property of the original probabilistic guidance term, building on an idea introduced by Chen et al. (2023b), and construct a linear geometric guidance term that plays the same role but more tractable for theoretical analysis.

As a next step, the analysis of the geometric guidance model requires certain regularity conditions on the score function, such as the Lipschitz continuity. However, because of the multi-modality of data distributions, these conditions generally fail to hold (Lee et al., 2022; Gao et al., 2025). To overcome this issue, we introduce a mollification technique inspired by mollifiers in mathematical analysis (Evans, 2018) to construct a surrogate score function that satisfies the required properties for our analysis.

Building on this, we construct a well-posed geometric guidance model through which we address two questions: (i) whether the model can recover the target data manifold, and (ii) what is the upper bound on the distance between the generated distribution and the target conditional distribution. Our results reveal the effects of the guidance scale: increasing the scale encourages the generated data to lie closer to the target manifold, and large guidance scales do not significantly increase an upper bound on the generation error.

Finally, for the nonlinear case and real-world data distributions, we extend our framework by constructing a nonlinear geometric guidance model. This model builds on the same principles as the linear case, with the theoretical foundation obtained by extending the results of Chung et al. (2022) to nonlinear data manifolds. Experimentally, we evaluate the nonlinear geometric guidance model on CIFAR-10 (Krizhevsky, 2009) and demonstrate its effectiveness for conditional generation. We also report how performance varies with the guidance scale, providing empirical evidence consistent with the behavior suggested by our linear analysis.

In summary, our contributions are:

1. We construct a new linear geometric guidance term to replace the original probabilistic guidance term by studying its geometric property under the linear manifold hypothesis.

2. To ensure the regularity of the unconditional score function, we apply a mollification technique to construct a a surrogate score function, and build a well-posed geometric guidance model.

3. By analyzing the geometric guidance model, we uncover the role of the guidance scale: a large guidance scale encourages the generated data to lie closer to the target data manifold and does not significantly affect the upper bound of the generation error.

4. We propose a principled nonlinear geometric guidance model and evaluate it on CIFAR-10; the experiments demonstrate its effectiveness in conditional generation and illustrate guidance-scale effects beyond the linear setting.

The remainder of this paper is organized as follows. Section 2 reviews related work, and Section 3 summarizes the technical background on diffusion models. Section 4 introduces the construction of the geometric guidance term, and Section 5 presents the theoretical analysis of the geometric guidance model. Section 6 extends the model to nonlinear settings and reports experimental results. Section 7 concludes the paper and discusses limitations. Notation is summarized in Appendix A.

## 2 Related Works

**Convergence analysis:** A number of recent works have analyzed the convergence properties of diffusion models under various assumptions (De Bortoli et al., 2021; Lee et al., 2022; 2023; Chen et al., 2023c;a; Gao et al., 2025). De Bortoli et al. (2021) established total variation bounds under $C^3$-regularity assumptions

on the score for the target distribution. Chen et al. (2023c) relaxed this requirement to Lipschitz continuity of the score function but for each intermediate density, which was further weakened in Chen et al. (2023a) to the Lipschitz continuity of the score only for the target density. Using functional inequalities, Lee et al. (2022; 2023) and Gao et al. (2025) have derived convergence guarantees under the assumption that the target density function is log-concave, with results in both total variation and Wasserstein distances. In contrast, our setting involves multi-modal target distributions for which log-concavity and smoothness assumptions do not hold (Lee et al., 2022). To address this, we introduce a technique that constructs a surrogate distribution satisfying the required regularity properties while closely approximating the original target.

**Geometric structure:** For real-world datasets, it is widely believed that high-dimensional data lie on a low-dimensional submanifold of the ambient space, a perspective known as the manifold hypothesis (Bengio et al., 2013). When generating such data distributions, deep generative models often encounter challenges such as the curse of dimensionality (Bronstein et al., 2021) and manifold overfitting (Loaiza-Ganem et al., 2022). However, the strong empirical performance suggests that diffusion models can avoid these issues. As a result, understanding the theoretical behavior of diffusion models under the manifold hypothesis has attracted increasing attention. For example, De Bortoli (2022) established a Wasserstein convergence bound assuming that the target distribution is supported on a compact set. Under the additional assumption that the target data manifold is linear, Oko et al. (2023) showed that diffusion models can avoid the curse of dimensionality by providing a Wasserstein bound that depends only on the intrinsic dimension. Chen et al. (2023b) further derived a total variation bound in terms of the intrinsic dimension based on a decomposition of the score function under the linear manifold assumption. Following this line of work, we further investigate the geometric structure of this decomposition to clarify the role of the score function in recovering the target data manifold, which in turn helps us construct the geometric guidance model.

**Conditional generation:** To control the generation (Song et al., 2021b), Dhariwal & Nichol (2021) and Ho & Salimans (2022) applied the probabilistic guidance term to generate conditional distributions. Following their works and based on the geometric structure of noisy data manifolds under the linear assumption of the target data manifold (Chung et al., 2022), Chung et al. (2022; 2024) and He et al. (2024) proposed using a new time-dependent guidance in conditional generation to constrain geometric structure of the generation process. From a different perspective, Song et al. (2023) and Bansal et al. (2023) constructed a time-independent guidance constructed by a loss function that is designed to enforce desired constraints on the generated data. Instead, our geometric guidance is constructed by studying the geometric property of the probabilistic guidance, with the goal of replacing its role in conditional generation.

To adjust the strength of guidance, Dhariwal & Nichol (2021) and Ho & Salimans (2022) also introduced a guidance scale, and their experiments showed that selecting an appropriate scale can significantly improve performance. However, there are limited works on theoretically analyzing the effects of the guidance scale in conditional generation. Chidambaram et al. (2024) studied one-dimensional case and showed that increasing the scale not only reduces diversity of generated distributions but also leads generated data to drift to the extreme points in the support of the conditional distribution. Wu et al. (2024) theoretically analyzed the influence of the guidance scale in the context of Gaussian mixture models, demonstrating that a large guidance scale diminishes distributional diversity while boosting classification confidence. Due to the analytical complexity of the probability guidance term, previous works have focused on special cases. Therefore, we propose a geometric guidance term that plays the same role as the probabilistic guidance but is more amenable to theoretical analysis of the guidance scale.

## 3 Background

### 3.1 Diffusion Model

Let $\boldsymbol{X} \sim \mathbb{P}_X \in \mathcal{P}(\mathbb{R}^D)$ denote the target data distribution. The forward process in denoising diffusion probabilistic models (DDPMs) (Ho et al., 2020) is governed by the stochastic differential equation (SDE)

$$\mathrm{d}\boldsymbol{X}_t = -\frac{1}{2}\beta(t)\boldsymbol{X}_t\mathrm{d}t + \sqrt{\beta(t)}\mathrm{d}\boldsymbol{W}_t, \quad \forall\, t \in [0, T], \tag{1}$$

with the initial condition $\boldsymbol{X}_0 \sim \mathbb{P}_X$, where $(\boldsymbol{W}_t)_{t \geq 0}$ is a standard Brownian motion and $\beta \colon [0, T] \to (0, \infty)$ is smooth; see Song et al. (2021b). This SDE admits the following analytical solution:

$$\boldsymbol{X}_t \overset{\mathrm{d}}{=} \sqrt{\alpha_t}\boldsymbol{X}_0 + \sqrt{1 - \alpha_t}\boldsymbol{\xi}, \quad \forall t \in [0, T], \tag{2}$$

where $\boldsymbol{\xi} \sim \mathcal{N}(\boldsymbol{0}, \boldsymbol{I}_D)$ a standard Gaussian, $\alpha_t := \exp\left(-\int_0^t \beta(s)\mathrm{d}s\right)$, and "$\overset{\mathrm{d}}{=}$" means equal in distribution. The derivation is provided in Appendix B.1.

The reverse process of DDPMs aims to generate $\mathbb{P}_X$, which corresponds to the time-reversal process of (1). To this end, we need to consider the process

$$\boldsymbol{X}_t^{\leftarrow} := \boldsymbol{X}_{T-t}$$

and study its stochastic dynamics. As shown in Anderson (1982) and Haussmann & Pardoux (1986), the process $(\boldsymbol{X}_t^{\leftarrow})_{t \in [0, T]}$ satisfies the following SDE:

$$\mathrm{d}\boldsymbol{X}_t^{\leftarrow} = \left(\frac{1}{2}\beta(T - t)\boldsymbol{X}_t^{\leftarrow} + \beta(T - t)\nabla_{\boldsymbol{x}} \log p_{T-t}(\boldsymbol{X}_t^{\leftarrow})\right)\mathrm{d}t + \sqrt{\beta(T - t)}\mathrm{d}\overline{\boldsymbol{W}}_t, \tag{3}$$

where $p_t$ is the density function of $\boldsymbol{X}_t$, and $(\overline{\boldsymbol{W}}_t)_{t \in [0, T]}$ is the Brownian motion in reverse time. A simplified proof can be found in Tang & Zhao (2024).

In practice, a neural network $\boldsymbol{s}_\theta(t, \cdot)$ with parameter $\theta$ is trained to estimate the score function $\nabla_{\boldsymbol{x}} \log p_t(\cdot)$ using the score matching method (Vincent, 2011). By substituting $\nabla_{\boldsymbol{x}} \log p_t$ with the estimator $\boldsymbol{s}_\theta(t, \cdot)$ in (3), experiments (Song & Ermon, 2019; Song et al., 2021b; Dhariwal & Nichol, 2021) showed that DDPMs achieve state-of-the-art performance in data generation tasks.

## 3.2 Probability Flow ODE

Instead of simulating the stochastic process (3), denoising diffusion implicit models (DDIMs) (Song et al., 2021a) employ a deterministic approach for generation, which corresponds to the following ordinary differential equation (ODE):

$$\frac{\mathrm{d}}{\mathrm{d}t}\boldsymbol{X}_t^{\leftarrow} = \frac{1}{2}\beta(T - t)\left(\boldsymbol{X}_t^{\leftarrow} + \nabla_{\boldsymbol{x}} \log p_{T-t}(\boldsymbol{X}_t^{\leftarrow})\right), \quad \forall t \in [0, T], \tag{4}$$

with the initial condition $\boldsymbol{X}_0^{\leftarrow} \sim p_T$, which is called the probability flow ODE. The evolution of the density functions of $\boldsymbol{X}_t^{\leftarrow}$ under this deterministic process is equivalent to that of the stochastic reverse process (3), as the continuity equation associated with the ODE coincides the Fokker–Planck equation corresponding to the SDE (1); see Song et al. (2021b) for details.

In this paper, we focus on the deterministic dynamics, as the Wasserstein distance used as the main metric makes analyzing the ODE formulation more convenient than the SDE. It naturally extends to the SDE via Itô's formula (Gao et al., 2025). Following Chen et al. (2023c) and Chen et al. (2023a), we consider the Ornstein–Uhlenbeck process by setting $\beta(t) \equiv 2$ in Equation (1) for simplicity, where this constant choice is unimportant, as varying it merely rescales time.

## 3.3 Conditional Diffusion Model

When working with paired data $(\boldsymbol{X}, Y) \sim \mathbb{P}_{XY}$, the goal of conditional generation is to generate the conditional distribution $\mathbb{P}_{X|Y}(\cdot \mid Y)$. In Song et al. (2021b), diffusion models are directly applied to $\mathbb{P}_{X|Y}(\cdot \mid Y)$. Specifically, the forward process (1) is first run with the initial condition $\boldsymbol{X}_0 \sim \mathbb{P}_{X|Y}(\cdot \mid Y)$ to obtain the density functions $p_t^y$ of $\boldsymbol{X}_t$. Then, the stochastic reverse process (3), or the deterministic process (4), is simulated to generate samples from $\mathbb{P}_{X|Y}(\cdot \mid Y)$.

Moreover, these intermediate densities $p_t^y$ admit more explicit expressions. Suppose $(\boldsymbol{X}, Y) \sim \mathbb{P}_{XY}$ and we run the SDE (1) with initial condition $\boldsymbol{X}_0 \sim \mathbb{P}_X = \int \mathbb{P}_{XY}(\cdot, dy)$ to obtain $\boldsymbol{X}_t$. Let $p_t(\boldsymbol{x}_t, y)$ denote the joint density function of $(\boldsymbol{X}_t, Y)$. Then, it can be shown that

$$p_t(\boldsymbol{x}_t \mid y) = p_t^y(\boldsymbol{x}_t);$$

see Appendix B.2 for details.

Therefore, the score function for generating $\mathbb{P}_{X|Y}(\cdot \mid Y)$ can be decomposed as

$$\nabla_{\boldsymbol{x}} \log p_t^y(\boldsymbol{x}) = \nabla_{\boldsymbol{x}} \log p_t(\boldsymbol{x} \mid y) = \nabla_{\boldsymbol{x}} \log p_t(\boldsymbol{x}) + \nabla_{\boldsymbol{x}} \log p_t(y \mid \boldsymbol{x}), \tag{5}$$

where $p_t(\boldsymbol{x})$ is the marginal density of $\boldsymbol{X}_t$ obtained by running (1) with the initial condition $\boldsymbol{X}_0 \sim \mathbb{P}_X$. This term can be estimated using standard methods from unconditional DDPMs. The remaining term, $\nabla_{\boldsymbol{x}} \log p_t(y \mid \boldsymbol{x})$, is known as the guidance term, and there are two main approaches for approximating it: classifier guidance and classifier-free guidance (Dhariwal & Nichol, 2021; Ho & Salimans, 2022). In classifier guidance, a time-dependent classifier is trained to approximate $p_t(y \mid \cdot)$ on all noisy data. In classifier-free guidance, a new neural network $\boldsymbol{s}_\theta(t, \boldsymbol{x}, y)$ is trained to estimate the conditional score $\nabla_{\boldsymbol{x}} \log p_t(\boldsymbol{x} \mid y)$, while $\boldsymbol{s}_\theta(t, \boldsymbol{x}, \emptyset)$ approximates the unconditional score $\nabla_{\boldsymbol{x}} \log p_t(\boldsymbol{x})$. The guidance term is then computed as

$$\nabla_{\boldsymbol{x}} \log p_t(y \mid \boldsymbol{x}) \approx \boldsymbol{s}_\theta(t, \boldsymbol{x}, y) - \boldsymbol{s}_\theta(t, \boldsymbol{x}, \emptyset).$$

In practice, a scaling parameter $\eta > 0$, known as the guidance scale, is typically introduced to control the strength of the guidance term (Dhariwal & Nichol, 2021). When using the deterministic dynamics (4), this modification is mathematically expressed as

$$\frac{\mathrm{d}}{\mathrm{d}t} \boldsymbol{X}_t^{\leftarrow} = \boldsymbol{X}_t^{\leftarrow} + \nabla_{\boldsymbol{x}} \log p_{T-t}(\boldsymbol{X}_t^{\leftarrow}) + \eta \nabla_{\boldsymbol{x}} \log p_{T-t}(y \mid \boldsymbol{X}_t^{\leftarrow}), \quad \forall\, t \in [0, T], \tag{6}$$

with the initial condition $\boldsymbol{X}_0^{\leftarrow} \sim p_T(\cdot \mid y)$.

As mentioned in Section 2, although setting $\eta \neq 1$ may seem counterintuitive from a theoretical perspective, empirical studies (Dhariwal & Nichol, 2021; Ho & Salimans, 2022) have shown that selecting an appropriate value of $\eta$ can significantly improve performance. In particular, increasing the guidance scale $\eta$ enhances the distinguishability of generated samples, but at the cost of reduced diversity (Ho & Salimans, 2022; Chidambaram et al., 2024; Wu et al., 2024). However, theoretical understanding of how the guidance scale $\eta$ influences generation remains limited, due to the analytical complexity of the guidance term $\nabla_{\boldsymbol{x}} \log p_t(y \mid \boldsymbol{x})$ (Chidambaram et al., 2024; Wu et al., 2024).

Therefore, the main objective of this work is to provide a theoretical analysis of the guidance scale $\eta$, under the assumption that the target data concentrate on a low-dimensional linear subspace $\mathcal{M}_y \subset \mathbb{R}^D$, called the target data manifold, i.e., $\operatorname{supp} \mathbb{P}_{X|Y}(\cdot \mid Y = y) \subset \mathcal{M}_y$. This analysis consists of two main steps:

(i) First, we replace the probabilistic guidance term with a geometric guidance term in order to avoid the difficulty of handling $\nabla_{\boldsymbol{x}} \log p_t(y \mid \boldsymbol{x})$ (see Section 4).

(ii) Second, we analyze the modified dynamics under the geometric guidance from two perspectives: (a) how $\eta$ influences the recovery of the target data manifold, and (b) how it affects the distance between the generated distribution and the target distribution (see Section 5).

A central technical challenge in analyzing the geometric guidance dynamics is that $\nabla_{\boldsymbol{x}} \log p_t(\boldsymbol{x})$ may fail to satisfy desirable properties, such as the $L$-Lipschitz continuity (and the log-concavity of $p_t(\boldsymbol{x})$), due to the fact that $p_t(\boldsymbol{x})$ arises from a diffusion process initialized with a multi-modal distribution (Lee et al., 2022; Gao et al., 2025). To address this issue, we introduce a novel technique inspired by mollification in mathematical analysis (Evans, 2018), which yields a surrogate distribution $p_t^\sigma(\boldsymbol{x})$ for which the geometric guidance dynamics is well-posed.

## 4 Geometric Guidance Model

In this section, our main objective is to construct a new guidance term to replace $\nabla_{\boldsymbol{x}} \log p_t(y \mid \boldsymbol{x})$ in Equation (6) from a geometric perspective. Specifically, the key idea is to understand the role that $\nabla_{\boldsymbol{x}} \log p_t(y \mid \boldsymbol{x})$ plays in recovering the target data manifold $\mathcal{M}_y$.

Note that $\nabla_{\boldsymbol{x}} \log p_t(y \mid \boldsymbol{x})$ appears as a component of $\nabla_{\boldsymbol{x}} \log p_t^y(\boldsymbol{x})$ by Equation (5). This motivates us to investigate the geometric interpretation of the score function $\nabla_{\boldsymbol{x}} \log p_t(\boldsymbol{x})$ in the setting of unconditional

DDPMs (see Section 4.1). Based on Equation (5) and a basic property of $p_t(y \mid \boldsymbol{x})$, we then propose a replacement for $\nabla_{\boldsymbol{x}} \log p_t(y \mid \boldsymbol{x})$, which preserves its geometric role but is more tractable for theoretical analysis (see Section 4.2).

## 4.1 Geometric Interpretation of Score Function

To study the geometric properties of the score function $\nabla_{\boldsymbol{x}} \log p_t(\boldsymbol{x})$, we first examine the geometric structure of the noisy data manifolds that arise during the DDPM process. Chung et al. (2022) showed that, under the assumption that the target data lie on $\mathcal{M} \in \mathbb{R}^D$, a linear subspace of the ambient space $\mathbb{R}^D$ with significantly lower dimension, the noisy data $\boldsymbol{X}_t$ concentrate on a hypersurface, i.e., a $(D-1)$-dimensional manifold embedded in $\mathbb{R}^D$, for any $t > 0$. We generalize this result (Chung et al., 2022, Proposition 1) in the following proposition; the proof is provided in Appendix C.1.

**Proposition 1.** *Assume $\boldsymbol{Z} \sim \mathbb{P}^Z$ on $\mathbb{R}^d$, and $\boldsymbol{X} = A\boldsymbol{Z} \sim \mathbb{P}_X$ on $\mathbb{R}^D$ for an $A \in \mathcal{O}^{D \times d}$, i.e., $A \in \mathbb{R}^{D \times d}$ and $A^\top A = \boldsymbol{I}_d$. Define*

$$\mathcal{M}^t := \left\{ \boldsymbol{x} \in \mathbb{R}^D : \left\| (\boldsymbol{I}_D - AA^\top)\boldsymbol{x} \right\| = r(t) \right\},$$

*where $r(t) := \sqrt{(D-d)(1-\alpha_t)}$ and $\alpha_t = e^{-2t}$. Let $\boldsymbol{X}_t$ be generated by the DDPM forward process (1) with the initial condition $\boldsymbol{X}_0 = \boldsymbol{X}$. If $d \ll D$, then $\boldsymbol{X}_t$ concentrates on $\mathcal{M}^t$ with high probability.*

Based on this result, the next question is how the score function $\nabla_{\boldsymbol{x}} \log p_t(\boldsymbol{x})$ contributes to recovering these noisy data manifolds $\mathcal{M}^t$ during the reverse process (4).

Under the same assumptions as those in Proposition 1, Chen et al. (2023b) showed that

$$\nabla_{\boldsymbol{x}} \log p_t(\boldsymbol{x}) = A \, \nabla_{\boldsymbol{z}} \log p_t^Z(\boldsymbol{z}) \big|_{\boldsymbol{z} = A^\top \boldsymbol{x}} - \frac{1}{1 - \alpha_t}(\boldsymbol{I}_D - AA^\top)\boldsymbol{x}, \tag{7}$$

where $p_t^Z$ is the density associated with the forward process (1) initialized from $p^Z$. An alternative derivation of this formula, along with an analysis of its geometric properties, is provided in Appendix C.2.

Based on this orthogonal decomposition, we observe that the role of $\nabla_{\boldsymbol{x}} \log p_t(\boldsymbol{x})$ can be understood as two components: (i) the first term serves as generating the distribution $\mathbb{P}^Z$ in the latent space, and (ii) the second term controls the reconstruction of the noisy data manifolds $\mathcal{M}^t$ in the ambient space. Informally, this decomposition can be summarized as

$$\nabla_{\boldsymbol{x}} \log p_t(\boldsymbol{x}) = \text{Generate Latent Distribution} \ + \ \text{Recover Data Manifolds } \mathcal{M}^t.$$

We formalize this intuition in the following theorem; see the proof in Appendix C.1.

**Theorem 2.** *Under the same setting as that in Proposition 1, let $\boldsymbol{X}_{t,\|}^{\leftarrow} = AA^\top \boldsymbol{X}_t^{\leftarrow}$ and $\boldsymbol{X}_{t,\perp}^{\leftarrow} = \boldsymbol{X}_t^{\leftarrow} - \boldsymbol{X}_{t,\|}^{\leftarrow}$, where $\boldsymbol{X}_t^{\leftarrow} = \boldsymbol{X}_{T-t}$.*

*(a) Let $\boldsymbol{X}_{t,\|}^{\leftarrow} = A\boldsymbol{Z}_t^{\leftarrow}$ with $\boldsymbol{Z}_t^{\leftarrow} = A^\top \boldsymbol{X}_t^{\leftarrow}$. Then $\boldsymbol{Z}_t^{\leftarrow}$ satisfies*

$$\frac{\mathrm{d}}{\mathrm{d}t} \boldsymbol{Z}_t^{\leftarrow} = \boldsymbol{Z}_t^{\leftarrow} + \nabla_{\boldsymbol{z}} \log p_{T-t}^Z(\boldsymbol{Z}_t^{\leftarrow}),$$

*which implies that $\boldsymbol{Z}_t = A^\top \boldsymbol{X}_t = \boldsymbol{Z}_{T-t}^{\leftarrow}$ follows the forward process (1) initialized from $p^Z$.*

*(b) $\boldsymbol{X}_{t,\perp}^{\leftarrow}$ satisfies*

$$\frac{\mathrm{d}}{\mathrm{d}t} \boldsymbol{X}_{t,\perp}^{\leftarrow} = \boldsymbol{X}_{t,\perp}^{\leftarrow} - \frac{1}{1 - \alpha_{T-t}} \boldsymbol{X}_{t,\perp}^{\leftarrow}.$$

*Moreover, $\|\boldsymbol{X}_{t_0,\perp}^{\leftarrow}\| = r(T - t_0)$ implies $\|\boldsymbol{X}_{t_0+\delta,\perp}^{\leftarrow}\| = r(T - t_0 - \delta)$, where $r(t) = \sqrt{(D-d)(1-\alpha_t)}$.*

In Theorem 2, statement (a) demonstrates that the parallel part $\nabla_{\boldsymbol{z}} \log p^Z(\boldsymbol{z})$ in the decomposition (7) is responsible for generating the target latent distribution $p^Z$ via the reverse process of DDPMs, which has been thoroughly studied in Chen et al. (2023b). Meanwhile, statement (b) shows that, since

$$\left\| (\boldsymbol{I}_D - AA^\top)\boldsymbol{X}_t^{\leftarrow} \right\| = \left\| \boldsymbol{X}_{t,\perp}^{\leftarrow} \right\|,$$

the orthogonal part $(\boldsymbol{I}_D - AA^\top)\boldsymbol{x}$ plays a key role in guiding the recovery of the noisy data manifolds $\mathcal{M}^t$, which provides an insight for designing geometric guidance in conditional generation.

## 4.2 Geometric Guidance for Conditional Generation

Let us return to the conditional diffusion model. To apply the results from Section 4.1 in studying the role of $\nabla_{\boldsymbol{x}} \log p_t(y \mid \boldsymbol{x})$ in guidance, we first impose the linear assumption for the target data manifold.

We consider a two-class dataset $(\boldsymbol{X}, Y) \sim \mathbb{P}_{XY}$ on $\mathbb{R}^D \times \{1, 2\}$ for simplicity; the following analysis readily extends to the multi-class case. Let $\mathbb{P}(Y = 1) = w_1$ and $\mathbb{P}(Y = 2) = w_2$ so that

$$\mathbb{P}_X = w_1 \mathbb{P}_{X|Y}(\cdot \mid Y = 1) + w_2 \mathbb{P}_{X|Y}(\cdot \mid Y = 2).$$

The linear assumption states as follows.

**Assumption I.** *For $i = 1, 2$, there exists a $\boldsymbol{Z}_i \sim p_i^Z$ on $\mathbb{R}^{d_i}$ and an $A_i \in \mathcal{O}^{D \times d_i}$ such that*

$$\boldsymbol{X}_i := A_i \boldsymbol{Z}_i \sim \mathbb{P}_{X|Y}(\cdot \mid Y = i),$$

*and we further assume $A_1^\top A_2 = \boldsymbol{O}$.*

*Remark* 1. For this assumption, we provide two remarks.

(i) It basically means that the support $\operatorname{supp} \mathbb{P}_{X|Y}(\cdot \mid Y = i) \subset \mathcal{M}_i := \operatorname{Im} A_i$, the image of $\boldsymbol{x} \mapsto A\boldsymbol{x}$; in other words, $\mathbb{P}_{X|Y}(\cdot \mid Y = i)$ is supported on the linear space $\operatorname{Im} A_i$. The definition of the support of a probability measure is provided in Appendix A.

(ii) $A_1^\top A_2 = \boldsymbol{O}$ indicates $\mathcal{M}_1 \perp \mathcal{M}_2$. This orthogonality assumption is introduced to simplify the subsequent analysis, but it does not significantly affect our conclusions regarding the guidance scale; see Appendix E.1 for further discussion.

Next, we fix $Y = 1$ and our goal is to generate the conditional distribution, which needs to consider the geometric structure of the condition score function $\nabla_{\boldsymbol{x}} \log p_t(\boldsymbol{x} \mid y = 1)$. By combining the results in Section 4.1 with Equation (5), the conditional score function has two different types of decomposition:

$$\begin{aligned} \nabla_{\boldsymbol{x}} \log p_t(\boldsymbol{x} \mid y = 1) &= \text{Generate Latent Distribution } + \text{ Recover Data Manifolds } \mathcal{M}_1^t \\ &= \nabla_{\boldsymbol{x}} \log p_t(\boldsymbol{x}) + \nabla_{\boldsymbol{x}} \log p_t(y = 1 \mid \boldsymbol{x}). \end{aligned} \tag{8}$$

We will show that $\nabla_{\boldsymbol{x}} \log p_t(y = 1 \mid \boldsymbol{x})$ plays the role of recovering the data manifolds $\mathcal{M}_1^t$ with respect to the first decomposition.

For the first decomposition in (8), based on Assumption I and Proposition 1, because the noisy data manifolds generated by the forward process starting from $\mathcal{M}_1$ are given by

$$\mathcal{M}_1^t = \left\{ \boldsymbol{x} \in \mathbb{R}^D : \left\| (\boldsymbol{I}_D - A_1 A_1^\top) \boldsymbol{x} \right\| = r(t) \right\}, \tag{9}$$

the orthogonal part of $\nabla_{\boldsymbol{x}} \log p_t(\boldsymbol{x} \mid y = 1)$ in the first decomposition responsible for recovering $\mathcal{M}_1^t$ is parallel to $(\boldsymbol{I}_D - A_1 A_1^\top) \boldsymbol{x}$ as shown in Section 4.1.

Intuitively, for the second decomposition in (8), since $p_t(y = 1 \mid \boldsymbol{x})$ acts as a classifier for $\mathcal{M}_1^t$, we have $p_t(y = 1 \mid \boldsymbol{x}) \approx 1$ for any $\boldsymbol{x} \in \mathcal{M}_1^t$, i.e., $\log p_t(y = 1 \mid \boldsymbol{x})$ is approximately constant on $\mathcal{M}_1^t$. Therefore, by Lemma I.1, $\nabla_{\boldsymbol{x}} \log p_t(y = 1 \mid \boldsymbol{x})$ is almost normal to $\mathcal{M}_1^t$,

$$\nabla_{\boldsymbol{x}} \log p_t(y = 1 \mid \boldsymbol{x}) \approx -\eta (\boldsymbol{I}_D - A_1 A_1^\top) \boldsymbol{x}, \quad \text{for some } \eta > 0,$$

because $(\boldsymbol{I}_D - A_1 A_1^\top) \boldsymbol{x}$ is normal to $\mathcal{M}_1^t$ by Lemma I.1. Rigorous details are provided in Appendix C.3.

Therefore, the guidance term $\nabla_{\boldsymbol{x}} \log p_t(y = 1 \mid \boldsymbol{x})$ partially contributes to the recovery of the data manifolds $\mathcal{M}_1^t$ during the reverse process. Consequently, it can be replaced by $(\boldsymbol{I}_D - A_1 A_1^\top) \boldsymbol{x}$. Based on this insight, we propose the following geometric guidance model for conditional generation:

$$\frac{\mathrm{d}}{\mathrm{d}t} \boldsymbol{X}_t^\leftarrow = \boldsymbol{X}_t^\leftarrow + \nabla_{\boldsymbol{x}} \log p_{T-t}(\boldsymbol{X}_t^\leftarrow) - \eta P_1 \boldsymbol{X}_t^\leftarrow, \quad P_1 := \boldsymbol{I}_D - A_1 A_1^\top. \tag{10}$$

# 5 Main Results: Analysis of Geometric Guidance Model

In this section, we analyze the geometric guidance model (10) with the aim of uncovering the role of the guidance scale $\eta$. To understand its effects, we consider two related questions: whether the model can approximately estimate the target data manifold $\mathcal{M}_1$ (see Section 5.2), and how to quantify the distance between the generated and target distributions (see Section 5.3). These two problems serve as a lens through which we investigate the influence of $\eta$ in conditional generation.

Before addressing these two questions, it is necessary to ensure the well-posedness of the ODE (10); that is, we must establish regularities of $\nabla_{\boldsymbol{x}} \log p_t(\boldsymbol{x})$ such as its Lipschitz continuity and the log-concavity of $p_t(\boldsymbol{x})$, which requires careful analysis (see Section 5.1) because it is obtained from a multi-modal distribution $\mathbb{P}_X$.

## 5.1 Well-posedness of Geometric Guidance Model

In general, the Lipschitz continuity of $\nabla_{\boldsymbol{x}} \log p_t(\boldsymbol{x})$ and the log-concavity of $p_t(\boldsymbol{x})$ induced by the DDPM forward process depend on properties of the initial distribution $\mu$. A basic requirement is that $\mu$ admit a density $p(\boldsymbol{x})$. Log-concavity of $p(\boldsymbol{x})$ then implies log-concavity of $p_t(\boldsymbol{x})$ (Gao et al., 2025), and Lipschitz continuity of $\nabla_{\boldsymbol{x}} \log p(\boldsymbol{x})$ implies the Lipschitz continuity of $\nabla_{\boldsymbol{x}} \log p_t(\boldsymbol{x})$ (Chen et al., 2023a).

However, in our setting, it is clear that $\mathbb{P}_X$ does not admit a density function. We therefore first deduce the necessary conditions on the latent distribution implied by $\mathbb{P}_X$; see Sections 5.1.1 and 5.1.2. Second, the multi-modality of $\mathbb{P}_X$ introduces irregularities in $p_t(\boldsymbol{x})$ (Lee et al., 2022), which we discuss in Section 5.1.3. By solving these two problems, we construct a surrogate $p_t^\sigma(\boldsymbol{x})$ for use in the geometric guidance model (10), which is well-posed; see Section 5.1.4.

### 5.1.1 Problems in Latent Distribution

When $\mu$ does not admit a density function—for instance, when the support of $\mu$ lies on a lower-dimensional manifold in the ambient space—De Bortoli (2022) showed that the score function $\nabla_{\boldsymbol{x}} \log p_t$ is Lipschitz continuous under the assumption that $\operatorname{supp} \mu$ is compact, i.e., closed and bounded. This setting aligns with our problem but guarantees only Lipschitz continuity. In contrast, we establish a stronger result in the following Proposition 3, which does not require the compactness, under the assumption that the target data manifold is linear. The proof is provided in Appendix D.1.

**Proposition 3.** *Let $\boldsymbol{Z}$ be a random variable on $\mathbb{R}^k$ with the density function $p^Z$, and let $B \in \mathbb{R}^{n \times k}$. Assume there are $m_0, \Lambda > 0$ such that*

$$-\nabla_{\boldsymbol{z}}^2 \log p^Z(\boldsymbol{z}) \succeq m_0 I_k, \quad \|B\|_{\mathrm{op}}^2 \leq \Lambda,$$

*and $\lambda := \lambda_{\min}(B^\top B) \geq 0$, the minimum of all eigenvalues of $B^\top B$. For $\alpha \in \mathbb{R}$ and $\beta > 0$, let*

$$\boldsymbol{X} = \alpha B \boldsymbol{Z} + \beta \boldsymbol{\xi}, \quad \boldsymbol{\xi} \sim \mathcal{N}(\boldsymbol{0}, \boldsymbol{I}_n)$$

*with the density function $p_X$ on $\mathbb{R}^n$. We have*

$$\left\| \nabla_{\boldsymbol{x}}^2 \log p_X(\boldsymbol{x}) \right\|_{\mathrm{op}} \leq L, \quad L := \frac{1}{\beta^2} + \frac{\alpha^2 \Lambda}{\beta^2(\alpha^2 \lambda + m_0 \beta^2)}.$$

*Remark* 2. A direct application of this proposition is that it extends the result of De Bortoli (2022) to a non-compact setting, under the additional assumption that the latent distribution is strongly log-concave, i.e., $-\nabla_{\boldsymbol{z}}^2 \log p^Z(\boldsymbol{z}) \succeq m_0 I_k$. If we are only concerned with the $L$-smoothness [1] of $\log p_X$, the log-concavity of $p^Z$ can be relaxed to the $L$-smoothness; see Appendix F for details.

**Corollary 4.** *Using the same notations as in Proposition 3, we have*

$$\nabla_{\boldsymbol{x}}^2 \log p_X(\boldsymbol{x}) \preceq \left( \frac{\alpha^2 \Lambda}{\beta^2(\alpha^2 \lambda + m_0 \beta^2)} - \frac{1}{\beta^2} \right) \boldsymbol{I}_n.$$

---

[1] $L$-smoothness of $f$ and $L$-Lipschitz continuity of $\nabla f$ are equivalent for $C^2$ functions; we use them interchangeably.

*Remark* 3. Note that, if $\alpha^2 \Lambda < \alpha^2 \lambda + m_0 \beta^2$, such as $\Lambda = \lambda$, then

$$-\nabla_{\boldsymbol{x}}^2 \log p_X(\boldsymbol{x}) \succeq m_x \boldsymbol{I}_n, \quad m_x := \frac{1}{\beta^2}\left(1 - \frac{\alpha^2 \Lambda}{\alpha^2 \lambda + m_0 \beta^2}\right) > 0,$$

which implies that $p_X$ is $m_x$-strongly log-concave. Therefore, it shows that the strong log-concavity of $p^Z$ ensures not only the $L$-smoothness, but also the concavity of $\log p_X$.

Based on results in Proposition 3, even if $\mathbb{P}_X$ does not admit a density function, the desired properties of the score function can still be guaranteed, provided that the latent distribution admits a density and satisfies strong log-concavity. However, in our setting, these two conditions are not satisfied:

(i) For the latent distribution of $\mathbb{P}_X$, because $\boldsymbol{Z}_i \sim \mathbb{P}_i^Z$ on $\mathbb{R}^{d_i}$, we first lift them on $\mathbb{R}^d$ with $d := d_1 + d_2$ by defining
$$\tilde{\boldsymbol{Z}}_1 = (\boldsymbol{I}_{d_1}, \boldsymbol{O}_{d_1 \times d_2})^\top \boldsymbol{Z}_1 \sim \tilde{\mathbb{P}}_1^Z, \quad \tilde{\boldsymbol{Z}}_2 = (\boldsymbol{O}_{d_2 \times d_1}, \boldsymbol{I}_{d_2})^\top \boldsymbol{Z}_2 \sim \tilde{\mathbb{P}}_2^Z.$$
Let $A = (A_1, A_2) \in \mathcal{O}^{D \times d}$. It follows that
$$A\tilde{\boldsymbol{Z}}_i = A_i \boldsymbol{Z}_i \sim \mathbb{P}_{X|Y}(\cdot \mid Y = i), \text{ i.e., } A_\# \tilde{\mathbb{P}}_i^Z = \mathbb{P}_{X|Y}(\cdot \mid Y = i).$$
Therefore, by Lemma H.7, if $\boldsymbol{Z} \sim \mathbb{P}^Z := w_1 \tilde{\mathbb{P}}_1^Z + w_2 \tilde{\mathbb{P}}_2^Z$, we have
$$\boldsymbol{X} = A\boldsymbol{Z} \sim \mathbb{P}_X = w_1 \mathbb{P}_{X|Y}(\cdot \mid Y = 1) + w_2 \mathbb{P}_{X|Y}(\cdot \mid Y = 2).$$
But the problem is that the latent distribution $\mathbb{P}^Z$ does not admit a density function on $\mathbb{R}^d$.

(ii) For log-concavity, even if the latent distribution admits a density function, it typically does not satisfy strong log-concavity due to its multi-modality (Lee et al., 2022).

Therefore, in the following, we first introduce a technique to address the log-concavity of the latent density (Sections 5.1.2 and 5.1.3), and then apply Proposition 3 to establish the desired properties of the score function (Section 5.1.4).

### 5.1.2 Mollification Technique

Mollification (Evans, 2018) is a standard technique in mathematical analysis to address non-smoothness of functions. When dealing with a non-smooth function $f$, the idea is to find a smooth kernel function $k$ such that the convolution $g := f * k$, which is clearly smooth, is closed to $f$.

Following this idea, we choose a Gaussian distribution $\mathcal{N}(\boldsymbol{0}, \sigma^2 \boldsymbol{I}_d)$ with some $\sigma > 0$ as the kernel, and consider its convolution with $\tilde{\mathbb{P}}^Z$; see Remark B.1 for the definition of convolution between measures. Let

$$\mathbb{P}_\sigma^Z := \tilde{\mathbb{P}}^Z * \mathcal{N}(\boldsymbol{0}, \sigma^2 \boldsymbol{I}_d) = w_1 \mathbb{P}_{1,\sigma}^Z + w_2 \mathbb{P}_{2,\sigma}^Z,$$

where $\mathbb{P}_{i,\sigma}^Z := \tilde{\mathbb{P}}_i^Z * \mathcal{N}(\boldsymbol{0}, \sigma^2 \boldsymbol{I}_d)$. Note that both $\mathbb{P}_\sigma^Z$ and $\mathbb{P}_{i,\sigma}^Z$ admit density functions, denoted by $p_\sigma^Z$ and $p_{i,\sigma}^Z$ respectively, and

$$p_\sigma^Z = w_1 p_{1,\sigma}^Z + w_2 p_{2,\sigma}^Z. \tag{11}$$

Moreover, by the definition of convolution, if $\boldsymbol{Z}_i \sim p_i^Z$, then

$$\boldsymbol{Z}_{1,\sigma} := (\boldsymbol{Z}_1, \boldsymbol{O})^\top + \sigma \boldsymbol{\zeta}_1 \sim p_{1,\sigma}^Z, \quad \boldsymbol{Z}_{2,\sigma} := (\boldsymbol{O}, \boldsymbol{Z}_2)^\top + \sigma \boldsymbol{\zeta}_2 \sim p_{2,\sigma}^Z, \tag{12}$$

for $\boldsymbol{\zeta}_i \sim \mathcal{N}(\boldsymbol{0}, \boldsymbol{I}_d)$ independent of $\boldsymbol{Z}_i$. Therefore, we obtain a smooth density $p_\sigma^Z$ on the latent space $\mathbb{R}^d$.

Next, for $\mathbb{P}_X$, the following Proposition 5 addresses the question of whether sampling from $p_\sigma^Z$ yields a $\mathbb{P}_X^\sigma := A_\# \mathbb{P}_\sigma^Z$ that is close $\mathbb{P}_X$. The proof is provided in Appendix D.1.

To measure the distance between probability measures, we use the 1-Wasserstein distance in this work for analytical convenience. For $\mu, \nu \in \mathcal{P}(\mathbb{R}^D)$, it is defined by

$$\mathcal{W}_1(\mu, \nu) := \inf\left\{\int \|\boldsymbol{x} - \boldsymbol{y}\| \mathrm{d}\gamma(\boldsymbol{x}, \boldsymbol{y}) : \gamma \in \Gamma(\mu, \nu)\right\} = \inf\left\{\mathbb{E}\left[\|\boldsymbol{X} - \boldsymbol{Y}\|\right] : \boldsymbol{X} \sim \mu, \boldsymbol{Y} \sim \nu\right\},$$

where $\Gamma(\mu, \nu) := \{\gamma \in \mathcal{P}(\mathbb{R}^D \times \mathbb{R}^D) \colon \gamma(A \times \mathbb{R}^D) = \mu(A),\ \gamma(\mathbb{R}^D \times B) = \nu(B)\}$; see Chewi et al. (2024) for more details.

**Proposition 5.** *Using the above notation, if $\boldsymbol{Z}_\sigma \sim \mathbb{P}_\sigma^Z$, then for $\boldsymbol{X}^\sigma := A\boldsymbol{Z}_\sigma \sim \mathbb{P}_X^\sigma$, we have*

$$\mathbb{P}_X^\sigma = w_1 \mathbb{P}_{X|Y}^\sigma(\cdot \mid Y = 1) + w_2 \mathbb{P}_{X|Y}^\sigma(\cdot \mid Y = 2),$$

*where $\mathbb{P}_{X|Y}^\sigma(\cdot \mid Y = i) := A_\# \mathbb{P}_{i,\sigma}^Z$ for $i = 1, 2$, and it follows that*

$$\mathcal{W}_1(\mathbb{P}_X^\sigma, \mathbb{P}_X) \leq \sigma \sqrt{d}.$$

Therefore, the mollification technique provides a smooth latent density function $p_\sigma^Z$ that induces a distribution $\mathbb{P}_X^\sigma$ approximating $\mathbb{P}_X$.

### 5.1.3 Log-Concavity of Latent Density

In general, even if $p_\sigma^Z$ is smooth, we cannot directly assume that it is strongly log-concave, as it is multi-modal by Equation (11). However, we can still assume that each of its components $p_{i,\sigma}^Z$ is strongly log-concave, which, in fact, follows from the assumption of strong log-concavity of the original latent density $p_i^Z$.

**Assumption II.** *Let $p_i^Z$ be the density function of $\mathbb{P}_i^Z$ defined on $\mathbb{R}^{d_i}$. There exits a large $m > 1$ such that*

$$-\nabla_{\boldsymbol{z}}^2 \log p_i^Z(\boldsymbol{z}) \succeq m\boldsymbol{I}_{d_i},$$

*i.e., $p_i^Z$ is $m$-strongly log-concave for $i = 1, 2$.*

Assumption II ensures the strong log-concavity of each component $p_{i,\sigma}^Z$, but it does not guarantee that the overall mixture $p_\sigma^Z$ is strongly log-concave—this is a common difficulty in the case of multi-modal distributions. However, due to the mollification construction, the parameter $\sigma$ can be freely chosen, which enables us to establish the strong log-concavity of $p_\sigma^Z$ under the following assumption.

**Assumption III.** *For a chosen $\sigma$, we assume that*

$$M := \sup_{\boldsymbol{z}} \left\| \nabla_{\boldsymbol{z}} \log p_{1,\sigma}^Z(\boldsymbol{z}) - \nabla_{\boldsymbol{z}} \log p_{2,\sigma}^Z(\boldsymbol{z}) \right\| < 2\sqrt{m-1}.$$

*Remark* 4. This assumption is novel but essential for addressing log-concavity in multi-modal settings. Characterizing the classes of $p_i^Z$ that satisfy it is nontrivial. As a concrete example, if each $p_i^Z(z)$ is a Gaussian truncated to a compact, convex set, then compactness implies that the difference of $\nabla \log p_{i,\sigma}^Z$ is uniformly bounded by a quantity depending on $\sigma$. Therefore, with an appropriate choice of $\sigma$, Assumption III holds. The details, with a sufficient condition for Assumption III, are discussed in Appendix E.2.

Assumption III is required to obtain an upper bound on $\nabla^2 \log p$, even when the density $p$ is multi-modal, as shown in the following lemma; see the proof in Appendix D.1.

**Lemma 6.** *Let $p_1, p_2$ be two probability density functions on $\mathbb{R}^n$ such that $\nabla^2 \log p_i \preceq L_i \boldsymbol{I}_n$ for some constant $L_i \in \mathbb{R}$. Suppose that*

$$\sup_{\boldsymbol{x}} \left\| \nabla \log p_1(\boldsymbol{x}) - \nabla \log p_2(\boldsymbol{x}) \right\| \leq M < \infty.$$

*Then, for the mixture density $p = wp_1 + (1-w)p_2$ with $w \in (0, 1)$, it holds that*

$$\nabla^2 \log p \preceq \left( \max\{L_1, L_2\} + \frac{1}{4}M^2 \right) \boldsymbol{I}_n.$$

By Lemma 6 and Proposition 3, the strong log-concavity of the multi-modal latent density function $p_\sigma^Z$ can be guaranteed.

**Theorem 7.** *Under Assumptions II and III, if $\sigma^2 < (4m - M^2)/(M^2 m)$, then $p_\sigma^Z$ is strongly log-concave for $p_\sigma^Z = w_1 p_{1,\sigma}^Z + w_2 p_{2,\sigma}^Z$, i.e.,*

$$-\nabla_{\boldsymbol{z}}^2 \log p_\sigma^Z(\boldsymbol{z}) \succeq m_0^z \boldsymbol{I}_d, \quad m_0^z := \frac{4m - M^2(1 + m\sigma^2)}{4(1 + m\sigma^2)}.$$

*Proof.* Note that

$$\boldsymbol{Z}_{1,\sigma} = (\boldsymbol{I}_{d_1}, \boldsymbol{O})^\top \boldsymbol{Z}_1 + \sigma\boldsymbol{\zeta}_1 \sim p^Z_{1,\sigma}.$$

By Assumption II and Corollary 4, with the choices $B = (\boldsymbol{I}_{d_1}, \boldsymbol{O})^\top$, $m_0 = m$, $\alpha = 1$, and $\beta = \sigma$, we obtain

$$\nabla^2_{\boldsymbol{z}} \log p^Z_{1,\sigma}(\boldsymbol{z}) \preceq \left( \frac{1}{\sigma^2(1+m\sigma^2)} - \frac{1}{\sigma^2} \right) \boldsymbol{I}_d.$$

For $p^Z_{2,\sigma}$, we similarly have

$$\nabla^2_{\boldsymbol{z}} \log p^Z_{2,\sigma}(\boldsymbol{z}) \preceq \left( \frac{1}{\sigma^2(1+m\sigma^2)} - \frac{1}{\sigma^2} \right) \boldsymbol{I}_d.$$

Then, because $p^Z_\sigma = w_1 p^Z_{1,\sigma} + w_2 p^Z_{2,\sigma}$, it follows from Assumption III and Lemma 6 that

$$\nabla^2_{\boldsymbol{z}} \log p^Z_\sigma(\boldsymbol{z}) \preceq \left( \frac{1}{\sigma^2(1+m\sigma^2)} - \frac{1}{\sigma^2} + \frac{1}{4}M^2 \right) \boldsymbol{I}_d = -m^z_0 \boldsymbol{I}_d. \qquad \square$$

### 5.1.4 Smoothness and Concavity

Before proceeding, let us recall that the latent distribution $\mathbb{P}^Z$ of $\mathbb{P}_X$ is not "good". To address this, we construct a new distribution $\mathbb{P}^\sigma_X$ whose latent distribution $\mathbb{P}^Z_\sigma$ admits a "good" density function $p^Z_\sigma$, and which is close to $\mathbb{P}_X$. Consequently, instead of considering the score function associated with a DDPM initialized from $\mathbb{P}_X$, we consider a DDPM initialized from $\mathbb{P}^\sigma_X$, i.e.,

$$\boldsymbol{X}^\sigma_t = \sqrt{\alpha_t} A\boldsymbol{Z}_\sigma + \sqrt{1-\alpha_t}\boldsymbol{\xi} \quad \sim \quad p^\sigma_t, \tag{13}$$

where $\boldsymbol{Z}_\sigma \sim p^Z_\sigma$ and $\boldsymbol{\xi} \sim \mathcal{N}(\boldsymbol{0}, \boldsymbol{I}_D)$. We then modify the dynamics in (10) to define our final version of the geometric guidance model:

**Definition 1** (Geometric Guidance Model). For any $t \in [0, T-\delta]$,

$$\frac{\mathrm{d}}{\mathrm{d}t}\tilde{\boldsymbol{X}}_t = \tilde{\boldsymbol{X}}_t + \nabla_{\boldsymbol{x}} \log p^\sigma_{T-t}(\tilde{\boldsymbol{X}}_t) - \eta P_1 \tilde{\boldsymbol{X}}_t, \quad \tilde{\boldsymbol{X}}_0 \sim \mathcal{N}(\boldsymbol{0}, \boldsymbol{I}_D), \tag{$*$}$$

where $P_1 = \boldsymbol{I}_D - A_1 A_1^\top$.

*Remark* 5. (i) The initial condition is taken as $\mathcal{N}(\boldsymbol{0}, \boldsymbol{I}_D)$ instead of $p_T(\cdot \mid y)$ to reflect practical implementation settings. (ii) The time interval is chosen as $[0, T-\delta]$ for some $\delta > 0$ to avoid the singularity at time $T$.

Therefore, our main objective now becomes establishing the Lipschitz continuity of $\nabla_{\boldsymbol{x}} \log p^\sigma_t(\boldsymbol{x})$ and the log-concavity of $p^\sigma_t(\boldsymbol{x})$, which follows from the strong log-concavity of the latent density $p^Z_\sigma(\boldsymbol{z})$.

**Theorem 8.** *Under Assumption I and the same settings as in Theorem 7, for the density function $p^\sigma_t$ defined in Equation (13), we have*

$$\left\| \nabla^2_{\boldsymbol{x}} \log p^\sigma_t(\boldsymbol{x}) \right\|_{\mathrm{op}} \le L_t, \quad L_t := \frac{2\alpha_t + (1-\alpha_t)m^z_0}{(1-\alpha_t)\left(\alpha_t + (1-\alpha_t)m^z_0\right)},$$

*and*

$$-\nabla^2_{\boldsymbol{x}} \log p^\sigma_t(\boldsymbol{x}) \succeq m_t \boldsymbol{I}_D, \quad m_t := \frac{m^z_0}{\alpha_t + (1-\alpha_t)m^z_0}.$$

*Proof.* First, by Theorem 7, the latent density $p^Z_\sigma$ is $m^z_0$-strongly log-concave. By the definition of $p^\sigma_t$, Proposition 3 implies that

$$\left\| \nabla^2_{\boldsymbol{x}} \log p^\sigma_t(\boldsymbol{x}) \right\|_{\mathrm{op}} \le \frac{2\alpha_t + (1-\alpha_t)m^z_0}{(1-\alpha_t)\left(\alpha_t + (1-\alpha_t)m^z_0\right)},$$

with the choices $B = A$, $m_0 = m^z_0$, $\alpha = \sqrt{\alpha_t}$, and $\beta = \sqrt{1-\alpha_t}$. This follows from the fact that $A^\top A = \boldsymbol{I}_d$ (Assumption I), which indicates $\|A\|^2_{\mathrm{op}} = 1$ and $\lambda_{\min}(A^\top A) = 1$.

For the log-concavity, Corollary 4 directly yields

$$-\nabla^2_{\boldsymbol{x}} \log p^\sigma_t(\boldsymbol{x}) \succeq \left( \frac{1}{1-\alpha_t}\left( 1 - \frac{\alpha_t}{\alpha_t + (1-\alpha_t)m^z_0} \right) \right) \boldsymbol{I}_D. \qquad \square$$

Therefore, we have established the desired properties of $p_t^\sigma$, which ensure the well-posedness of the geometric guidance model (∗). Moreover, from the definition of $m_t$ in Theorem 8, we can derive a lower bound that will be useful in the following analysis; see Appendix D.1 for the proof.

**Corollary 9.** *There exists a small $\sigma > 0$ such that $m_0^z > 1$ and $m_I := \inf_{t \in (0,T]} m_t > 1$.*

## 5.2 Estimating Target Data Manifold

For the geometric guidance model (∗), the first problem is whether it can estimate the target data manifold $\mathcal{M}_1$. Specifically, we aim to show that the generated sample $\tilde{\boldsymbol{X}}_{T-\delta}$ approximately lies in $\mathcal{M}_1$. Since $\mathcal{M}_1 =$ Im $A_1$ is a linear subspace by Assumption I, it suffices to examine whether $\mathbb{E}\left[\left\|\tilde{\boldsymbol{Y}}_{T-\delta}\right\|\right] \approx 0$, where

$$\tilde{\boldsymbol{Y}}_t = P_1 \tilde{\boldsymbol{X}}_t, \ P_1 = \boldsymbol{I}_D - A_1 A_1^\top.$$

Multiplying both sides of Equation (∗) by $P_1$, we obtain that $\tilde{\boldsymbol{Y}}_t$ satisfies the following dynamics:

$$\frac{\mathrm{d}}{\mathrm{d}t}\tilde{\boldsymbol{Y}}_t = \tilde{\boldsymbol{Y}}_t + P_1 \nabla_{\boldsymbol{x}} \log p_{T-t}^\sigma(\tilde{\boldsymbol{X}}_t) - \eta\tilde{\boldsymbol{Y}}_t, \quad \tilde{\boldsymbol{Y}}_0 \sim \mathcal{N}(0, P_1), \tag{14}$$

for $t \in [0, T - \delta]$. By analyzing the dynamics (14), the following theorem provides a convergence rate of $\mathbb{E}\left[\left\|\tilde{\boldsymbol{Y}}_{T-\delta}\right\|\right] \to 0$ with respect to the guidance scale $\eta$.

**Theorem 10.** *Consider the dynamics (14) under Assumptions II and III. Then,*

$$\mathbb{E}\left[\left\|\tilde{\boldsymbol{Y}}_{T-\delta}\right\|\right] \leq \mathcal{O}\left(e^{-\eta} + \frac{1}{\eta}\right).$$

*In particular, for any $\varepsilon > 0$, by choosing $\eta = \Theta(\max\{\log(1/\varepsilon), 1/\varepsilon\})$, $\mathbb{E}\left[\left\|\tilde{\boldsymbol{Y}}_{T-\delta}\right\|\right] < \varepsilon$.*

*Proof sketch.* We provide a sketch of the proof here; the full proof is given in Appendix D.2.

The key idea is to derive a differential inequality for $\mathbb{E}\left[\left\|\tilde{\boldsymbol{Y}}_t\right\|\right]$. First, we have

$$\frac{\mathrm{d}}{\mathrm{d}t}\mathbb{E}\left[\left\|\tilde{\boldsymbol{Y}}_t\right\|\right] \leq (1 - \eta)\mathbb{E}\left[\left\|\tilde{\boldsymbol{Y}}_t\right\|\right] + \mathbb{E}\left[\left\|\nabla_{\boldsymbol{x}} \log p_{T-t}^\sigma(\tilde{\boldsymbol{X}}_t)\right\|\right]. \tag{15}$$

To bound $\mathbb{E}\left[\left\|\nabla_{\boldsymbol{x}} \log p_{T-t}^\sigma(\tilde{\boldsymbol{X}}_t)\right\|\right]$, the $L_t$-smoothness of $\log p_t^\sigma$ is required, which follows from Theorem 8. The smoothness implies that

$$\left\|\nabla_{\boldsymbol{x}} \log p_{T-t}^\sigma(\tilde{\boldsymbol{X}}_t)\right\| \leq L_S\left\|\tilde{\boldsymbol{X}}_t\right\| + C$$

for some constants $L_S$ and $C$. Therefore, it suffices to bound $\mathbb{E}\left[\left\|\tilde{\boldsymbol{X}}_t\right\|\right]$. By deriving a differential inequality from Equation (∗) and applying Grönwall's inequality (Lemma H.11), we obtain $\mathbb{E}\left[\left\|\tilde{\boldsymbol{X}}_t\right\|\right] \leq M_1$ for some constant $M_1$, and thus $\mathbb{E}\left[\left\|\nabla_{\boldsymbol{x}} \log p_{T-t}^\sigma(\tilde{\boldsymbol{X}}_t)\right\|\right] \leq M_2$. Substituting this bound into (15) and applying Grönwall's inequality once more yields the desired result. $\qquad\square$

*Remark* 6. For this theorem, we provide two remarks.

(i) Note that this result depends only on the $L_t$-smoothness of $\log p_t^\sigma$, and not on strong log-concavity. Therefore, Assumptions II and III can be relaxed; see further discussion in Appendix F.

(ii) The universal guidance model,

$$\frac{\mathrm{d}\boldsymbol{X}_t^\leftarrow}{\mathrm{d}t} = \boldsymbol{X}_t^\leftarrow + \nabla_{\boldsymbol{x}} \log p_{T-t}(\boldsymbol{X}_t^\leftarrow) - \eta\nabla f(\boldsymbol{X}_t^\leftarrow),$$

was proposed by Bansal et al. (2023) to control the generation process such that the generated images match the prompt $g(\boldsymbol{X}_T^\leftarrow) \approx \boldsymbol{c}$. In their setting, $f(\boldsymbol{x}) = \ell(\boldsymbol{c}, g(\boldsymbol{x}))$ for some loss function $\ell$. A similar idea used in the proof of Theorem 10 can be extended to theoretically analyze the universal guidance model. If the $L$-smoothness of $\log p_t$ holds (see Appendix F) and $f$ is strongly convex,

$$\mathbb{E}\left[f(\boldsymbol{X}_T^\leftarrow)\right] \to \min f, \text{ as } \eta \to \infty;$$

see Appendix D.3 for more details.

Theorem 10 shows that the geometric guidance model can approximate the target data manifold. Specifically, as the guidance scale $\eta$ increases, the generated data increasingly lie close to the target manifold. This result is consistent with empirical observations on both synthetic datasets (Wu et al., 2024; Chidambaram et al., 2024) and real-world datasets (Dhariwal & Nichol, 2021; Sadat et al., 2024; 2025), as well as with the theoretical results in the one-dimensional case studied by Chidambaram et al. (2024), which demonstrate that increasing $\eta$ causes the generated data to move toward the extreme points in the support of the target conditional distribution.

### 5.3 Distance to Target Distribution

Let $\tilde{p}_t$ be the density function of $\tilde{X}_t$ in the geometric guidance model ($*$). The second question is how to measure the 1-Wasserstein distance between the generated density $\tilde{p}_{T-\delta}$ and the target conditional distribution $\mathbb{P}_{X|Y}(\cdot \mid Y = 1)$. Specifically, the goal is to provide an upper bound on $\mathcal{W}_1\left(\tilde{p}_{T-\delta}, \mathbb{P}_{X|Y}(\cdot \mid Y = 1)\right)$.

First, we require an additional assumption: the boundedness of the first moment of each conditional distribution $\mathbb{P}_{X|Y}(\cdot \mid Y = i)$, which can be reduced to the same condition on the latent distribution $p_i^Z$.

**Assumption IV.** *For $i = 1, 2$ and $\boldsymbol{Z}_i \sim p_i^Z$, $\mathfrak{m}_i^Z := \mathbb{E}\left[\|\boldsymbol{Z}_i\|\right] < \infty$.*

**Theorem 11.** *Under Assumptions I, II, III, and IV, we obtain that*

$$\mathcal{W}_1\left(\tilde{p}_{T-\delta}, \mathbb{P}_{X|Y}(\cdot \mid Y = 1)\right) \leq \mathcal{O}(e^{-T} + \delta^{1/2} + \sigma + \eta^{-1}) + \tilde{C}$$

*for some constant $\tilde{C}$.*

*Proof sketch.* The proof consists of two main steps:

(i) Let $Q_1 = A_1 A_1^\top$ be the orthogonal projection onto $\mathcal{M}_1 = \operatorname{Im} A_1$. By Theorem 10, we have

$$\mathcal{W}_1\left(\tilde{p}_{T-\delta}, \mathbb{P}_{X|Y}(\cdot \mid Y = 1)\right) \leq \mathcal{W}_1\left((Q_1)_\# \tilde{p}_{T-\delta}, \mathbb{P}_{X|Y}(\cdot \mid Y = 1)\right) + \mathcal{O}(e^{-T} + \eta^{-1}).$$

(ii) For $\mathcal{W}_1\left((Q_1)_\# \tilde{p}_{T-\delta}, \mathbb{P}_{X|Y}(\cdot \mid Y = 1)\right)$, it has

$$\mathcal{W}_1\left((Q_1)_\# \tilde{p}_{T-\delta}, \mathbb{P}_{X|Y}(\cdot \mid Y = 1)\right) \leq \mathcal{W}_1\left(\tilde{p}_{T-\delta}, \mathbb{P}_X\right) + \mathcal{W}_1\left((Q_1)_\# \mathbb{P}_X, \mathbb{P}_{X|Y}(\cdot \mid Y = 1)\right),$$

where the first term $\mathcal{W}_1\left(\tilde{p}_{T-\delta}, \mathbb{P}_X\right)$ can be bounded by comparing the geometric guidance model ($*$) with the unconditional reverse dynamics, and the second term is directly bounded by Lemma D.3.

The full proof is provided in Appendix D.4. □

*Remark* 7. Since geometric guidance cannot carry as much information as probabilistic guidance due to its analytical simplicity, the error floor $\tilde{C}$ does not vanish as $\eta = 1$, unlike in probabilistic guidance models; see further discussion in Remark D.1.

This result suggests that increasing the guidance scale does not harm the generating performance, which may appear counterintuitive and inconsistent with empirical observations. In practice, however, ODE dynamics are typically approximated using the Euler discretization (or the Euler–Maruyama scheme for SDEs), which introduces additional discretization error. In our setting, the Euler discretization error for the geometric guidance model ($*$) is bounded by $\mathcal{O}(h\eta^2)$, where $h$ denotes the step size; see Appendix D.5 for details. Therefore, the performance degradation observed at large guidance scales arises not from the model formulation itself, but from the discretization algorithm. For example, Wu et al. (2024, Figure 3) showed that the large guidance scale would harm the modality of the original data, but this problem can be mitigated by reducing the discretization step size.

## 6 Nonlinear Extension

In this section, our main objective is to construct a nonlinear geometric guidance model suitable for real-world image datasets, and to evaluate its generation performance under varying guidance scales $\eta$.

The first challenge is to construct the geometric guidance term for image datasets, which may not lie in a linear subspace. To this end, we study the geometric structure of noisy data manifolds without assuming linearity of the target data manifold, by extending the result of Proposition 1 to the nonlinear case (see Section 6.1). Then, following the idea of Ross et al. (2024), we train functions $F_\theta^t \colon \mathbb{R}^D \to \mathbb{R}$ to model noisy data manifolds via $\mathcal{M}^t = (F_\theta^t)^{-1}(0)$ so that $\nabla_{\boldsymbol{x}} F_\theta^t$ can replace $(\boldsymbol{I}_D - AA^\top)\boldsymbol{x}$ to be the nonlinear geometric guidance term (see Section 6.2). Finally, we examine this nonlinear geometric guidance model on CIFAR-10 (Krizhevsky, 2009), and evaluate its performance under the different guidance scale (see Section 6.3).

## 6.1 Noisy Data Manifolds for Nonlinear Case

The geometric guidance term $(\boldsymbol{I}_D - AA^\top)\boldsymbol{x}$ is constructed based on the result in Proposition 1, which assumes that the target data manifold $\mathcal{M} = \operatorname{Im} A$ is linear. However, for real-world image datasets, it may unrealistic to assume that the data lie in a linear subspace. Instead, it is more reasonable to assume that the target image data lie on a nonlinear manifold $\mathcal{M} \subset \mathbb{R}^D$ with intrinsic dimension $d \ll D$; see Appendix I for basic knowledge of manifolds. This assumption is known as the manifold hypothesis (Bengio et al., 2013), and it has been supported by both theoretical analyses (Fefferman et al., 2016) and empirical studies (Brown et al., 2022; Loaiza-Ganem et al., 2022).

To construct a new geometric guidance term, because of the nonlinearity of $\mathcal{M}$, we must extend the result of Proposition 1 to uncover the geometric structure of noisy data manifolds. Although the $d$-dimensional manifold $\mathcal{M} \subset \mathbb{R}^D$ is not assumed to be linear, we additionally require that it is locally isometric to $\mathbb{R}^d$. More precisely, we assume the existence of a $C^\infty$ function $\phi \colon \mathbb{R}^d \to \mathbb{R}^D$ such that $\operatorname{Im} \phi = \mathcal{M}$ and $\phi$ is an isometry; that is, $J\phi^\top J\phi \equiv \boldsymbol{I}_d$. Then, by Lemma 12, we obtain an analogue of Proposition 1 in Theorem 13, which shows that the noisy data manifolds $\mathcal{M}^t$ are hypersurfaces—i.e., $(D-1)$-dimensional submanifolds of $\mathbb{R}^D$; see the proofs in Appendix G.1.

**Lemma 12.** *Let $\phi \colon \mathbb{R}^d \to \mathbb{R}^D$ be a $C^\infty$ isometry such that $\mathcal{M} = \operatorname{Im} \phi \subset \mathbb{R}^D$ is a $d$-dimensional submanifold. Then, there exists a $C^\infty$ function $\phi^* \colon \mathbb{R}^D \to \mathbb{R}^d$ such that $\phi^* \circ \phi = \operatorname{id}_{\mathbb{R}^d}$ and*

$$J\phi^*(\phi(\boldsymbol{z})) = J\phi(\boldsymbol{z})^\top, \quad \forall \, \boldsymbol{z} \in \mathbb{R}^d.$$

*Remark* 8. In fact, the isometry of $\phi$ implies that $\operatorname{Im} \phi$ is a submanifold, because it is proper (i.e., the preimage of every compact set is compact) by the Hopf–Rinow theorem (Jost, 2008).

**Theorem 13.** *Let $\mathcal{M} \subset \mathbb{R}^D$ be a $d$-dimensional submanifold as defined in Lemma 12, and let $\mathbb{P}_X$ on $\mathbb{R}^D$ such that $\operatorname{supp} \mathbb{P}_X \subset \mathcal{M}$. Let $\boldsymbol{X}_t$ be generated by DDPM (1) initialized from $\mathbb{P}_X$. If $d \ll D$, then $\boldsymbol{X}_t$ concentrates on a hypersurface $\mathcal{M}^t \subset \mathbb{R}^D$ with high probability, where*

$$\mathcal{M}^t := \left\{ \boldsymbol{x} \colon f^t(\boldsymbol{x}) = r(t) \right\}, \quad r(t) = \sqrt{(D-d)(1-\alpha_t)},$$

*for some $C^\infty$ function $f^t \colon \mathbb{R}^D \to \mathbb{R}$.*

## 6.2 Learning Geometric Guidance

For an image dataset $(\boldsymbol{X}, Y) \sim \mathbb{P}_{XY}$ with class label $Y \in \{1, 2, \ldots, K\}$, we adopt the union of manifold hypothesis (Brown et al., 2022), that is,

$$\operatorname{supp} \mathbb{P}_{X|Y}(\cdot \mid Y = y) \subset \mathcal{M}_y,$$

where $\mathcal{M}_y \subset \mathbb{R}^D$ is a $d_y$-dimensional submanifold. To apply Theorem 13, we further assume that, for each $\mathcal{M}_y$, there exists an isometry $\phi_y \colon \mathbb{R}^{d_y} \to \mathbb{R}^D$ such that $\operatorname{Im} \phi_y = \mathcal{M}_y$. Then, the noisy data manifolds generated by the forward process initialized from $\mathcal{M}_y$ are given by

$$\mathcal{M}_y^t := \left\{ \boldsymbol{x} \in \mathbb{R}^D \colon f_y^t(\boldsymbol{x}) = r(t) \right\}, \quad r(t) = \sqrt{(D-d)(1-\alpha_t)},$$

for some function $f_y^t \colon \mathbb{R}^D \to \mathbb{R}$.

By adopting the same idea as in Section 4.2, for $\boldsymbol{x} \in \mathcal{M}_y^t$, the guidance term $\nabla_{\boldsymbol{x}} \log p_t(y \mid \boldsymbol{x})$ is approximately normal to $\mathcal{M}_y^t$ at $\boldsymbol{x}$—that is, it is approximately parallel to $\nabla_{\boldsymbol{x}} f_y^t(\boldsymbol{x})$. Therefore, we construct the nonlinear

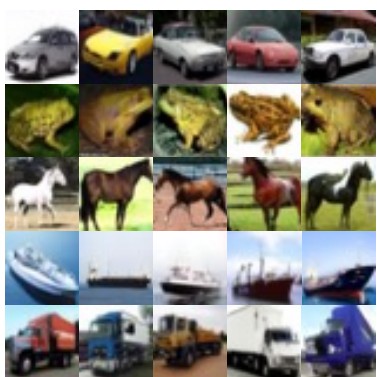

Figure 1: Images generated by GeGM on CIFAR-10

geometric guidance term as $\nabla_{\boldsymbol{x}} f_y^t(\boldsymbol{x})$ to replace the probabilistic guidance $\nabla_{\boldsymbol{x}} \log p_t(y \mid \boldsymbol{x})$ in the reverse process for conditional generation. The resulting nonlinear geometric guidance model (in deterministic form) is defined by

$$\frac{\mathrm{d}}{\mathrm{d}t} \boldsymbol{X}_t^{\leftarrow} = \boldsymbol{X}_t^{\leftarrow} + \nabla_{\boldsymbol{x}} \log p_{T-t}(\boldsymbol{X}_t^{\leftarrow}) - \eta \nabla_{\boldsymbol{x}} f_y^{T-t}(\boldsymbol{X}_t^{\leftarrow}), \tag{16}$$

where $\nabla_{\boldsymbol{x}} \log p_t$ is the score function of the unconditional DDPM initialized from $\mathbb{P}_X$.

To implement the nonlinear geometric guidance model, one must estimate both the score function and the nonlinear geometric guidance term. The score function $\nabla_{\boldsymbol{x}} \log p_t(\boldsymbol{x})$ can be estimated using an unconditional diffusion model—specifically, by training a network $\boldsymbol{s}_\theta(t, \boldsymbol{x})$ via the score matching method (Vincent, 2011) on the unconditional data $\boldsymbol{X}$. The main task, then, is to estimate $\nabla_{\boldsymbol{x}} f_y^t(\boldsymbol{X}_t^{\leftarrow})$.

First, Theorem 10 shows that $\mathcal{M}_y^t = (f_y^t)^{-1}(r(t))$, so such function $f_y^t$ is called a manifold-defining function in Ross et al. (2024). Following a similar idea, we train a network $F_{y,\theta}^t : \mathbb{R}^D \to \mathbb{R}$ to estimate $f_y^t - r(t)$, so $F_{y,\theta}^t$ needs to satisfy

$$F_{y,\theta}^t(\boldsymbol{x}) = 0, \text{ and } \nabla_{\boldsymbol{x}} F_{y,\theta}^t(\boldsymbol{x}) \neq \boldsymbol{0}, \quad \forall \; \boldsymbol{x} \in \mathcal{M}_y^t,$$

where the first condition follows directly from the definition of $\mathcal{M}_y^t$, and the second condition, called the rank condition, ensures $F_{y,\theta}^t$ a manifold-defining function, as guaranteed by the Constant Rank Theorem (Lemma I.2). Therefore, the loss function for training $F_{y,\theta}^t$ is designed as

$$\mathcal{L}_y^t(\theta) := \mathbb{E}_{\boldsymbol{X} \sim p_t(\cdot|y)} \left[ \left| F_{y,\theta}^t(\boldsymbol{X}) \right|^2 - \kappa \left\| \nabla_{\boldsymbol{x}} F_{y,\theta}^t(\boldsymbol{X}) \right\|^2 \right], \tag{17}$$

where $\kappa > 0$ is chosen for controlling the strength of the rank condition. We simply set $\kappa = 1$.

### 6.3 Experiments

**Effectiveness of GeGM.** We use the Fréchet Inception Distance (FID) (Heusel et al., 2017) as the metric for evaluating generation performance, because it can be regarded as a practical surrogate for the Wasserstein distance. We compare the FID of samples generated by the nonlinear geometric guidance model (16) (GeGM) with those generated by the classifier guidance model (CGM) (6). The results are reported in Table1, where we present results for selected classes; the remaining classes are provided in Appendix G.2. Note that the guidance scales used for CGM and GeGM differ, since the norms of the probabilistic and geometric guidance terms are not comparable. For visualization, Figure 1 displays images generated by the nonlinear GeGM. These results demonstrate the effectiveness of the nonlinear GeGM in generating real-world images.

**Performance vs. guidance scale.** By applying the nonlinear GeGM (16), we evaluate how generation performance varies with the guidance scale $\eta$ on selected classes from CIFAR-10; results for the remaining classes are provided in Appendix G.2. As shown in Figure 2, performance improves with increasing $\eta$ within

Table 1: Comparison of FID on CIFAR-10

|                   | Automobile | Frog  | Horse | Ship  | Truck |
|-------------------|------------|-------|-------|-------|-------|
| CGM ($\eta = 1$)  | 13.46      | 17.87 | 13.97 | 11.61 | 16.85 |
| GeGM ($\eta = 50$)| 9.70       | 16.28 | 12.65 | 13.84 | 11.02 |

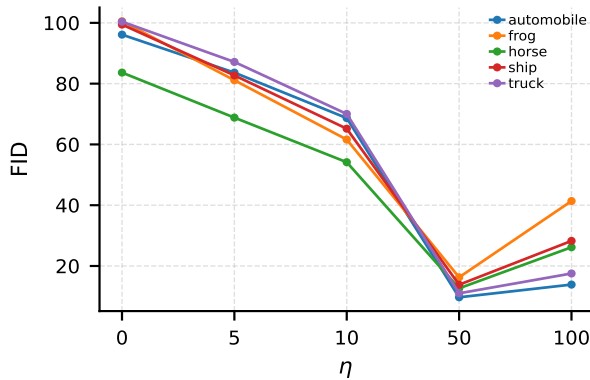

Figure 2: FID v.s. guidance scale $\eta$ of GeGM on selected classes of CIFAR-10

a reasonable range. Since FID serves as a practical approximation of the Wasserstein distance, this trend is consistent with Theorem 11, even in the nonlinear setting.

*Remark* 9. We emphasize that the observed trends are consistent with the spirit of Theorem 11 in nonlinear regimes, but they are not derived from it. Establishing nonlinear analogues of Theorems 10–11 will require additional analysis and is left for future work.

## 7  Conclusion

In this work, we studied the role of the guidance scale in conditional generation with diffusion models. To address the analytical intractability of the probabilistic guidance term, we introduced a geometric guidance model that enables theoretical analysis under the linear manifold hypothesis. To facilitate this analysis, we proposed a mollification technique to ensure the regularity of the score function in the presence of multi-modality. Our results showed that increasing the guidance scale within a reasonable range can enhance generation performance, in line with empirical observations reported in prior studies. We further extended the model to nonlinear settings, and experiments on real-world datasets demonstrated the effectiveness of the geometric guidance model and provided additional evidence consistent with our theoretical findings.

**Limitations:** While the geometric guidance offers a more tractable alternative to probabilistic guidance, it comes with certain limitations. Notably, our analysis showed that the upper bound of the Wasserstein distance between the generated and target conditional distributions is bounded by a constant, regardless of the choice of the guidance scale. This implies that, unlike probabilistic guidance, which can approximate the target conditional distribution by setting the scale to 1, the geometric guidance does not guarantee convergence to the target distribution. This is a trade-off made for the sake of analytical tractability.

Although our experiments on the nonlinear extension partially supported the theoretical results, our current theoretical analysis is restricted to the linear manifold setting. In the nonlinear case, the geometric structure of the score function remains unclear. Regarding regularity of the score function, while Lipschitz continuity can be ensured under compactness assumptions, extending this to the non-compact setting remains an open problem. Furthermore, the log-concavity of the score function cannot be guaranteed, even in compact nonlinear cases.

**Acknowledgments**

We thank Ming Li and Luheng Wang for the helpful discussions. ZZ was supported by Institute for AI and Beyond at the University of Tokyo. MS was supported by JST ASPIRE Grant Number JPMJAP25B1. The authors also thank the anonymous reviewers for their careful reviews and insightful comments, which have been invaluable in improving both the clarity and rigor of this work.

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

## A    Notation

The symbols used throughout this paper are clarified below.

1. **Letters:** Unless otherwise specified, lowercase letters such as $x$ and $\boldsymbol{x}$ denote deterministic variables, while uppercase letters such as $X$ and $\boldsymbol{X}$ denote random variables. Scalars are typically represented by non-bold symbols such as $x$ and $Y$, whereas vectors are denoted using bold symbols such as $\boldsymbol{x}$ and $\boldsymbol{X}$. In particular, we use $\boldsymbol{I}_n \in \mathbb{R}^{n \times n}$ to denote the identity matrix and $\boldsymbol{0} \in \mathbb{R}^n$ to denote the zero vector.

2. **Linear Algebra:**

    (i) Let $\mathcal{O}^{m \times n} \subset \mathbb{R}^{m \times n}$ (with $m > n$) denote the set of matrices whose columns are orthonormal, i.e., those satisfying $A^\top A = \boldsymbol{I}_n$.

    (ii) For a vector $\boldsymbol{x} \in \mathbb{R}^n$, the notation $\|\boldsymbol{x}\|$ refers to the $\ell_2$-norm. For a matrix $A \in \mathbb{R}^{m \times n}$, the operator norm is defined as
    $$\|A\|_{\mathrm{op}} = \sup_{\|\boldsymbol{x}\|=1} \|A\boldsymbol{x}\| = \sqrt{\lambda_{\max}(A^\top A)},$$
    where $\lambda_{\max}(\cdot)$ denotes the maximum eigenvalue.

    (iii) Let $A, B \in \mathbb{R}^{n \times n}$ be symmetric matrices, i.e., $A = A^\top$ and $B = B^\top$. We write $A \preceq B$ (or equivalently, $B \succeq A$) if $B - A$ is positive semi-definite, i.e.,
    $$\boldsymbol{x}^\top (B - A)\boldsymbol{x} \geq 0, \quad \forall\, \boldsymbol{x} \in \mathbb{R}^n.$$

3. **Calculus:**

    (i) For a scalar-valued function $f \colon \mathbb{R}^n \to \mathbb{R}$, the gradient with respect to $\boldsymbol{x}$ is denoted by $\nabla_{\boldsymbol{x}} f(\boldsymbol{x})$, and the Hessian matrix by $\nabla_{\boldsymbol{x}}^2 f(\boldsymbol{x})$.

    (ii) For a vector-valued function $F \colon \mathbb{R}^n \to \mathbb{R}^m$, $JF$ denotes the Jacobian matrix of $F$, and the second-order derivative $D^2 F$ is a bilinear map $D^2 F(\boldsymbol{x}) \colon \mathbb{R}^n \times \mathbb{R}^n \to \mathbb{R}^m$ defined by
    $$D^2 F(\boldsymbol{x})[\boldsymbol{v}, \boldsymbol{w}] = \frac{\partial^2}{\partial s \partial t} F(\boldsymbol{x} + s\boldsymbol{v} + t\boldsymbol{w}) = \left(\boldsymbol{v}^\top \nabla_{\boldsymbol{x}}^2 F^1(\boldsymbol{x})\boldsymbol{w}, \cdots, \boldsymbol{v}^\top \nabla_{\boldsymbol{x}}^2 F^m(\boldsymbol{x})\boldsymbol{w}\right)^\top,$$
    where $F = (F^1, \ldots, F^m)$. If each $F^i$ has continuous derivative of order $k$, $F$ is called $C^k$.

    (iii) For any set $U \subset \mathbb{R}^n$, the characteristic function $\chi_U \colon \mathbb{R}^n \to \mathbb{R}$ is defined by $\chi_U(\boldsymbol{x}) = 1$ if $\boldsymbol{x} \in U$, and $\chi_U(\boldsymbol{x}) = 0$ otherwise.

    (iv) For integrable functions $f, g \colon \mathbb{R}^n \to \mathbb{R}$, their convolution is denoted by
    $$f * g(\boldsymbol{x}) = \int_{\mathbb{R}^n} f(\boldsymbol{y}) g(\boldsymbol{x} - \boldsymbol{y}) \mathrm{d}\boldsymbol{y}.$$

    (v) For a function $f \colon \mathbb{R}^n \to \mathbb{R}^m$, let $\operatorname{Im} f = f(\mathbb{R}^n)$ denote the image of $f$. In particular, for a matrix $A \in \mathbb{R}^{m \times n}$, $\operatorname{Im} A$ refers the image of the linear map $\boldsymbol{x} \mapsto A\boldsymbol{x}$.

4. **Probability-related Symbols:**

   (i) We fix the base probability space $(\Omega, \mathcal{F}, \mathbb{P})$, where $\Omega$ is the sample space, $\mathcal{F}$ is a $\sigma$-algebra, and $\mathbb{P}$ is a probability measure on $\mathcal{F}$.

   (ii) On $\mathbb{R}^n$, we typically work with the Borel $\sigma$-algebra $\mathcal{B}(\mathbb{R}^n)$, and let $\mathcal{P}(\mathbb{R}^n)$ denote the set of all probability measures defined on $\mathcal{B}(\mathbb{R}^n)$. Symbols such as $\mu$ and $\nu$ represent elements of $\mathcal{P}(\mathbb{R}^n)$. The integral with respect to a measure $\mu$ is denoted by $\int f(\boldsymbol{x})\mathrm{d}\mu(\boldsymbol{x})$ or equivalently by $\int f(\boldsymbol{x})\mu(\mathrm{d}\boldsymbol{x})$.

   (iii) For a measurable map $f\colon \Omega \to \mathbb{R}^n$, the push-forward measure of $\mathbb{P}$ under $f$ is denoted by $f_\#\mathbb{P}$, and is defined as
   $$f_\#\mathbb{P}(U) = \mathbb{P}(f^{-1}(U)), \quad \forall\, U \in \mathcal{B}(\mathbb{R}^n).$$

   (iv) A random variable (or vector) $\boldsymbol{X}\colon \Omega \to \mathbb{R}^n$ is a measurable map. Its distribution, denoted by $\mathbb{P}_X$ (or $\mathbb{P}^X$), is a probability measure on $\mathbb{R}^n$ defined by $\mathbb{P}_X = \boldsymbol{X}_\#\mathbb{P}$. For some $\mu \in \mathcal{P}(\mathbb{R}^n)$, we say $\boldsymbol{X} \sim \mu$ if $\mu = \mathbb{P}_X$. Two random variables $\boldsymbol{X}$ and $\boldsymbol{Y}$ are said to be equal in distribution, denoted by $\boldsymbol{X} \overset{\mathrm{d}}{=} \boldsymbol{Y}$, if $\mathbb{P}_X = \mathbb{P}_Y$.

   (v) For $\boldsymbol{X} \sim \mathbb{P}_X$, if $\mathbb{P}_X$ is absolutely continuous with respect to the Lebesgue measure $\mathrm{d}\boldsymbol{x}$, then by the Radon-Nikodym Theorem, there is a function $p_X$ (or denoted by $p^X$) such that
   $$\mathbb{P}_X(U) = \int_U p_X(\boldsymbol{x})\mathrm{d}\boldsymbol{x}, \quad \forall\, U \in \mathcal{B}(\mathbb{R}^n),$$
   and $p_X$ is said the density function [2] of $\boldsymbol{X}$. For a measurable function $g\colon \mathbb{R}^n \to \mathbb{R}^m$, if $\boldsymbol{X} \sim \mathbb{P}_X$, then $g(\boldsymbol{X}) \sim g_\#\mathbb{P}_X$. When $\mathbb{P}_X$ admits a density $p_X$, the density of $g(\boldsymbol{X})$ is denoted by $g_\#p_X$. In particular, if $g(\boldsymbol{x}) = A\boldsymbol{x}$ for a matrix $A \in \mathbb{R}^{m\times n}$, $g_\#\mathbb{P}_X$ is also denoted by $A_\#\mathbb{P}_X$ for simplicity.

   (vi) For random variables $\boldsymbol{X}\colon \Omega \to \mathbb{R}^n$ and $Y\colon \Omega \to \mathbb{R}$, the joint distribution of $(\boldsymbol{X}, Y)\colon \Omega \to \mathbb{R}^n \times \mathbb{R}$ is denoted by $\mathbb{P}_{XY} = (\boldsymbol{X}, Y)_\#\mathbb{P}$, a probability measure on $\mathbb{R}^n \times \mathbb{R}$. The conditional distribution $\mathbb{P}_{X|Y}(\cdot \mid Y)$ is defined as
   $$\mathbb{P}_{X|Y}(U \mid Y) := \mathbb{P}(\boldsymbol{X} \in U \mid Y), \quad \forall\, U \in \mathcal{B}(\mathbb{R}^n),$$
   which is a probability measure on $\mathbb{R}^n$.

   (vii) For a probability measure $\mu \in \mathcal{P}(\mathbb{R}^n)$, the support of $\mu$ is denoted by
   $$\operatorname{supp}\mu = \{\boldsymbol{x} \in \mathbb{R}^n \colon \mu(B_r(\boldsymbol{x})) > 0,\ \forall\, r > 0\}$$
   where $B_r(\boldsymbol{x}) \subset \mathbb{R}^n$ denotes the open ball centered at $\boldsymbol{x}$ with radius $r$. When $\mu$ admits a density function $p$,
   $$\operatorname{supp}\mu = \overline{\{\boldsymbol{x} \in \mathbb{R}^n \colon p(\boldsymbol{x}) > 0\}}$$

# B   More Details in Background

## B.1   Analytic Solution for DDPMs

To solve the SDE
$$\mathrm{d}\boldsymbol{X}_t = -\frac{1}{2}\beta(t)\boldsymbol{X}_t\mathrm{d}t + \sqrt{\beta(t)}\mathrm{d}\boldsymbol{W}_t, \quad \forall\, t \in [0, T],$$

we multiply both sides by the integrating factor $e^{\frac{1}{2}\int_0^t \beta(s)\mathrm{d}s}$. This gives
$$e^{\frac{1}{2}\int_0^t \beta(s)\mathrm{d}s}\mathrm{d}\boldsymbol{X}_t + \frac{1}{2}\beta(t)e^{\frac{1}{2}\int_0^t \beta(s)\mathrm{d}s}\boldsymbol{X}_t\mathrm{d}t = \sqrt{\beta(t)}e^{\frac{1}{2}\int_0^t \beta(s)\mathrm{d}s}\mathrm{d}\boldsymbol{W}_t,$$

which leads to
$$\mathrm{d}\left(e^{\frac{1}{2}\int_0^t \beta(s)\mathrm{d}s}\boldsymbol{X}_t\right) = \sqrt{\beta(t)}e^{\frac{1}{2}\int_0^t \beta(s)\mathrm{d}s}\mathrm{d}\boldsymbol{W}_t,$$

---

[2] When unambiguous, $p_X$ is also occasionally referred to as the distribution.

by applying Itô's formula to $e^{\frac{1}{2}\int_0^t \beta(s)\mathrm{d}s}\boldsymbol{X}_t$. Therefore, we obtain the solution

$$\boldsymbol{X}_t = \sqrt{\alpha_t}\boldsymbol{X}_0 + \boldsymbol{\xi}_t,$$

where $\alpha_t := \exp\left(-\int_0^t \beta(s)\mathrm{d}s\right)$, and

$$\boldsymbol{\xi}_t := \int_0^t e^{-\frac{1}{2}\int_s^t \beta(r)\mathrm{d}r}\sqrt{\beta(s)}\mathrm{d}\boldsymbol{W}_s.$$

Since $(\boldsymbol{W}_t)_{t\geq 0}$ is a standard Brownian motion on $\mathbb{R}^D$, it follows that $\boldsymbol{\xi}_t \sim \mathcal{N}(\boldsymbol{0}, \sigma_{\xi_t}^2 \boldsymbol{I}_D)$. To compute $\sigma_{\xi_t}^2$, let $[\cdot,\cdot]$ denote the quadratic variation. Then

$$\begin{aligned}
\sigma_{\xi_t}^2 &= \mathbb{E}\left[[\boldsymbol{\xi},\boldsymbol{\xi}]_t\right] \\
&= \mathbb{E}\left[\int_0^t e^{-\int_s^t \beta(r)\mathrm{d}r}\beta(s)\mathrm{d}\left[\boldsymbol{W},\boldsymbol{W}\right]_s\right] \\
&= \int_0^t e^{-\int_s^t \beta(r)\mathrm{d}r}\beta(s)\mathrm{d}s = 1 - \exp\left(-\int_0^t \beta(s)\mathrm{d}s\right).
\end{aligned}$$

(see Le Gall (2016) for details). As a result,

$$\boldsymbol{\xi}_t \overset{\mathrm{d}}{=} \sqrt{1-\alpha_t}\boldsymbol{\xi}, \quad \boldsymbol{\xi} \sim \mathcal{N}(\boldsymbol{0}, \boldsymbol{I}_D).$$

## B.2 Density Functions in Conditional DDPMs

**Proposition B.1.** *Consider a joint data density function $p(\boldsymbol{x}, y)$ and the process governed by the SDE:*

$$\mathrm{d}\boldsymbol{X}_t = -\frac{1}{2}\beta(t)\boldsymbol{X}_t\mathrm{d}t + \sqrt{\beta(t)}\mathrm{d}\boldsymbol{W}_t.$$

*For the following two scenarios:*

*(a) Let $\boldsymbol{X} \sim p(\boldsymbol{x} \mid Y = y)$, and run the SDE for $\boldsymbol{X}_0 = \boldsymbol{X}$. Let $p_t^y(\boldsymbol{x}_t)$ be the distribution of $\boldsymbol{X}_t$,*

*(b) Let $(\boldsymbol{X}, Y) \sim p(\boldsymbol{x}, y)$, and run the SDE for $\boldsymbol{X}_0 = \boldsymbol{X}$. Let $p_t(\boldsymbol{x}_t, y)$ be the distribution of $(\boldsymbol{X}_t, Y)$,*

*Then, we have*

$$p_t^y(\boldsymbol{x}_t) = p_t(\boldsymbol{x}_t \mid y).$$

*Proof.* As shown in Equation (2),

$$\boldsymbol{X}_t \overset{\mathrm{d}}{=} \sqrt{\alpha_t}\boldsymbol{X}_0 + \sqrt{1-\alpha_t}\boldsymbol{\xi}, \quad \boldsymbol{\xi} \sim \mathcal{N}(\boldsymbol{0}, \boldsymbol{I}_D),$$

where $\alpha_t = \exp\left(-\int_0^t \beta(s)\mathrm{d}s\right)$. Therefore, in the first case, we have

$$p_t^y(\boldsymbol{x}_t) = (\sqrt{\alpha_t})_\#p(\boldsymbol{x} \mid y) * \mathcal{N}(\boldsymbol{0}, (1-\alpha_t)\boldsymbol{I}_D).$$

Moreover, by Lemma B.2, since $\boldsymbol{\xi}$ is independent of $(\boldsymbol{X}_0, Y)$, it follows that

$$\begin{aligned}
p_t(\boldsymbol{x}_t \mid y) &= p\left(\sqrt{\alpha_t}\boldsymbol{x}_0 + \sqrt{1-\alpha_t}\boldsymbol{\xi} \mid y\right) \\
&= p\left(\sqrt{\alpha_t}\boldsymbol{x} \mid y\right) * \mathcal{N}(\boldsymbol{0}, (1-\alpha_t)\boldsymbol{I}_D) \\
&= (\sqrt{\alpha_t})_\#p(\boldsymbol{x} \mid y) * \mathcal{N}(\boldsymbol{0}, (1-\alpha_t)\boldsymbol{I}_D).
\end{aligned}$$

Consequently, we obtain:

$$p_t^y(\boldsymbol{x}_t) = p_t(\boldsymbol{x}_t \mid y). \qquad \square$$

**Lemma B.2.** *Consider three random variables, $\boldsymbol{X}, \boldsymbol{Y} \in \mathbb{R}^n$, and $Z \in \mathbb{R}$. Let $\boldsymbol{Y}$ be independent of paired $(\boldsymbol{X}, Z)$, and $\boldsymbol{W} = \boldsymbol{X} + \boldsymbol{Y}$. Then, we have*

$$p_{W|Z}(\boldsymbol{w} \mid z) = \big(p_{X|Z}(\cdot \mid z) * p_Y(\cdot)\big)(\boldsymbol{w}).$$

*Or informally,*

$$p_{XY|Z}(\boldsymbol{x} + \boldsymbol{y} \mid z) = p_{X|Z}(\boldsymbol{x} \mid z) * p_Y(\boldsymbol{y}).$$

*Proof.* Because $\boldsymbol{Y}$ is independent of $(\boldsymbol{X}, Z)$,

$$p_{XYZ}(\boldsymbol{x}, \boldsymbol{y}, z) = p_{XZ}(\boldsymbol{x}, z) p_Y(\boldsymbol{y}).$$

Let $D_w = \{(\boldsymbol{x}, \boldsymbol{y}) \colon \boldsymbol{x} + \boldsymbol{y} \le \boldsymbol{w}\}$. Then, we have

$$
\begin{aligned}
\mathbb{P}(\boldsymbol{W} \le \boldsymbol{w}, Z \le z) &= \mathbb{P}(\boldsymbol{X} + \boldsymbol{Y} \le \boldsymbol{w}, Z \le z) \\
&= \int_0^z \left( \iint_{D_w} p_{XYZ}(\boldsymbol{x}, \boldsymbol{y}, z) \mathrm{d}\boldsymbol{x}\mathrm{d}\boldsymbol{y} \right) \mathrm{d}z \\
&= \int_0^z \left( \iint_{D_w} p_{XZ}(\boldsymbol{x}, z) p_Y(\boldsymbol{y}) \mathrm{d}\boldsymbol{x}\mathrm{d}\boldsymbol{y} \right) \mathrm{d}z \\
&= \int_0^z \int_{\boldsymbol{0}}^{\boldsymbol{w}} (p_{XZ}(\cdot, z) * p_Y(\cdot))(\boldsymbol{s}) \mathrm{d}\boldsymbol{s}\mathrm{d}z,
\end{aligned}
$$

where $\boldsymbol{W} = (W_i)_i \le \boldsymbol{w} = (w_i)_i$ means $W_i \le w_i$ for all $i = 1, \dots, n$, and $\int_{\boldsymbol{0}}^{\boldsymbol{w}} \mathrm{d}\boldsymbol{s} = \int_0^{w_n} \cdots \int_0^{w_1} \mathrm{d}s_1 \cdots \mathrm{d}s_n$. It follows that

$$p_{WZ}(\boldsymbol{w}, z) = (p_{XZ}(\cdot, z) * p_Y(\cdot))(\boldsymbol{w}).$$

Therefore,

$$p_{W|Z}(\boldsymbol{w} \mid z) = \frac{p_{WZ}(\boldsymbol{w}, z)}{p_Z(z)} = \left( \frac{p_{XZ}(\cdot, z)}{p_Z(z)} * p_Y(\cdot) \right)(\boldsymbol{w}) = \big(p_{X|Z}(\cdot \mid z) * p_Y(\cdot)\big)(\boldsymbol{w}). \qquad \square$$

*Remark* B.1. In Lemma B.2, the existence of density functions is assumed, which also makes it necessary to assume the existence of the density for $\boldsymbol{X}_0$ in the proof of Proposition B.1. However, this condition is often not satisfied in practice. To address this limitation, consider the convolution of two probability measures $\mu, \nu \in \mathcal{P}(\mathbb{R}^n)$, defined by

$$\mu * \nu(U) := \int_{\mathbb{R}^n} \int_{\mathbb{R}^n} \chi_U(\boldsymbol{x} + \boldsymbol{y}) \mathrm{d}\mu(\boldsymbol{x}) \mathrm{d}\nu(\boldsymbol{y}).$$

Note that $\mu * \nu$ is still a probability measure. Moreover, it follows that if $\boldsymbol{X} \sim \mu$ and $\boldsymbol{Y} \sim \nu$ with $\boldsymbol{X}$ independent of $\boldsymbol{Y}$, then $\boldsymbol{X} + \boldsymbol{Y} \sim \mu * \nu$. Under this formulation, the conclusion of Lemma B.2 remains valid in the general case:

$$\mathbb{P}_{W|Z}(\cdot \mid Z) = \mathbb{P}_{X|Z}(\cdot \mid Z) * \mathbb{P}_{Y|Z}(\cdot \mid Z) = \mathbb{P}_{X|Z}(\cdot \mid Z) * \mathbb{P}_Y(\cdot),$$

where the first equality follows from the fact that independence of $\boldsymbol{Y}$ and $(\boldsymbol{X}, Z)$ implies that $\boldsymbol{Y}$ is independent of $\boldsymbol{X}$ conditional on $Z$, and the second equality holds because $\boldsymbol{Y}$ is independent of $Z$ due to its independence from the pair $(\boldsymbol{X}, Z)$. Therefore, by following a similar line of reasoning as in the proof of Proposition B.1—replacing statements about densities with statements about distributions—we can obtain the same result even when $\boldsymbol{X}_0$ does not admit a density function.

# C    More Details of Geometric Guidance

## C.1    Omitted Poofs in Section 4

*Proof of Proposition 1.* Fix a time $t > 0$. By Equation (2),

$$\boldsymbol{X}_t = \sqrt{\alpha_t} A \boldsymbol{Z} + \sqrt{1 - \alpha_t} \boldsymbol{\xi},$$

for some $\boldsymbol{\xi} \sim \mathcal{N}(\mathbf{0}, \boldsymbol{I}_D)$. It follows that

$$f(\boldsymbol{X}_t) := \left\|(\boldsymbol{I}_D - AA^\top)\boldsymbol{X}_t\right\| = \sqrt{1 - \alpha_t}\left\|(\boldsymbol{I}_D - AA^\top)\boldsymbol{\xi}\right\|.$$

Note that $AA^\top$ is the orthogonal projection to $\operatorname{Im} A$. Therefore, there exists a $U \in \mathcal{O}^{D \times D}$ such that

$$\boldsymbol{I}_D - AA^\top = U^\top \operatorname{diag}(\underbrace{1, \ldots, 1}_{D-d}, 0, \ldots, 0)U.$$

Moreover, the orthogonality of $U$ implies that $\boldsymbol{\nu} = (\nu_1, \ldots, \nu_D)^\top = U\boldsymbol{\xi} \sim \mathcal{N}(0, \boldsymbol{I}_D)$. Hence,

$$f(\boldsymbol{X}_t) = \sqrt{1 - \alpha_t}\left\|(\boldsymbol{I}_D - AA^\top)\boldsymbol{\xi}\right\| = \sqrt{1 - \alpha_t}\left(\nu_1^2 + \cdots + \nu_{D-d}^2\right)^{\frac{1}{2}}.$$

For any $\varepsilon > 0$, by setting $\alpha = (D - d)\varepsilon$ in the Laurent-Massart bound (Lemma H.1), we obtain

$$\mathbb{P}\left(r(t)\sqrt{1 - 2\sqrt{\varepsilon}} \leq f(\boldsymbol{X}_t) \leq r(t)\sqrt{1 + 2\sqrt{\varepsilon} + 2\varepsilon}\right) \geq 1 - 2e^{-2(D-d)\varepsilon}.$$

Since $d \ll D$, we can choose $\varepsilon$ sufficiently small such that $\delta = e^{-2(D-d)\varepsilon}$ is also sufficiently small. As a result, $\mathbb{P}(f(\boldsymbol{X}_t) \approx r(t)) \geq 1 - \delta$, i.e., $\boldsymbol{X}_t$ concentrates on $\mathcal{M}^t = f^{-1}(r(t))$ with high probability. $\qquad\square$

*Proof of Theorem 2.* First, by applying the orthogonal decomposition of the score function in Equation (7), the deterministic reverse process (4) can be rewritten as

$$\frac{\mathrm{d}}{\mathrm{d}t}\boldsymbol{X}_t^\leftarrow = \boldsymbol{X}_t^\leftarrow + A\nabla_{\boldsymbol{z}} \log p_{T-t}^Z(A^\top \boldsymbol{X}_t^\leftarrow) - \frac{1}{1 - \alpha_{T-t}}(\boldsymbol{I}_D - AA^\top)\boldsymbol{X}_t^\leftarrow. \qquad (18)$$

(a) Because $A \in \mathcal{O}^{D \times d}$, we have $A^\top A = \boldsymbol{I}_d$ and $A^\top(\boldsymbol{I}_D - AA^\top) = \boldsymbol{O}$. Therefore, by multiplying $A^\top$ on the both sides of (18),

$$\frac{\mathrm{d}}{\mathrm{d}t}\boldsymbol{Z}_t^\leftarrow = \boldsymbol{Z}_t^\leftarrow + \nabla_{\boldsymbol{z}} \log p_{T-t}^Z(\boldsymbol{Z}_t^\leftarrow),$$

for $\boldsymbol{Z}_t^\leftarrow = A^\top \boldsymbol{X}_t$. Moreover, by the equivalence of the continuity equation of the Fokker-Planck equation (or by the statements in Appendix C.2), $\boldsymbol{Z}_t = \boldsymbol{Z}_{T-t}^\leftarrow$ satisfies the forward process of DDPMs starting from $p^Z$.

(b) Similarly, by multiplying $\boldsymbol{I}_D - AA^\top$ on the both sides of (18),

$$\frac{\mathrm{d}}{\mathrm{d}t}\boldsymbol{X}_{t,\perp}^\leftarrow = \boldsymbol{X}_{t,\perp}^\leftarrow - \frac{1}{1 - \alpha_{T-t}}\boldsymbol{X}_{t,\perp}^\leftarrow = -\frac{\alpha_{T-t}}{1 - \alpha_{T-t}}\boldsymbol{X}_{t,\perp}^\leftarrow,$$

for $\boldsymbol{X}_{t,\perp}^\leftarrow = (\boldsymbol{I}_D - AA^\top)\boldsymbol{X}_t$. Note that $\alpha_{T-t} = e^{-2(T-t)}$. Therefore, this equation has the analytical solution given by

$$\boldsymbol{X}_{t_0+\delta,\perp}^\leftarrow = \sqrt{\frac{1 - e^{-2(T-(t_0+\delta))}}{1 - e^{-2(T-t_0)}}}\boldsymbol{X}_{t_0,\perp}^\leftarrow.$$

When $\|\boldsymbol{X}_{t_0,\perp}^\leftarrow\| = \sqrt{(D-d)\left(1 - e^{-2(T-t_0)}\right)}$, it follows that

$$\left\|\boldsymbol{X}_{t_0+\delta,\perp}^\leftarrow\right\| = \sqrt{\frac{1 - e^{-2(T-(t_0+\delta))}}{1 - e^{-2(T-t_0)}}}\left\|\boldsymbol{X}_{t_0,\perp}^\leftarrow\right\| = \sqrt{(D-d)\left(1 - e^{-2(T-(t_0+\delta))}\right)}. \qquad\square$$

## C.2 Decomposition of Score Function

By Equation (2) and the assumption $\boldsymbol{X}_0 = A\boldsymbol{Z}$, we have

$$
\begin{aligned}
\boldsymbol{X}_t &= \sqrt{\alpha_t}\boldsymbol{X}_0 + \sqrt{1-\alpha_t}\boldsymbol{\xi} \\
&= \underbrace{\sqrt{\alpha_t}\boldsymbol{X}_0 + \sqrt{1-\alpha_t}Q\boldsymbol{\xi}}_{=:\boldsymbol{X}_{t,\|}} + \underbrace{\sqrt{1-\alpha_t}(\boldsymbol{I}_D - Q)\boldsymbol{\xi}}_{=:\boldsymbol{X}_{t,\perp}}
\end{aligned}
$$

for some $\boldsymbol{\xi} \sim \mathcal{N}(\boldsymbol{0}, \boldsymbol{I}_D)$, where $Q = AA^\top$ is the orthogonal projection onto $\operatorname{Im} A$.

We compute the covariance:

$$
\begin{aligned}
\operatorname{Cov}(Q\boldsymbol{\xi}, (\boldsymbol{I}_D - Q)\boldsymbol{\xi}) &= \mathbb{E}\left[Q\boldsymbol{\xi} \cdot ((\boldsymbol{I}_D - Q)\boldsymbol{\xi})^\top\right] - \mathbb{E}\left[Q\boldsymbol{\xi}\right] \cdot \mathbb{E}\left[(\boldsymbol{I}_D - Q)\boldsymbol{\xi}\right]^\top \\
&= \mathbb{E}\left[Q\boldsymbol{\xi} \cdot ((\boldsymbol{I}_D - Q)\boldsymbol{\xi})^\top\right] = Q\mathbb{E}\left[\boldsymbol{\xi}\boldsymbol{\xi}^\top\right](\boldsymbol{I}_D - Q) \\
&= Q(\boldsymbol{I}_D - Q) = 0,
\end{aligned}
$$

which shows that $Q\boldsymbol{\xi}$ and $(\boldsymbol{I}_D - Q)\boldsymbol{\xi}$ are uncorrelated. Since both are Gaussian, they are independent. Hence, $\boldsymbol{X}_{t,\perp}$ is independent of $\sqrt{1-\alpha_t}Q\boldsymbol{\xi}$. Combined with the fact that $\boldsymbol{\xi}$ is independent of $\boldsymbol{X}_0$, it follows that $\boldsymbol{X}_{t,\|}$ is independent of $\boldsymbol{X}_{t,\perp}$. By Lemma H.2, the density of $\boldsymbol{X}_t$ admits the decomposition

$$
p_t(\boldsymbol{x}) = p_{t,\|}(\boldsymbol{x}_\|)p_{t,\perp}(\boldsymbol{x}_\perp), \tag{19}
$$

where $p_{t,\|}$ and $p_{t,\perp}$ are the densities of $\boldsymbol{X}_{t,\|}$ and $\boldsymbol{X}_{t,\perp}$ with respect to the canonical volume measures on $\operatorname{Im} A$ and $(\operatorname{Im} A)^\perp$, respectively. Here, $\boldsymbol{x}_\| = Q\boldsymbol{x}$ and $\boldsymbol{x}_\perp = \boldsymbol{x} - \boldsymbol{x}_\|$.

Next, let us analyze $p_{t,\|}$ and $p_{t,\perp}$, respectively.

(i) For the parallel part, first define $\boldsymbol{Z}_t := A^\top \boldsymbol{X}_t$. Then, by multiplying $A^\top$ on the both sides of Equation (1), we obtain

$$
\mathrm{d}\boldsymbol{Z}_t = -\boldsymbol{Z}_t\mathrm{d}t + \sqrt{2}\mathrm{d}\boldsymbol{B}_t,
$$

where $(\boldsymbol{B}_t)_{t\geq 0} = (A^\top \boldsymbol{W}_t)_{t\geq 0}$ is a standard Brownian motion on $\mathbb{R}^d$ by Lemma H.3. Therefore, the process $\boldsymbol{Z}_t \sim p_t^Z$ is governed by the DDPM dynamics initialized from $p^Z$. Since

$$
\boldsymbol{X}_{t,\|} = Q\boldsymbol{X}_t = A\boldsymbol{Z}_t,
$$

this shows that $\boldsymbol{X}_{t,\|}$ evolves as a diffusion process on the target data manifold $\mathcal{M} = \operatorname{Im} A$.

Moreover, applying Lemma H.4 gives

$$
p_{t,\|}(\boldsymbol{x}_\|) = A_\# p_t^Z(\boldsymbol{x}_\|) = p_t^Z(A^\top \boldsymbol{x}_\|) = p_t^Z(A^\top \boldsymbol{x}). \tag{20}
$$

(ii) For the orthogonal part, we have

$$
\boldsymbol{X}_{t,\perp} = \sqrt{1-\alpha_t}P\boldsymbol{\xi} \sim \mathcal{N}(\boldsymbol{0}, (1-\alpha_t)P),
$$

where $P = \boldsymbol{I}_D - Q$ is an orthogonal projection with rank $D-d$. So $P = B^\top B$ for some $B \in \mathcal{O}^{D\times(D-d)}$. It follows that $\boldsymbol{X}_{t,\perp}$ is a Gaussian on $\operatorname{Im} B$, i.e., $\boldsymbol{X}_{t,\perp} = B\boldsymbol{W}$ for some $\boldsymbol{W} \sim \mathcal{N}(\boldsymbol{0}, (1-\alpha_t)\boldsymbol{I}_{D-d})$. Therefore, $\boldsymbol{X}_{t,\perp}$ is basically a $(D-d)$-dimensional Gaussian. When $d \ll D$, as shown in the proof in Proposition 1,

$$
\left\|(\boldsymbol{I}_D - AA^\top)\boldsymbol{X}_t\right\| = \|\boldsymbol{X}_{t,\perp}\| \approx r(t),
$$

which implies that the orthogonal part $\boldsymbol{X}_{t,\perp}$ is responsible for the concentration of $\boldsymbol{X}_t$ on $\mathcal{M}^t$ and endows $\boldsymbol{X}_t$ with its geometric structure. Furthermore, by Lemma H.5, $p_t^\perp$ is approximately uniform on the sphere $\mathbb{S}^{(D-d)-1}(r(t))$. In other words, the density $p_t$, which is concentrated on the cylindrical-like surface $\mathcal{M}^t$, remains constant along radial directions and varies only in the longitudinal direction governed by $p_{t,\|}$—a consequence of diffusion along the subspace $\operatorname{Im} A$.

Moreover, applying Lemma H.4 again, we obtain

$$p_{t,\perp}(\boldsymbol{x}_\perp) = B_\# p^W(\boldsymbol{x}_\perp) = p^W(B^\top \boldsymbol{x}_\perp) = p^W(B^\top \boldsymbol{x}), \tag{21}$$

where

$$p^W(\boldsymbol{w}) = (2\pi(1-\alpha_t))^{-\frac{D-d}{2}} \exp\left(-\frac{\|\boldsymbol{w}\|^2}{2(1-\alpha_t)}\right).$$

Finally, for the decomposition, by combining (20) and (21) with (19), we get

$$\log p_t(\boldsymbol{x}) = \log p_t^Z(A^\top \boldsymbol{x}) + \log p^W(B^\top \boldsymbol{x}),$$

from which the orthogonal decomposition formula immediately follows:

$$\nabla_{\boldsymbol{x}} \log p_t(\boldsymbol{x}) = A \, \nabla_{\boldsymbol{z}} \log p_t^Z(\boldsymbol{z})\big|_{\boldsymbol{z}=A^\top \boldsymbol{x}} - \frac{1}{1-\alpha_t}(\boldsymbol{I}_D - P)\boldsymbol{x},$$

as originally derived via direct computation by Chen et al. (2023b).

For the geometric property, the randomness of $\boldsymbol{X}_t$ arises from the diffusion process on the target data manifold $\mathcal{M} = \operatorname{Im} A$, while the geometric structure of $\boldsymbol{X}_t$ results from the concentration behavior of the orthogonal part.

### C.3  Construction of Geometric Guidance

To clarify our intuition about $\nabla_{\boldsymbol{x}} \log p_t(y = 1 \mid \boldsymbol{x})$ "almost normal" to $\mathcal{M}_1^t$, we will show that there exists a small $\beta_t > 0$ such that

$$\|\nabla_{\boldsymbol{x}} \log p_t(y = 1 \mid \boldsymbol{x}) + \eta_t P_1 \boldsymbol{x}\| \leq \beta_t, \quad \forall \, \boldsymbol{x} \in \mathcal{M}_1^t,$$

for some scalar $\eta_t > 0$. But first, we need the following lemma.

**Lemma C.1.** *Let $\mathcal{M} \subset \mathbb{R}^D$ be a smooth manifold with dimension $D-1$. Let $V \subset \mathbb{R}^D$ be a tubular neighborhood of $\mathcal{M}$ with the orthogonal projection $\pi\colon V \to \mathcal{M}$. Let $f\colon V \to \mathbb{R}$ be a $C^2$-function satisfying the following two conditions.*

*(a)* $\|\nabla_{\boldsymbol{x}}^2 f(\boldsymbol{x})\|_{\mathrm{op}} \leq L.$

*(b)* $f|_{\mathcal{M}}$ *is $\beta$-Lipschitz with the induced distance of $\mathbb{R}^n$ on $\mathcal{M}$.*

*Then for any $\boldsymbol{x} \in V$,*

$$\|\nabla_{\boldsymbol{x}} f(\boldsymbol{x}) - \partial_n f(\pi(\boldsymbol{x}))n(\pi(\boldsymbol{x}))\| \leq \beta + L \operatorname{dist}(\boldsymbol{x}, \mathcal{M}),$$

*where $n\colon \mathcal{M} \to \mathbb{R}^n$ is a continuous unit normal vector field along $\mathcal{M}$, $\partial_n f = \langle \nabla f, n \rangle$ the derivative along $n$, and $\operatorname{dist}(\boldsymbol{x}, \mathcal{M}) = \inf\{\|\boldsymbol{x} - \boldsymbol{y}\| \colon \boldsymbol{y} \in \mathcal{M}\}$ is the distance from $\boldsymbol{x}$ to $\mathcal{M}$.*

*Proof.* Let $\mathcal{M}$ be equipped with the induced Riemannian structure of $\mathbb{R}^n$ and $\nabla^M$ be the corresponding Levi-Civita connection. Because $\mathcal{M} \subset \mathbb{R}^D$ is a hypersurface, i.e., submanifold with dimension $D-1$,

$$\nabla f = \nabla^M f + (\partial_n f)n, \tag{22}$$

see the details in Lee (2019, Chapter 8). Fix $\boldsymbol{x} \in V$ with $\boldsymbol{y} = \pi(\boldsymbol{x}) \in \mathcal{M}$. Note that

$$\operatorname{dist}(\boldsymbol{x}, \mathcal{M}) = \|\boldsymbol{x} - \boldsymbol{y}\|, \tag{23}$$

by Lee (2019, Proposition 5.26 (c)). Writing

$$\|\nabla_{\boldsymbol{x}} f(\boldsymbol{x}) - \partial_n f(\boldsymbol{y})n(\boldsymbol{y})\| \leq \|\nabla_{\boldsymbol{x}} f(\boldsymbol{x}) - \nabla_{\boldsymbol{x}} f(\boldsymbol{y})\| + \|\nabla_{\boldsymbol{x}} f(\boldsymbol{y}) - \partial_n f(\boldsymbol{y})n(\boldsymbol{y})\|. \tag{24}$$

I. For the first term, by

$$\nabla_{\boldsymbol{x}}f(\boldsymbol{x}) - \nabla_{\boldsymbol{x}}f(\boldsymbol{y}) = \int_0^1 \nabla_{\boldsymbol{x}}^2 f(\boldsymbol{y} + s(\boldsymbol{x} - \boldsymbol{y}))(\boldsymbol{x} - \boldsymbol{y})\mathrm{d}s,$$

the fact that $\|\nabla_{\boldsymbol{x}}^2 f(\boldsymbol{x})\|_{\mathrm{op}} \leq L$, and Equation (23), we have

$$\|\nabla_{\boldsymbol{x}}f(\boldsymbol{x}) - \nabla_{\boldsymbol{x}}f(\boldsymbol{y})\| \leq L\|\boldsymbol{x} - \boldsymbol{y}\| = L\operatorname{dist}(\boldsymbol{x}, \mathcal{M}). \tag{25}$$

II. For the second term, first, by (22),

$$\|\nabla_{\boldsymbol{x}}f(\boldsymbol{y}) - \partial_n f(\boldsymbol{y})n(\boldsymbol{y})\| = \|\nabla^M f(\boldsymbol{y})\|.$$

By assumption, $f|_{\mathcal{M}}$ is $\beta$-Lipschitz with the induced distance of $\mathbb{R}^n$ on $\mathcal{M}$, i.e.,

$$|f(\boldsymbol{y}_1) - f(\boldsymbol{y}_2)| \leq \beta d_{\mathcal{M}}(\boldsymbol{y}_1, \boldsymbol{y}_2),$$

where $d_{\mathcal{M}}$. It implies that

$$\|\nabla^M f(\boldsymbol{z})\| \leq \beta, \quad \forall\, \boldsymbol{z} \in \mathcal{M}, \tag{26}$$

see the details in Boumal (2023, Proposition 10.43).

Then combining the inequalities (25) and (26) with (24),

$$\|\nabla_{\boldsymbol{x}}f(\boldsymbol{x}) - \partial_n f(\boldsymbol{y})n(\boldsymbol{y})\| \leq \beta + L\operatorname{dist}(\boldsymbol{x}, \mathcal{M}). \qquad \square$$

Let $f_t(\boldsymbol{x}) = \log p_t(y = 1 \mid \boldsymbol{x})$. It is natural to assume that $f_t$ is $C^2$ on a tubular neighborhood $V$ of $\mathcal{M}_1^t$, that $\|\nabla^2 f_t\|_{\mathrm{op}} \leq L_t$ on $V$, and that $f_t$ is $\beta_t$-Lipschitz continuous on $\mathcal{M}_1^t$. Then by Lemma C.1,

$$\|\nabla_{\boldsymbol{x}}f_t(\boldsymbol{x}) - \partial_n f_t(\pi(\boldsymbol{x}))n_t(\pi(\boldsymbol{x}))\| \leq \beta_t + L_t \operatorname{dist}(\boldsymbol{x}, \mathcal{M}_1^t).$$

In particular, for any $\boldsymbol{x} \in \mathcal{M}_1^t$ and $\pi(\boldsymbol{x}) = \boldsymbol{x}$, we have

$$\|\nabla_{\boldsymbol{x}}f_t(\boldsymbol{x}) - \partial_n f_t(\boldsymbol{x})n_t(\boldsymbol{x})\| \leq \beta_t.$$

Two questions remain: whether $\partial_n f_t(\boldsymbol{x})n_t(\boldsymbol{x}) = -\eta_t P_1\boldsymbol{x}$ for some scalar $\eta_t > 0$, and how to bound $\beta_t$.

For the first question, we can choose $n_t(\boldsymbol{x}) = P_1\boldsymbol{x}/\|P_1\boldsymbol{x}\|$ by the definition (9) of $\mathcal{M}_1^t$ and Lemma I.1. So

$$\partial_n f_t(\boldsymbol{x})n_t(\boldsymbol{x}) = -\eta_t P_1\boldsymbol{x},$$

for

$$\eta_t = -\frac{\partial_n f_t(\boldsymbol{x})}{\|P_x\boldsymbol{x}\|}.$$

Moreover, because $p_t(y = 1 \mid \boldsymbol{x})$ is the classifier for $(\boldsymbol{X}_t, y = 1)$ and such $\boldsymbol{X}_t$ concentrates on $\mathcal{M}_1^t$ by Proposition 1, $f_t(\boldsymbol{x}) = \log p_t(y = 1 \mid \boldsymbol{x})$ decreases when $\boldsymbol{x}$ moves away from $M_1^t$. So

$$\partial_n f_t(\boldsymbol{x}) < 0 \quad \Rightarrow \quad \eta_t > 0.$$

Next, to bound $\beta_t$, we introduce the following lemma.

**Lemma C.2.** *Let $p(y = k \mid \boldsymbol{x})$ be a softmax classifier with logits $g_k(\boldsymbol{x})$ for $k = 1, 2, \cdots, K$, that is,*

$$p(y = k \mid \boldsymbol{x}) = \frac{\exp(g_k(\boldsymbol{x}))}{\sum_{k=1}^K \exp(g_k(\boldsymbol{x}))}.$$

*Assume $\|\nabla_{\boldsymbol{x}}g_k(\boldsymbol{x})\| \leq L$ for all $k, \boldsymbol{x}$. Let $\mathcal{M}_k$ be the region where points with label $y = k$ concentrate on. Assume classifier confidence*

$$p(y = k \mid \boldsymbol{x}) > 1 - \varepsilon, \quad \forall\, \boldsymbol{x} \in \mathcal{M}_k.$$

*Then*

$$\|\nabla_{\boldsymbol{x}}\log p(y = k \mid \boldsymbol{x})\| \leq 2L\varepsilon, \quad \forall\, \boldsymbol{x} \in \mathcal{M}_k.$$

*Proof.* Fix $k$. Let $f(\boldsymbol{x}) = \log p(y = k \mid \boldsymbol{x})$.

$$\nabla_{\boldsymbol{x}} f(\boldsymbol{x}) = \nabla_{\boldsymbol{x}} g_k(\boldsymbol{x}) - \sum_{j=1}^{K} p(y = j \mid \boldsymbol{x}) \nabla_{\boldsymbol{x}} g_j(\boldsymbol{x})$$

$$= \sum_{j=1}^{K} p(y = j \mid \boldsymbol{x}) \left( \nabla_{\boldsymbol{x}} g_j(\boldsymbol{x}) - \nabla_{\boldsymbol{x}} g_k(\boldsymbol{x}) \right)$$

$$= \sum_{j \neq k} p(y = j \mid \boldsymbol{x}) \left( \nabla_{\boldsymbol{x}} g_j(\boldsymbol{x}) - \nabla_{\boldsymbol{x}} g_k(\boldsymbol{x}) \right)$$

By assumption,

$$\|\nabla_{\boldsymbol{x}} g_j(\boldsymbol{x}) - \nabla_{\boldsymbol{x}} g_k(\boldsymbol{x})\| \leq \|\nabla_{\boldsymbol{x}} g_j(\boldsymbol{x})\| + \|\nabla_{\boldsymbol{x}} g_k(\boldsymbol{x})\| \leq 2L.$$

Therefore,

$$\|\nabla_{\boldsymbol{x}} f(\boldsymbol{x})\| \leq \sum_{j \neq k} p(y = j \mid \boldsymbol{x}) \|\nabla_{\boldsymbol{x}} g_j(\boldsymbol{x}) - \nabla_{\boldsymbol{x}} g_k(\boldsymbol{x})\|$$

$$\leq 2L \sum_{j \neq k} p(y = j \mid \boldsymbol{x}) = 2L(1 - p(y = k \mid \boldsymbol{x})).$$

It implies that

$$\|\nabla_{\boldsymbol{x}} f(\boldsymbol{x})\| \leq 2L\varepsilon, \quad \forall\, \boldsymbol{x} \in \mathcal{M}_k,$$

by the assumption that classifier confidence $> 1 - \varepsilon$ on $\mathcal{M}_k$. $\square$

Therefore, for all $p_t(y = 1 \mid \boldsymbol{x})$, we assume that they satisfy the conditions in Lemma C.2. Then if

$$p_t(y = 1 \mid \boldsymbol{x}) > 1 - \varepsilon_t, \quad \forall\, \boldsymbol{x} \in \mathcal{M}_1^t,$$

for a small $\varepsilon_t$, then

$$\|\nabla^M f_t(\boldsymbol{x})\| \leq \|\nabla_{\boldsymbol{x}} f_t(\boldsymbol{x})\| = \sqrt{\|\nabla^M f_t(\boldsymbol{x})\|^2 + |\partial_n f_t(\boldsymbol{x})|^2} \leq 2C\varepsilon_t, \quad \forall\, \boldsymbol{x} \in \mathcal{M}_1^t.$$

So $\beta_t \leq 2C\varepsilon_t$.

Combining above results, we have

$$\|\nabla_{\boldsymbol{x}} \log p_t(y = 1 \mid \boldsymbol{x}) + \eta_t P_1 \boldsymbol{x}\| \leq \beta_t, \quad \forall\, \boldsymbol{x} \in \mathcal{M}_1^t,$$

for some $\eta_t > 0$. Moreover, $\beta_t = \mathcal{O}(\varepsilon_t)$ for $p_t(y = 1 \mid \boldsymbol{x}) > 1 - \varepsilon_t$ on $\mathcal{M}_1^t$.

## D  More Details related to Main Results

### D.1  Omitted Proofs in Section 5.1

*Proof of Proposition 3.* First, the Hessian is

$$\nabla_{\boldsymbol{x}}^2 \log p_X(\boldsymbol{x}) = \frac{\nabla_{\boldsymbol{x}}^2 p_X(\boldsymbol{x})}{p_X(\boldsymbol{x})} - \frac{\nabla_{\boldsymbol{x}} p_X(\boldsymbol{x}) \nabla_{\boldsymbol{x}} p_X(\boldsymbol{x})^\top}{p_X(\boldsymbol{x})^2}.$$

To express the above formula explicitly, by the definition, for any $\boldsymbol{x} \in \mathbb{R}^n$,

$$p_X(\boldsymbol{x}) = \int_{\mathbb{R}^k} K_z(\boldsymbol{x}) p^Z(\boldsymbol{z}) \mathrm{d}\boldsymbol{z}, \quad K_z(\boldsymbol{x}) := (2\pi\beta^2)^{-\frac{n}{2}} \exp\left( -\frac{\|\boldsymbol{x} - \alpha B \boldsymbol{z}\|^2}{2\beta^2} \right),$$

and so

$$\nabla_{\boldsymbol{x}} K_z(\boldsymbol{x}) = \frac{\alpha B \boldsymbol{z} - \boldsymbol{x}}{\beta^2} K_z(\boldsymbol{x}),$$

$$\nabla_{\boldsymbol{x}}^2 K_z(\boldsymbol{x}) = -\frac{1}{\beta^2} K_z(\boldsymbol{x}) \boldsymbol{I}_n + \frac{(\boldsymbol{x} - \alpha B \boldsymbol{z})(\boldsymbol{x} - \alpha B \boldsymbol{z})^\top}{\beta^4} K_z(\boldsymbol{x}).$$

Let

$$\mathrm{d}\mu_x(\boldsymbol{z}) = \frac{K_z(\boldsymbol{x}) p^Z(\boldsymbol{z})}{p^X(\boldsymbol{x})} \mathrm{d}\boldsymbol{z}$$

be the posterior probability measure on $\mathbb{R}^k$. Then, for the first term

$$\frac{\nabla_{\boldsymbol{x}}^2 p_X(\boldsymbol{x})}{p_X(\boldsymbol{x})} = \frac{\int_{\mathbb{R}^k} \nabla_{\boldsymbol{x}}^2 K_z(\boldsymbol{x}) p^Z(\boldsymbol{z}) \mathrm{d}\boldsymbol{z}}{p_X(\boldsymbol{x})} = -\frac{1}{\beta^2} \boldsymbol{I}_n + \frac{1}{\beta^4} \mathbb{E}_{\boldsymbol{Z} \sim \mu_x} \left[ (\boldsymbol{x} - \alpha B \boldsymbol{Z})(\boldsymbol{x} - \alpha B \boldsymbol{Z})^\top \right],$$

and for the second term

$$\frac{\nabla_{\boldsymbol{x}} p_X(\boldsymbol{x}) \nabla_{\boldsymbol{x}} p_X(\boldsymbol{x})^\top}{p_X(\boldsymbol{x})^2} = \frac{1}{\beta^4} \mathbb{E}_{\boldsymbol{Z} \sim \mu_x} \left[ \boldsymbol{x} - \alpha B \boldsymbol{Z} \right] \mathbb{E}_{\boldsymbol{Z} \sim \mu_x} \left[ \boldsymbol{x} - \alpha B \boldsymbol{Z} \right]^\top.$$

Moreover, note that

$$\mathbb{E}_{\boldsymbol{Z} \sim \mu_x} \left[ (\boldsymbol{x} - \alpha B \boldsymbol{Z})(\boldsymbol{x} - \alpha B \boldsymbol{Z})^\top \right] - \mathbb{E}_{\boldsymbol{Z} \sim \mu_x} \left[ \boldsymbol{x} - \alpha B \boldsymbol{Z} \right] \mathbb{E}_{\boldsymbol{Z} \sim \mu_x} \left[ \boldsymbol{x} - \alpha B \boldsymbol{Z} \right]^\top$$
$$= \mathrm{Cov}_{\boldsymbol{Z} \sim \mu_x} (\boldsymbol{x} - \alpha B \boldsymbol{Z}) = \alpha^2 \, \mathrm{Cov}_{\mu_x}(B \boldsymbol{Z}) = \alpha^2 B \, \mathrm{Cov}_{\mu_x}(\boldsymbol{Z}) B^\top.$$

Therefore, we get

$$\nabla_{\boldsymbol{x}}^2 \log p_X(\boldsymbol{x}) = \frac{\alpha^2}{\beta^4} B \, \mathrm{Cov}_{\mu_x}(\boldsymbol{Z}) B^\top - \frac{1}{\beta^2} \boldsymbol{I}_n. \tag{27}$$

It follows that

$$\left\| \nabla_{\boldsymbol{x}}^2 \log p_X(\boldsymbol{x}) \right\|_{\mathrm{op}} \le \frac{1}{\beta^2} + \frac{\alpha^2 \Lambda}{\beta^4} \left\| \mathrm{Cov}_{\mu_x}(\boldsymbol{Z}) \right\|_{\mathrm{op}}. \tag{28}$$

It is sufficient to bound $\left\| \mathrm{Cov}_{\mu_x}(\boldsymbol{Z}) \right\|_{\mathrm{op}}$. To do that, $\mu_x$ is required to satisfy the Poincaré Inequality. Let $p^Z(\boldsymbol{z}) = \exp(-V(\boldsymbol{z}))$ for some $V \colon \mathbb{R}^k \to \mathbb{R}$ and

$$U_x(\boldsymbol{z}) := \frac{\|\boldsymbol{x} - \alpha B \boldsymbol{z}\|^2}{2\beta^2} + V(\boldsymbol{z}),$$

which indicates that $\mathrm{d}\mu_x(\boldsymbol{z}) = e^{-U_x(\boldsymbol{z})} \mathrm{d}\boldsymbol{z} / \int e^{-U_x}$. Because $\nabla_{\boldsymbol{z}}^2 V(\boldsymbol{z}) = -\nabla_{\boldsymbol{z}}^2 \log p^Z(\boldsymbol{z}) \succeq m_0 \boldsymbol{I}_k$,

$$\nabla_{\boldsymbol{z}}^2 U_x(\boldsymbol{z}) = \frac{\alpha^2}{\beta^2} B^\top B + \nabla_{\boldsymbol{z}}^2 V(\boldsymbol{z}) \succeq \left( \frac{\alpha^2 \lambda}{\beta^2} + m_0 \right) \boldsymbol{I}_k.$$

Then, by Lemma H.6, $\mu_x$ satisfies the Poincaré Inequality with constant $m := \alpha^2 \lambda / \beta^2 + m_0$. Thus, for any $C^1$ function $f \colon \mathbb{R}^k \to \mathbb{R}$,

$$\mathrm{Var}_{\mu_x}(f) \le \frac{1}{m} \mathbb{E}_{\mu_x} \left[ \|\nabla f\|^2 \right].$$

For any $\boldsymbol{u} \in \mathbb{R}^n$, let $f_u(\boldsymbol{z}) = \langle \boldsymbol{u}, \boldsymbol{z} \rangle$ with $\nabla_{\boldsymbol{z}} f_u(\boldsymbol{z}) = \boldsymbol{u}$. The above inequality implies that

$$\boldsymbol{u}^\top \mathrm{Cov}_{\mu_x}(\boldsymbol{Z}) \boldsymbol{u} = \mathrm{Var}_{\mu_x}(f_u) \le \frac{1}{m} \mathbb{E}_{\mu_x} \left[ \|\nabla_{\boldsymbol{z}} f_u\|^2 \right] \le \frac{1}{m} \|\boldsymbol{u}\|^2,$$

for any $\boldsymbol{u} \in \mathbb{R}^k$. Therefore,

$$\left\| \mathrm{Cov}_{\mu_x}(\boldsymbol{Z}) \right\|_{\mathrm{op}} \le \frac{1}{m}. \tag{29}$$

Finally, by plugging inequality (29) into Equation (28), we get the result

$$\left\| \nabla_{\boldsymbol{x}}^2 \log p_X(\boldsymbol{x}) \right\|_{\mathrm{op}} \le \frac{1}{\beta^2} + \frac{\alpha^2 \Lambda}{\beta^2(\alpha^2 \lambda + m_0 \beta^2)}. \qquad \square$$

*Proof of Corollary 4.* By Equation (29),

$$\left\| B \operatorname{Cov}_{\mu_x}(\boldsymbol{Z}) B^\top \right\|_{\mathrm{op}} \le \frac{\Lambda}{m} \;\Rightarrow\; B \operatorname{Cov}_{\mu_x}(\boldsymbol{Z}) B^\top \preceq \frac{\Lambda}{m} \boldsymbol{I}_n.$$

By combining this with Equation (27), we have

$$\nabla_{\boldsymbol{x}}^2 \log p_X(\boldsymbol{x}) \preceq \left( \frac{\alpha^2 \Lambda}{\beta^2(\alpha^2\lambda + m_0\beta^2)} - \frac{1}{\beta^2} \right) \boldsymbol{I}_n. \qquad \square$$

*Proof of Proposition 5.* By Lemma H.7,

$$\mathbb{P}_X^\sigma = A_\# \mathbb{P}_\sigma^Z = w_1 A_\# \mathbb{P}_{1,\sigma}^Z + w_2 A_\# \mathbb{P}_{2,\sigma}^Z.$$

Moreover, because $\boldsymbol{Z}_{1,\sigma} = (\boldsymbol{Z}_1, 0)^\top + \sigma\boldsymbol{\zeta} \sim \mathbb{P}_{i,\sigma}^Z$ with $\boldsymbol{\zeta} \sim \mathcal{N}(\boldsymbol{0}, \boldsymbol{I}_d)$,

$$A\boldsymbol{Z}_{1,\sigma} = A_1 \boldsymbol{Z}_1 + \sigma A\boldsymbol{\zeta} \sim \mathbb{P}_{X|Y}^\sigma(\cdot \mid Y = 1).$$

Note that $A_1 \boldsymbol{Z}_1 \sim \mathbb{P}_{X|Y}(\cdot \mid Y = 1)$. Therefore,

$$\mathcal{W}_1(\mathbb{P}_{X|Y}^\sigma(\cdot \mid Y = 1), \mathbb{P}_{X|Y}(\cdot \mid Y = 1)) \le \mathbb{E}\left[\|A\boldsymbol{Z}_{1,\sigma} - A_1\boldsymbol{Z}_1\|\right] = \sigma \mathbb{E}\left[\|A\boldsymbol{\zeta}\|\right] \le \sigma\sqrt{d},$$

where the final inequality is because $A\boldsymbol{\zeta} \sim \mathcal{N}(0, \boldsymbol{I}_d)$ and Lemma H.8. Similarly, it can get

$$\mathcal{W}_1(\mathbb{P}_{X|Y}^\sigma(\cdot \mid Y = 2), \mathbb{P}_{X|Y}(\cdot \mid Y = 2)) \le \sigma\sqrt{d}.$$

Combining these two inequality and by Lemma H.9, we have

$$\begin{aligned} \mathcal{W}_1(\mathbb{P}_X^\sigma, \mathbb{P}_X) &\le w_1 \mathcal{W}_1(\mathbb{P}_{X|Y}^\sigma(\cdot \mid Y = 1), \mathbb{P}_{X|Y}(\cdot \mid Y = 1)) \\ &\quad + w_2 \mathcal{W}_1(\mathbb{P}_{X|Y}^\sigma(\cdot \mid Y = 2), \mathbb{P}_{X|Y}(\cdot \mid Y = 2)) \\ &\le \sigma\sqrt{d}. \qquad\qquad\qquad\qquad\qquad\qquad\qquad \square \end{aligned}$$

*Proof of Lemma 6.* Let

$$r_1(\boldsymbol{x}) := \frac{w p_1(\boldsymbol{x})}{p(\boldsymbol{x})}, \quad r_2(\boldsymbol{x}) := 1 - r_1(\boldsymbol{x}) = \frac{(1-w)p_2(\boldsymbol{x})}{p(\boldsymbol{x})}.$$

We have

$$\nabla \log p = \frac{w\nabla p_1 + (1-w)\nabla p_2}{p} = r_1 \nabla \log p_1 + r_2 \nabla \log p_2,$$

and

$$\nabla^2 \log p = r_1 \nabla^2 \log p_1 + r_2 \nabla^2 \log p_2 + \nabla r_1 (\nabla \log p_1 - \nabla \log p_2)^\top.$$

For $r_1 = w p_1 / p$,

$$\begin{aligned} \nabla r_1 &= w \frac{p\nabla p_1 - p_1 \nabla p}{p^2} \\ &= w \frac{(w p_1 + (1-w)p_2)\nabla p_1 - p_1(w\nabla p_1 + (1-w)\nabla p_2)}{p^2} \\ &= \frac{w(1-w)}{p^2}(p_2 \nabla p_1 - p_1 \nabla p_2) \\ &= r_1 r_2 (\nabla \log p_1 - \nabla \log p_2). \end{aligned}$$

Therefore,

$$\nabla^2 \log p = r_1 \nabla^2 \log p_1 + r_2 \nabla^2 \log p_2 + r_1 r_2 (\nabla \log p_1 - \nabla \log p_2)(\nabla \log p_1 - \nabla \log p_2)^\top.$$

For the first two terms, by the assumption,

$$r_1 \nabla^2 \log p_1 + r_2 \nabla^2 \log p_2 \preceq r_1 L_1 \boldsymbol{I}_n + r_2 L_2 \boldsymbol{I}_n \preceq \max\{L_1, L_2\} \boldsymbol{I}_n.$$

For the third term, because $\sup_{\boldsymbol{x}} \|\nabla \log p_1(\boldsymbol{x}) - \nabla \log p_2(\boldsymbol{x})\| \leq M$,

$$\left\| (\nabla \log p_1 - \nabla \log p_2) (\nabla \log p_1 - \nabla \log p_2)^\top \right\|_{\mathrm{op}} \leq M^2,$$

which implies that

$$(\nabla \log p_1 - \nabla \log p_2) (\nabla \log p_1 - \nabla \log p_2)^\top \preceq M^2 \boldsymbol{I}_n.$$

For the coefficients $r_1 r_2$, because $r_1, r_2 \in (0, 1)$, $r_1 r_2 \leq 1/4$. Combining these results, we have

$$\nabla^2 \log p \preceq \left( \max\{L_1, L_2\} + \frac{1}{4} M^2 \right) \boldsymbol{I}_n. \qquad \square$$

*Proof of Corollary 9.* Because

$$m_0^z = m_0^z(\sigma) = \frac{m}{1 + m\sigma^2} - \frac{M^2}{4}$$

is decreasing in $\sigma$,

$$m_0^z \leq m_0^z(0) = m - \frac{M^2}{4}.$$

With the Assumption III, we have

$$m - \frac{M^2}{4} > 1$$

Therefore, by choosing a small $\sigma$, we can also have $m_0^z > 1$. It follows that

$$m_t = \frac{m_0^z}{m_0^z + (1 - m_0^z)e^{-2t}}$$

is decreasing in $t$. So

$$m_I := \inf_{t \in (0, T]} m_t = m_T = \frac{m_0^z}{m_0^z + (1 - m_0^z)e^{-2T}} > 1. \qquad \square$$

## D.2  Proof of Theorem 10

*Proof of Theorem 10.* By differentiating $\left\| \tilde{\boldsymbol{Y}}_t \right\|^2$ from (14),

$$\begin{aligned}
\frac{1}{2} \frac{\mathrm{d}}{\mathrm{d}t} \left\| \tilde{\boldsymbol{Y}}_t \right\|^2 &= \left\langle \tilde{\boldsymbol{Y}}_t, \frac{\mathrm{d}}{\mathrm{d}t} \tilde{\boldsymbol{Y}}_t \right\rangle \\
&= \left\langle \tilde{\boldsymbol{Y}}_t, \tilde{\boldsymbol{Y}}_t + P_1 \nabla_{\boldsymbol{x}} \log p_{T-t}^\sigma(\tilde{\boldsymbol{X}}_t) - \eta \tilde{\boldsymbol{Y}}_t \right\rangle \\
&= (1 - \eta) \left\| \tilde{\boldsymbol{Y}}_t \right\|^2 + \left\langle \tilde{\boldsymbol{Y}}_t, P_1 \nabla_{\boldsymbol{x}} \log p_{T-t}^\sigma(\tilde{\boldsymbol{X}}_t) \right\rangle \\
&\leq (1 - \eta) \left\| \tilde{\boldsymbol{Y}}_t \right\|^2 + \left\| \tilde{\boldsymbol{Y}}_t \right\| \left\| \nabla_{\boldsymbol{x}} \log p_{T-t}^\sigma(\tilde{\boldsymbol{X}}_t) \right\|.
\end{aligned}$$

Therefore,

$$\frac{\mathrm{d}}{\mathrm{d}t} \left\| \tilde{\boldsymbol{Y}}_t \right\| \leq (1 - \eta) \left\| \tilde{\boldsymbol{Y}}_t \right\| + \left\| \nabla_{\boldsymbol{x}} \log p_{T-t}^\sigma(\tilde{\boldsymbol{X}}_t) \right\|.$$

Taking the expectation on the both sides yields

$$\frac{\mathrm{d}}{\mathrm{d}t} \mathfrak{m}_t \leq (1 - \eta) \mathfrak{m}_t + \mathbb{E}\left[ \left\| \nabla_{\boldsymbol{x}} \log p_{T-t}^\sigma(\tilde{\boldsymbol{X}}_t) \right\| \right], \qquad (30)$$

where $\mathfrak{m}_t := \mathbb{E}\left[ \left\| \tilde{\boldsymbol{Y}}_t \right\| \right]$. Therefore, the next step is to bound $\mathbb{E}\left[ \left\| \nabla_{\boldsymbol{x}} \log p_{T-t}^\sigma(\tilde{\boldsymbol{X}}_t) \right\| \right]$.

Let

$$L_S := \sup_{t \in [\delta, T]} L_t, \quad C := \sup_{t \in [\delta, T]} \left\| \nabla_{\boldsymbol{x}} \log p_t^\sigma(\boldsymbol{0}) \right\| < \infty, \tag{31}$$

where $L_t$ is defined in Theorem 8. By the $L_S$-Lipschitz of $\nabla_{\boldsymbol{x}} \log p_t^\sigma$ (Theorem 8),

$$\left\| \nabla_{\boldsymbol{x}} \log p_{T-t}^\sigma(\tilde{\boldsymbol{X}}_t) \right\| \le \left\| \nabla_{\boldsymbol{x}} \log p_{T-t}^\sigma(\tilde{\boldsymbol{X}}_t) - \nabla_{\boldsymbol{x}} \log p_{T-t}^\sigma(\boldsymbol{0}) \right\| + \left\| \nabla_{\boldsymbol{x}} \log p_{T-t}^\sigma(\boldsymbol{0}) \right\|$$
$$\le L_S \left\| \tilde{\boldsymbol{X}}_t \right\| + C \tag{32}$$

For $\tilde{\boldsymbol{X}}_t$ in Equation $(*)$, we have

$$\frac{\mathrm{d}}{\mathrm{d}t} \left\| \tilde{\boldsymbol{X}}_t \right\|^2 = 2 \left\| \tilde{\boldsymbol{X}}_t \right\|^2 + 2 \left\langle \tilde{\boldsymbol{X}}_t, \nabla_{\boldsymbol{x}} \log p_{T-t}^\sigma(\tilde{\boldsymbol{X}}_t) \right\rangle - 2\eta \left\langle \tilde{\boldsymbol{X}}_t, P_1 \tilde{\boldsymbol{X}}_t \right\rangle$$
$$\le 2 \left\| \tilde{\boldsymbol{X}}_t \right\|^2 + 2 \left\langle \tilde{\boldsymbol{X}}_t, \nabla_{\boldsymbol{x}} \log p_{T-t}^\sigma(\tilde{\boldsymbol{X}}_t) \right\rangle$$
$$\le 2 \left\| \tilde{\boldsymbol{X}}_t \right\|^2 + 2 \left\| \tilde{\boldsymbol{X}}_t \right\| \left\| \nabla_{\boldsymbol{x}} \log p_{T-t}^\sigma(\tilde{\boldsymbol{X}}_t) \right\|,$$

where the second inequality is because $\left\langle \tilde{\boldsymbol{X}}_t, P_1 \tilde{\boldsymbol{X}}_t \right\rangle \ge 0$. Combining this with (32),

$$\frac{\mathrm{d}}{\mathrm{d}t} \left\| \tilde{\boldsymbol{X}}_t \right\| \le (1 + L_S) \left\| \tilde{\boldsymbol{X}}_t \right\| + C.$$

By taking the expectation on the both sides of above inequality, Grönwall's Inequality (Lemma H.11) implies

$$\mathbb{E} \left[ \left\| \tilde{\boldsymbol{X}}_t \right\| \right] \le \mathbb{E} \left[ \left\| \tilde{\boldsymbol{X}}_0 \right\| \right] e^{(1+L_S)t} + \frac{C}{1 + L_S} \left( e^{(1+L_S)t} - 1 \right). \tag{33}$$

Because $\tilde{\boldsymbol{X}}_0 \sim \mathcal{N}(\boldsymbol{0}, \boldsymbol{I}_D)$, $\mathbb{E} \left[ \left\| \tilde{\boldsymbol{X}}_0 \right\| \right] \le \sqrt{D}$ (Lemma H.8). It follows that

$$\sup_{t \in [0, T-\delta]} \mathbb{E} \left[ \left\| \tilde{\boldsymbol{X}}_t \right\| \right] \le \sqrt{D} e^{(1+L_S)T} + \frac{C}{1 + L_S} \left( e^{(1+L_S)T} - 1 \right) =: M_1,$$

and (32) implies

$$\sup_{t \in [0, T-\delta]} \mathbb{E} \left[ \left\| \nabla_{\boldsymbol{x}} \log p_{T-t}^\sigma(\tilde{\boldsymbol{X}}_t) \right\| \right] \le \sup_{t \in [0, T-\delta]} L_S \mathbb{E} \left[ \left\| \tilde{\boldsymbol{X}}_t \right\| \right] + C$$
$$\le L_S M_1 + C =: M_2. \tag{34}$$

Then by substituting this into (30),

$$\frac{\mathrm{d}}{\mathrm{d}t} \mathfrak{m}_t \le -(\eta - 1) \mathfrak{m}_t + M_2.$$

Because $\mathfrak{m}_0 = \mathbb{E} \left[ \left\| \tilde{\boldsymbol{Y}}_0 \right\| \right] \le \sqrt{D - d_1}$ by Lemma H.8, by applying Grönwall's Inequality again, we obtain

$$\mathbb{E} \left[ \left\| \tilde{\boldsymbol{Y}}_t \right\| \right] = \mathfrak{m}_t \le \sqrt{D - d_1} e^{-(\eta-1)t} + \frac{M_2}{\eta - 1} \left( 1 - e^{-(\eta-1)t} \right) =: M_\eta(t), \tag{35}$$

which implies that

$$\mathbb{E} \left[ \left\| \tilde{\boldsymbol{Y}}_{T-\delta} \right\| \right] \le \sqrt{D - d_1} e^{-(\eta-1)(T-\delta)} + \frac{M_2}{\eta - 1}.$$

For any $\varepsilon > 0$,

$$\frac{M_2}{\eta - 1} \le \frac{\varepsilon}{2} \ \Rightarrow \ \eta \ge \frac{2M_2}{\varepsilon} + 1,$$

and

$$\sqrt{D - d_1} e^{-(\eta-1)(T-\delta)} \le \frac{\varepsilon}{2} \ \Rightarrow \ \eta \ge \frac{1}{T - \delta} \log \frac{2\sqrt{D - d_1}}{\varepsilon} + 1.$$

Therefore, for any $\varepsilon > 0$, by choosing

$$\eta \ge \max \left\{ \frac{2M_2}{\varepsilon}, \frac{1}{T - \delta} \log \frac{2\sqrt{D - d_1}}{\varepsilon} \right\} + 1,$$

we have

$$\mathbb{E} \left[ \left\| \tilde{\boldsymbol{Y}}_{T-\delta} \right\| \right] \le \varepsilon. \qquad \square$$

### D.3 Theoretical Analysis for Universal Guidance

Consider the universal guidance model

$$\frac{\mathrm{d}\boldsymbol{X}_t^{\leftarrow}}{\mathrm{d}t} = \boldsymbol{X}_t^{\leftarrow} + \nabla_{\boldsymbol{x}} \log p_{T-t}(\boldsymbol{X}_t^{\leftarrow}) - \eta \nabla_{\boldsymbol{x}} f(\boldsymbol{X}_t^{\leftarrow}), \ \ \boldsymbol{X}_0^{\leftarrow} \sim \mathcal{N}(\boldsymbol{0}, \boldsymbol{I}_D), \tag{36}$$

for $t \in [0, T]$, where $p_t$ is the density function in DDPMs.

**Theorem D.1.** *For the dynamics (36), assume that $\log p_t$ is $L$-smoothness, $f$ is $\rho$-strongly convex, and $\mathbb{E}\left[f(\boldsymbol{X}_0^{\leftarrow})\right] < \infty$. Then*

$$\mathbb{E}\left[f(\boldsymbol{X}_T^{\leftarrow})\right] - f(\boldsymbol{x}_*) = \mathcal{O}\left(e^{-\eta} + \frac{1}{\eta}\right),$$

*where $\boldsymbol{x}_*$ is the unique minimizer of $f$.*

*Proof.* By differentiating $f(\boldsymbol{X}_t^{\leftarrow})$,

$$
\begin{aligned}
\frac{\mathrm{d}}{\mathrm{d}t} f(\boldsymbol{X}_t^{\leftarrow}) &= \left\langle \nabla_{\boldsymbol{x}} f(\boldsymbol{X}_t^{\leftarrow}), \frac{\mathrm{d}}{\mathrm{d}t} \boldsymbol{X}_t^{\leftarrow} \right\rangle \\
&= \langle \nabla_{\boldsymbol{x}} f(\boldsymbol{X}_t^{\leftarrow}), \boldsymbol{X}_t^{\leftarrow} + \nabla_{\boldsymbol{x}} \log p_{T-t}(\boldsymbol{X}_t^{\leftarrow}) - \eta \nabla_{\boldsymbol{x}} f(\boldsymbol{X}_t^{\leftarrow}) \rangle \\
&\le \|\boldsymbol{X}_t^{\leftarrow}\| \|\nabla_{\boldsymbol{x}} f(\boldsymbol{X}_t^{\leftarrow})\| + \|\nabla_{\boldsymbol{x}} \log p_{T-t}(\boldsymbol{X}_t^{\leftarrow})\| \|\nabla_{\boldsymbol{x}} f(\boldsymbol{X}_t^{\leftarrow})\| - \eta \|\nabla_{\boldsymbol{x}} f(\boldsymbol{X}_t^{\leftarrow})\|^2.
\end{aligned}
$$

Let $C = \sup_{t \in [\delta, T]} \|\nabla_{\boldsymbol{x}} \log p_t(\boldsymbol{0})\| < \infty$. Then, the $L$-smoothness of $\log p_t$ implies that

$$\|\nabla_{\boldsymbol{x}} \log p_{T-t}(\boldsymbol{X}_t^{\leftarrow})\| \le L\|\boldsymbol{X}_t^{\leftarrow}\| + C.$$

Therefore, by $ab \le (a^2 + b^2)/2$, we have

$$
\begin{aligned}
\frac{\mathrm{d}}{\mathrm{d}t} f(\boldsymbol{X}_t^{\leftarrow}) &\le (1 + L)\|\boldsymbol{X}_t^{\leftarrow}\| \|\nabla_{\boldsymbol{x}} f(\boldsymbol{X}_t^{\leftarrow})\| + C\|\nabla_{\boldsymbol{x}} f(\boldsymbol{X}_t^{\leftarrow})\| - \eta \|\nabla_{\boldsymbol{x}} f(\boldsymbol{X}_t^{\leftarrow})\|^2 \\
&\le \frac{1 + L}{2}\left(\|\boldsymbol{X}_t^{\leftarrow}\|^2 + \|\nabla_{\boldsymbol{x}} f(\boldsymbol{X}_t^{\leftarrow})\|^2\right) + \frac{1}{2}\left(C^2 + \|\nabla_{\boldsymbol{x}} f(\boldsymbol{X}_t^{\leftarrow})\|^2\right) - \eta \|\nabla_{\boldsymbol{x}} f(\boldsymbol{X}_t^{\leftarrow})\|^2 \\
&= -\frac{1}{2}(\eta - 2 - L)\|\nabla_{\boldsymbol{x}} f(\boldsymbol{X}_t^{\leftarrow})\|^2 + \frac{1 + L}{2}\|\boldsymbol{X}_t^{\leftarrow}\|^2 + \frac{C^2}{2}
\end{aligned}
$$

Because $f$ is $\rho$-strongly convex, by Lemma H.12, it satisfies the $\rho$-PL inequality,

$$\|\nabla_{\boldsymbol{x}} f(\boldsymbol{X}_t^{\leftarrow})\|^2 \ge 2\rho\left(f(\boldsymbol{X}_t^{\leftarrow}) - f(\boldsymbol{x}_*)\right),$$

For $\eta > L + 2$, we obtain

$$\frac{\mathrm{d}}{\mathrm{d}t} f(\boldsymbol{X}_t^{\leftarrow}) \le -\rho(\eta - 2 - L)\left(f(\boldsymbol{X}_t^{\leftarrow}) - f(\boldsymbol{x}_*)\right) + \frac{1 + L}{2}\|\boldsymbol{X}_t^{\leftarrow}\|^2 + \frac{C^2}{2}.$$

Taking the expectation on the both sides yields that

$$\frac{\mathrm{d}}{\mathrm{d}t} \mathbb{E}\left[f(\boldsymbol{X}_t^{\leftarrow})\right] \le -\rho(\eta - 2 - L)\left(\mathbb{E}\left[f(\boldsymbol{X}_t^{\leftarrow})\right] - f(\boldsymbol{x}_*)\right) + \frac{1 + L}{2} \mathbb{E}\left[\|\boldsymbol{X}_t^{\leftarrow}\|^2\right] + \frac{C^2}{2}. \tag{37}$$

The next step is to bound $\mathbb{E}\left[\|\boldsymbol{X}_t^{\leftarrow}\|^2\right]$. Let $\boldsymbol{R}_t := \boldsymbol{X}_t^{\leftarrow} - \boldsymbol{x}_*$. Then

$$
\begin{aligned}
\frac{1}{2}\frac{\mathrm{d}}{\mathrm{d}t}\|\boldsymbol{R}_t\|^2 &= \langle \boldsymbol{R}_t, \boldsymbol{X}_t^{\leftarrow} + \nabla_{\boldsymbol{x}} \log p_{T-t}(\boldsymbol{X}_t^{\leftarrow}) - \eta \nabla_{\boldsymbol{x}} f(\boldsymbol{X}_t^{\leftarrow}) \rangle \\
&= \langle \boldsymbol{R}_t, \boldsymbol{X}_t^{\leftarrow} \rangle + \langle \boldsymbol{R}_t, \nabla_{\boldsymbol{x}} \log p_{T-t}(\boldsymbol{X}_t^{\leftarrow}) \rangle - \eta \langle \boldsymbol{R}_t, \nabla_{\boldsymbol{x}} f(\boldsymbol{X}_t^{\leftarrow}) \rangle.
\end{aligned} \tag{38}
$$

To obtain the desired inequality, we consider these three terms respectively. For the first term,

$$\langle \boldsymbol{R}_t, \boldsymbol{X}_t^{\leftarrow} \rangle = \|\boldsymbol{R}_t\|^2 + \langle \boldsymbol{R}_t, \boldsymbol{x}_* \rangle \le \|\boldsymbol{R}_t\|^2 + \|\boldsymbol{x}_*\| \|\boldsymbol{R}_t\|. \tag{39}$$

Let $c = \|\nabla_{\boldsymbol{x}} \log p_{T-t}(\boldsymbol{x}_*)\|$. By the $L$-smoothness of $\log p_t$, we have

$$\|\log p_{T-t}(\boldsymbol{X}_t^{\leftarrow})\| \leq \|\log p_{T-t}(\boldsymbol{X}_t^{\leftarrow}) - \nabla_{\boldsymbol{x}} \log p_{T-t}(\boldsymbol{x}_*)\| + \|\nabla_{\boldsymbol{x}} \log p_{T-t}(\boldsymbol{x}_*)\|$$
$$\leq L\|\boldsymbol{R}_t\| + c.$$

Therefore, for the second term,

$$\langle \boldsymbol{R}_t, \nabla_{\boldsymbol{x}} \log p_{T-t}(\boldsymbol{X}_t^{\leftarrow}) \rangle \leq \|\boldsymbol{R}_t\| \|\nabla_{\boldsymbol{x}} \log p_{T-t}(\boldsymbol{X}_t^{\leftarrow})\|$$
$$\leq L\|\boldsymbol{R}_t\|^2 + c\|\boldsymbol{R}_t\|. \tag{40}$$

For the third term, because $f$ is $\rho$-strongly convex, $\nabla_{\boldsymbol{x}} f(\boldsymbol{x}_*) = 0$ and

$$\langle \boldsymbol{R}_t, \nabla_{\boldsymbol{x}} f(\boldsymbol{X}_t^{\leftarrow}) \rangle = \langle \boldsymbol{R}_t, \nabla_{\boldsymbol{x}} f(\boldsymbol{X}_t^{\leftarrow}) - \nabla_{\boldsymbol{x}} f(\boldsymbol{x}_*) \rangle \geq \rho\|\boldsymbol{R}_t\|^2. \tag{41}$$

Then, by combining (38) with (39) (40) (41), we have

$$\frac{\mathrm{d}}{\mathrm{d}t} \|\boldsymbol{R}_t\|^2 \leq 2(L + 1 - \eta\rho)\|\boldsymbol{R}_t\|^2 + 2\tilde{c}\|\boldsymbol{R}_t\|$$
$$\leq (2L + 3 - 2\eta\rho)\|\boldsymbol{R}_t\|^2 + \tilde{c}^2.$$

where $\tilde{c} = \|\boldsymbol{x}_*\| + c$. By taking the expectation on the both sides, Grönwall's Inequality (Lemma H.11) implies that

$$\mathbb{E}\left[\|\boldsymbol{R}_t\|^2\right] \leq \mathbb{E}\left[\|\boldsymbol{R}_0\|^2\right] e^{-(2\eta\rho - 2L - 3)t} + \frac{\tilde{c}^2}{2\eta\rho - 2L - 3}\left(1 - e^{-(2\eta\rho - 2L - 3)t}\right)$$

By taking a sufficiently large $\eta$ such that $2\eta\rho - 2L - 3 > \tilde{c}^2 > 0$, we have

$$\mathbb{E}\left[\|\boldsymbol{R}_t\|^2\right] \leq \mathbb{E}\left[\|\boldsymbol{R}_0\|^2\right] + 1$$

Note that $\boldsymbol{X}_0^{\leftarrow} \sim \mathcal{N}(\boldsymbol{0}, \boldsymbol{I}_D)$, which implies that $\mathbb{E}\left[\|\boldsymbol{X}_0^{\leftarrow}\|^2\right] = D$. Therefore,

$$\mathbb{E}\left[\|\boldsymbol{R}_0\|^2\right] \leq \mathbb{E}\left[\|\boldsymbol{X}_0^{\leftarrow}\|^2\right] + \|\boldsymbol{x}_*\|^2 \leq D + \|\boldsymbol{x}_*\|^2,$$

and

$$\mathbb{E}\left[\|\boldsymbol{X}_t^{\leftarrow}\|^2\right] \leq \mathbb{E}\left[\|\boldsymbol{R}_t\|^2\right] + \|\boldsymbol{x}_*\|^2 \leq D + 2\|\boldsymbol{x}_*\|^2 + 1 =: M_3.$$

By substituting $M_3$ into (37), we obtain

$$\frac{\mathrm{d}}{\mathrm{d}t} \mathbb{E}\left[f(\boldsymbol{X}_t^{\leftarrow})\right] \leq -\rho(\eta - 2 - L)\left(\mathbb{E}\left[f(\boldsymbol{X}_t^{\leftarrow})\right] - f(\boldsymbol{x}_*)\right) + M_4$$

for $M_4 := ((1 + L)M_3 + C^2)/2$. Then, by Grönwall's Inequality,

$$\mathbb{E}\left[f(\boldsymbol{X}_T^{\leftarrow})\right] - f(\boldsymbol{x}_*) \leq \left(\mathbb{E}\left[f(\boldsymbol{X}_0^{\leftarrow})\right] - f(\boldsymbol{x}_*)\right) e^{-\rho(\eta - 2 - L)T} + \frac{M_4}{\rho(\eta - 2 - L)},$$

which means that

$$\mathbb{E}\left[f(\boldsymbol{X}_T^{\leftarrow})\right] - f(\boldsymbol{x}_*) = \mathcal{O}\left(e^{-\eta} + \frac{1}{\eta}\right). \qquad \square$$

### D.4  Proof of Theorem 11

*Proof of Theorem 11.* The proof consists two main steps:

(i) Let $Q_1 = A_1 A_1^\top$. For any coupling $(\tilde{\boldsymbol{X}}, \boldsymbol{X}) \sim (\tilde{p}_{T-\delta}, \mathbb{P}_{X|Y}(\cdot \mid Y = 1))$, we have

$$
\begin{aligned}
\mathcal{W}_1\left(\tilde{p}_{T-\delta}, \mathbb{P}_{X|Y}(\cdot \mid Y = 1)\right) &\leq \mathbb{E}\left[\|\tilde{\boldsymbol{X}} - \boldsymbol{X}\|\right] \\
&= \mathbb{E}\left[\|Q_1\tilde{\boldsymbol{X}} - Q_1\boldsymbol{X}\|\right] + \mathbb{E}\left[\|P_1\tilde{\boldsymbol{X}} - P_1\boldsymbol{X}\|\right] \\
&= \mathbb{E}\left[\|Q_1\tilde{\boldsymbol{X}} - \boldsymbol{X}\|\right] + \mathbb{E}\left[\|\tilde{\boldsymbol{Y}}_{T-\delta}\|\right],
\end{aligned}
$$

where the final equality holds because $\boldsymbol{X} \sim \mathbb{P}_{X|Y}(\cdot \mid Y = 1)$ implies that $Q_1\boldsymbol{X} = \boldsymbol{X}$, and $\tilde{\boldsymbol{X}} \sim \tilde{p}_{T-\delta}$ implies that $P_1\tilde{\boldsymbol{X}} = \tilde{\boldsymbol{Y}}_{T-\delta}$. And by (35),

$$
\mathbb{E}\left[\|\tilde{\boldsymbol{Y}}_{T-\delta}\|\right] \leq M_\eta(T - \delta) = \mathcal{O}(e^{-T} + \eta^{-1}).
$$

Let $(Q_1\tilde{\boldsymbol{X}}, \boldsymbol{X})$ be chosen as the optimal coupling for $((Q_1)_\# \tilde{p}_{T-\delta}, \mathbb{P}_{X|Y}(\cdot \mid Y = 1))$, i.e.,

$$
\mathcal{W}_1\left((Q_1)_\# \tilde{p}_{T-\delta}, \mathbb{P}_{X|Y}(\cdot \mid Y = 1)\right) = \mathbb{E}\left[\|Q_1\tilde{\boldsymbol{X}} - \boldsymbol{X}\|\right].
$$

Therefore, we have

$$
\mathcal{W}_1\left(\tilde{p}_{T-\delta}, \mathbb{P}_{X|Y}(\cdot \mid Y = 1)\right) \leq \mathcal{W}_1\left((Q_1)_\# \tilde{p}_{T-\delta}, \mathbb{P}_{X|Y}(\cdot \mid Y = 1)\right) + \mathcal{O}(e^{-T} + \eta^{-1}). \tag{42}
$$

(ii) For $\mathcal{W}_1\left((Q_1)_\# \tilde{p}_{T-\delta}, \mathbb{P}_{X|Y}(\cdot \mid Y = 1)\right)$, by the triangular inequality,

$$
\begin{aligned}
\mathcal{W}_1\left((Q_1)_\# \tilde{p}_{T-\delta}, \mathbb{P}_{X|Y}(\cdot \mid Y = 1)\right) &\leq \mathcal{W}_1\left((Q_1)_\# \tilde{p}_{T-\delta}, (Q_1)_\# \mathbb{P}_X\right) \\
&\quad + \mathcal{W}_1\left((Q_1)_\# \mathbb{P}_X, \mathbb{P}_{X|Y}(\cdot \mid Y = 1)\right) \\
&\leq \mathcal{W}_1\left(\tilde{p}_{T-\delta}, \mathbb{P}_X\right) + \mathcal{W}_1\left((Q_1)_\# \mathbb{P}_X, \mathbb{P}_{X|Y}(\cdot \mid Y = 1)\right),
\end{aligned} \tag{43}
$$

where the final inequality is because $Q_1$ is an orthogonal projection (Lemma H.13).

By Lemma D.3, the second term in above inequality is bounded by

$$
\mathcal{W}_1\left((Q_1)_\# \mathbb{P}_X, \mathbb{P}_{X|Y}(\cdot \mid Y = 1)\right) \leq \tilde{C}_1 \tag{44}
$$

for some constant $\tilde{C}_1$. For the first term, it can be divided into

$$
\mathcal{W}_1\left(\tilde{p}_{T-\delta}, \mathbb{P}_X\right) \leq \mathcal{W}_1\left(\tilde{p}_{T-\delta}, \hat{p}_\delta\right) + \mathcal{W}_1\left(\hat{p}_\delta, p_\delta^\sigma\right) + \mathcal{W}_1\left(p_\delta^\sigma, \mathbb{P}_X^\sigma\right) + \mathcal{W}_1\left(\mathbb{P}_X^\sigma, \mathbb{P}_X\right), \tag{45}
$$

where $\hat{p}_t$ is defined in dynamics (51), $p_t^\sigma$ is the density evolving in the DDPM initialized from $\mathbb{P}_X^\sigma$; see (13), and $\mathbb{P}_X^\sigma$ is defined in Proposition 5. For the four terms in (45):

(a) By Proposition D.5 and $m_I > 1$ (Corollary 9),

$$
\mathcal{W}_1\left(\tilde{p}_{T-\delta}, \hat{p}_\delta^\sigma\right) \leq \mathcal{O}(e^{-T} + \eta^{-1}) + \tilde{C}_2 \tag{46}
$$

for some constant $\tilde{C}_2$.

(b) By Proposition D.4,

$$
\mathcal{W}_1\left(\hat{p}_\delta, p_\delta^\sigma\right) \leq \mathcal{O}(e^{-T}). \tag{47}
$$

(c) Note that

$$
\boldsymbol{X}_\delta^\sigma = \sqrt{\alpha_\delta}A\boldsymbol{Z} + \sqrt{1 - \alpha_\delta}\boldsymbol{\xi} \sim p_\delta^\sigma, \quad \alpha_\delta = e^{-2\delta}
$$

for $\boldsymbol{Z} \sim p_\sigma^Z$. Moreover, $A\boldsymbol{Z} \sim \mathbb{P}_X^\sigma$. Therefore,

$$
\begin{aligned}
\mathcal{W}_1\left(p_\delta^\sigma, \mathbb{P}_X^\sigma\right) &\leq \mathbb{E}\left[\|\boldsymbol{X}_\delta^\sigma - A\boldsymbol{Z}\|\right] \\
&\leq \mathbb{E}\left[\|\boldsymbol{X}_\delta^\sigma - \sqrt{\alpha_\delta}A\boldsymbol{Z}\|\right] + (1 - \sqrt{\alpha_\delta})\mathbb{E}[\|A\boldsymbol{Z}\|] \\
&= \sqrt{1 - \alpha_\delta}\mathbb{E}\left[\|\boldsymbol{\xi}\|\right] + (1 - \sqrt{\alpha_\delta})\mathbb{E}_{\boldsymbol{Z} \sim p_\sigma^Z}\left[\|\boldsymbol{Z}\|\right] \\
&\leq \sqrt{2\delta D} + \delta\mathfrak{m}_\sigma^Z,
\end{aligned}
$$

where $\mathfrak{m}_\sigma^Z = \mathbb{E}_{\boldsymbol{Z} \sim p_\sigma^Z}\left[\|\boldsymbol{Z}\|\right] < \infty$ by Lemma D.2. It follows that

$$
\mathcal{W}_1\left(p_\delta^\sigma, \mathbb{P}_X^\sigma\right) \leq \mathcal{O}(\delta^{1/2}). \tag{48}
$$

(d) By Proposition 5,
$$\mathcal{W}_1\left(\mathbb{P}_X^\sigma, \mathbb{P}_X\right) \le \mathcal{O}(\sigma). \tag{49}$$

Then, combining (46) (47) (48) (49) with (45), we have
$$\mathcal{W}_1\left(\tilde{p}_{T-\delta}, \mathbb{P}_X\right) \le \mathcal{O}(e^{-T} + \delta^{1/2} + \sigma + \eta^{-1}) + \tilde{C}_2. \tag{50}$$

Combining (50) (44) with (43), it follows
$$\mathcal{W}_1\left((Q_1)_\#\tilde{p}_{T-\delta}, \mathbb{P}_{X|Y}(\cdot \mid Y = 1)\right) \le \mathcal{O}(e^{-T} + \delta^{1/2} + \sigma + \eta^{-1}) + \tilde{C},$$

where $\tilde{C} = \tilde{C}_1 + \tilde{C}_2$. Therefore, substituting this in (42), we obtain
$$\mathcal{W}_1\left(\tilde{p}_{T-\delta}, \mathbb{P}_{X|Y}(\cdot \mid Y = 1)\right) \le \mathcal{O}(e^{-T} + \delta^{1/2} + \sigma + \eta^{-1}) + \tilde{C}. \qquad \square$$

*Remark* D.1. For the error floor $\tilde{C}$, we provide two further discussions.

(i) First, it follows from the above proof that $\tilde{C} = \tilde{C}_1 + \tilde{C}_2$, where

- $\tilde{C}_1$ is determined by (44) and Lemma D.3,
$$\tilde{C}_1 = w_2 \mathfrak{m}_1^Z, \quad \mathfrak{m}_1^Z := \mathfrak{m}_1^Z = \mathbb{E}_{\boldsymbol{Z} \sim \mathbb{P}_1^Z}[\|\boldsymbol{Z}\|],$$

which is independent of the parameters $T, \delta, \sigma$.

- $\tilde{C}_2$ is given by (46) and Proposition D.5,
$$\tilde{C}_2 = \frac{M_2}{m_I - 1},$$

where $M_2$ is defined in (34) and depends on $L_S = \sup_{t \in [0, T-\delta]} L_t$ and $T$, while $m_I = \inf_{t \in [0, T-\delta]} m_t$. Since $L_t$ and $m_t$ are specified by Theorem 8 through $p_t^\sigma$, $\tilde{C}_2$ depends implicitly on $T$, $\delta$, and $\sigma$.

(ii) We believe the error floor is inherent to the geometric guidance model. Because of the analytical simplicity of the geometric guidance, it cannot provide as much information as the probability guidance term did. More precisely, in Appendix C.3, we show that
$$\|\nabla_{\boldsymbol{x}} \log p_t(y = 1 \mid \boldsymbol{x}) + \eta_t P_1 \boldsymbol{x}\| \le \beta_t, \quad \forall \, \boldsymbol{x} \in \mathcal{M}_1^t,$$

for some scalar $\eta_t > 0$, and $\beta_t = \mathcal{O}(\varepsilon_t)$, when $p_t(y = 1 \mid \boldsymbol{x}) > 1 - \varepsilon_t$ for all $\boldsymbol{x} \in \mathcal{M}_1^t$. This shows that the probabilistic guidance $\nabla_{\boldsymbol{x}} \log p_t(y = 1 \mid \boldsymbol{x})$ is "almost parallel" to the geometric guidance $P_1 \boldsymbol{x}$, but the norm of the probabilistic guidance carries additional information that the geometric term cannot capture. This is a trade-off made for the sake of analytical tractability.

**Lemma D.2.** *Let $\boldsymbol{Z}_i \sim p_i^Z$ for $i = 1, 2$. If $\mathfrak{m}_i^Z = \mathbb{E}\left[\|\boldsymbol{Z}_i\|\right] < \infty$, then for $p_\sigma^Z$ defined in (11) (12),*
$$\mathfrak{m}_\sigma^Z := \mathbb{E}_{\boldsymbol{Z} \sim p_\sigma^Z}\left[\|\boldsymbol{Z}\|\right] < \infty.$$

*Proof.* By the definition of (12),
$$\mathbb{E}\left[\|\boldsymbol{Z}_{i,\sigma}\|\right] \le \mathbb{E}\left[\|\boldsymbol{Z}_i\|\right] + \sigma \mathbb{E}\left[\|\boldsymbol{\zeta}_i\|\right] \le \mathfrak{m}_i^Z + \sigma\sqrt{d} < \infty$$

for $\boldsymbol{Z}_{i,\sigma} \sim p_{i,\sigma}^Z$, where the second inequality is by Lemma H.8. Then, by (11),
$$\mathbb{E}_{\boldsymbol{Z} \sim p_\sigma^Z}\left[\|\boldsymbol{Z}\|\right] = \int_{\mathbb{R}^d} \boldsymbol{z} p_\sigma^Z(\boldsymbol{z}) \mathrm{d}\boldsymbol{z}$$
$$= w_1 \int_{\mathbb{R}^d} \boldsymbol{z} p_{1,\sigma}^Z(\boldsymbol{z}) \mathrm{d}\boldsymbol{z} + w_2 \int_{\mathbb{R}^d} \boldsymbol{z} p_{2,\sigma}^Z(\boldsymbol{z}) \mathrm{d}\boldsymbol{z}$$
$$= w_1 \mathbb{E}\left[\|\boldsymbol{Z}_{1,\sigma}\|\right] + w_2 \mathbb{E}\left[\|\boldsymbol{Z}_{2,\sigma}\|\right] < \infty. \qquad \square$$

**Lemma D.3.** *For*
$$\mathbb{P}_X = w_1 \mathbb{P}_{X|Y}(\cdot \mid Y = 1) + w_2 \mathbb{P}_{X|Y}(\cdot \mid Y = 2)$$

*under Assumption I,*
$$\mathcal{W}_1((Q_1)_{\#}\mathbb{P}_X, \mathbb{P}_{X|Y}(\cdot \mid Y = 1)) \le w_2 \mathfrak{m}_1^Z,$$

*where* $Q_1 = A_1 A_1^\top$ *and* $\mathfrak{m}_1^Z = \mathbb{E}_{\boldsymbol{Z} \sim \mathbb{P}_1^Z}[\|\boldsymbol{Z}\|]$.

*Proof.* First, by Lemma H.7,
$$(Q_1)_{\#}\mathbb{P}_X = (Q_1)_{\#}\mathbb{P}_{X|Y}(\cdot \mid Y = 1) + (Q_1)_{\#}\mathbb{P}_{X|Y}(\cdot \mid Y = 2)$$

For the two terms, if $\boldsymbol{X} \sim \mathbb{P}_{X|Y}(\cdot \mid Y = 1)$, then $Q_1 \boldsymbol{X} = \boldsymbol{X}$, which implies that
$$(Q_1)_{\#}\mathbb{P}_{X|Y}(\cdot \mid Y = 1) = \mathbb{P}_{X|Y}(\cdot \mid Y = 1)$$

On the other hand, $\boldsymbol{X} \sim \mathbb{P}_{X|Y}(\cdot \mid Y = 2)$ implies that $Q_1 \boldsymbol{X} = 0$ so that
$$(Q_1)_{\#}\mathbb{P}_{X|Y}(\cdot \mid Y = 2) = \delta_0,$$

the Dirichlet measure at 0. Therefore, by Lemma H.9,
$$\mathcal{W}_1((Q_1)_{\#}\mathbb{P}_X, \mathbb{P}_{X|Y}(\cdot \mid Y = 1)) \le w_2 \mathcal{W}_1(\delta_0, \mathbb{P}_{X|Y}(\cdot \mid Y = 1)).$$

For any coupling $(\boldsymbol{D}, \boldsymbol{X}) \sim (\delta_0, \mathbb{P}_{X|Y}(\cdot \mid Y = 1))$,
$$\mathcal{W}_1(\delta_0, \mathbb{P}_{X|Y}(\cdot \mid Y = 1)) \le \mathbb{E}\left[\|\boldsymbol{D} - \boldsymbol{X}\|\right]$$
$$= \mathbb{E}\left[\|\boldsymbol{X}\|\right] = \mathbb{E}_{\boldsymbol{Z} \sim \mathbb{P}_1^Z}\left[\|A_1 \boldsymbol{Z}\|\right] = \mathbb{E}_{\boldsymbol{Z} \sim \mathbb{P}_1^Z}\left[\|\boldsymbol{Z}\|\right],$$

where the last two equalities are because $\mathbb{P}_{X|Y}(\cdot \mid Y = 1) = (A_1)_{\#}\mathbb{P}_1^Z$ and $A_1 \in \mathcal{O}^{D \times d_1}$ by Assumption I. $\square$

In the following, unless otherwise specified, we assume that Assumptions I, II, III, and IV hold.

**Proposition D.4.** *Let* $p_t^\sigma$ *be defined in (13). Consider the following two dynamics:*
$$\frac{d\hat{\boldsymbol{X}}_t}{dt} = \hat{\boldsymbol{X}}_t + \nabla_{\boldsymbol{x}} \log p_{T-t}^\sigma(\hat{\boldsymbol{X}}_t), \quad \hat{\boldsymbol{X}}_0 \sim \mathcal{N}(0, \boldsymbol{I}_D) \tag{51}$$

*with the notation* $\hat{\boldsymbol{X}}_t \sim \hat{p}_{T-t}^\sigma$, *and*
$$\frac{d\bar{\boldsymbol{X}}_t}{dt} = \bar{\boldsymbol{X}}_t + \nabla_{\boldsymbol{x}} \log p_{T-t}^\sigma(\bar{\boldsymbol{X}}_t), \quad \bar{\boldsymbol{X}}_0 \sim p_T^\sigma,$$

*where note that* $\bar{\boldsymbol{X}}_t \sim p_{T-t}^\sigma$. *For* $\delta > 0$, *we*
$$\mathcal{W}_1(\hat{p}_\delta^\sigma, p_\delta^\sigma) \le e^{-m_I(T-\delta)}\left(\mathfrak{m}_\sigma^Z + \sqrt{D}\right),$$

*where* $\mathfrak{m}_\sigma^Z = \mathbb{E}_{\boldsymbol{Z} \sim p_\sigma^Z}[\|\boldsymbol{Z}\|]$ *and* $m_I = \inf_{t \in [\delta, T]} m_t$ *is defined in Theorem 8.*

*Proof.* First, by the Theorem 8, $p_{T-t}^\sigma$ is $m_I$-strong log-concavity for $t \in [0, T - \delta]$, which follows that
$$\left\langle \hat{\boldsymbol{X}}_t - \bar{\boldsymbol{X}}_t, \nabla_{\boldsymbol{x}} \log p_{T-t}^\sigma(\hat{\boldsymbol{X}}_t) - \nabla_{\boldsymbol{x}} \log p_{T-t}^\sigma(\bar{\boldsymbol{X}}_t) \right\rangle = \left\langle \hat{\boldsymbol{X}}_t - \bar{\boldsymbol{X}}_t, \nabla_{\boldsymbol{x}}^2 \log p_{T-t}^\sigma(\boldsymbol{x}) \left(\hat{\boldsymbol{X}}_t - \bar{\boldsymbol{X}}_t\right) \right\rangle$$
$$\le -m_I \|\hat{\boldsymbol{X}}_t - \bar{\boldsymbol{X}}_t\|^2.$$

Therefore, we have
$$\frac{d}{dt} \|\hat{\boldsymbol{X}}_t - \bar{\boldsymbol{X}}_t\|^2 = 2 \left\langle \hat{\boldsymbol{X}}_t - \bar{\boldsymbol{X}}_t, \frac{d}{dt}\left(\hat{\boldsymbol{X}}_t - \bar{\boldsymbol{X}}_t\right) \right\rangle$$

$$= 2\|\hat{\boldsymbol{X}}_t - \bar{\boldsymbol{X}}_t\|^2 + 2\left\langle \hat{\boldsymbol{X}}_t - \bar{\boldsymbol{X}}_t, \nabla_{\boldsymbol{x}} \log p^\sigma_{T-t}(\hat{\boldsymbol{X}}_t) - \nabla_{\boldsymbol{x}} \log p^\sigma_{T-t}(\bar{\boldsymbol{X}}_t)\right\rangle$$

$$\leq -2(m_I - 1)\|\hat{\boldsymbol{X}}_t - \bar{\boldsymbol{X}}_t\|^2,$$

which indicates that

$$\frac{\mathrm{d}}{\mathrm{d}t}\|\hat{\boldsymbol{X}}_t - \bar{\boldsymbol{X}}_t\| \leq -(m_I - 1)\|\hat{\boldsymbol{X}}_t - \bar{\boldsymbol{X}}_t\|,$$

Then, by Grönwall's Inequality(Lemma H.11),

$$\|\hat{\boldsymbol{X}}_{T-\delta} - \bar{\boldsymbol{X}}_{T-\delta}\| \leq e^{-(m_I - 1)(T-\delta)}\|\hat{\boldsymbol{X}}_0 - \bar{\boldsymbol{X}}_0\|.$$

Therefore, by the definition of Wasserstein distance,

$$\mathcal{W}_1(\hat{p}^\sigma_\delta, p^\sigma_\delta) \leq \mathbb{E}\left[\|\hat{\boldsymbol{X}}_{T-\delta} - \bar{\boldsymbol{X}}_{T-\delta}\|\right]$$

$$\leq e^{-(m_I - 1)(T-\delta)}\mathbb{E}\left[\|\hat{\boldsymbol{X}}_0 - \bar{\boldsymbol{X}}_0\|\right].$$

By choosing $(\hat{\boldsymbol{X}}_0, \bar{\boldsymbol{X}}_0)$ as the optimal coupling, we obtain that

$$\mathcal{W}_1(\hat{p}^\sigma_\delta, p^\sigma_\delta) \leq e^{-(m_I - 1)(T-\delta)}\mathcal{W}_1(\mathcal{N}(0, \boldsymbol{I}_D), p^\sigma_T). \tag{52}$$

For the right hand side of (52), by the definition of $p^\sigma_t$ in Equation (13), i.e.,

$$\boldsymbol{X}^\sigma_t = \sqrt{\alpha_t}A\boldsymbol{Z} + \sqrt{1 - \alpha_t}\boldsymbol{\xi} \sim p^\sigma_t$$

for $\boldsymbol{Z} \sim p^Z_\sigma$, $\boldsymbol{\xi} \sim \mathcal{N}(0, \boldsymbol{I}_D)$, and $\alpha_t = e^{-2t}$, we have

$$\mathcal{W}_1(p^\sigma_t, \mathcal{N}(0, (1 - \alpha_t)\boldsymbol{I}_D)) \leq \sqrt{\alpha_t}\mathbb{E}\left[\|A\boldsymbol{Z}\|\right] = e^{-t}\mathbb{E}_{\boldsymbol{Z} \sim p^Z_\sigma}\left[\|\boldsymbol{Z}\|\right].$$

Moreover,

$$\mathcal{W}_1(\mathcal{N}(0, (1 - \alpha_T)\boldsymbol{I}_D), \mathcal{N}(0, \boldsymbol{I}_D)) \leq (1 - \sqrt{1 - \alpha_T})\mathbb{E}\left[\|\boldsymbol{\xi}\|\right] \leq e^{-T}\sqrt{D}.$$

Therefore,

$$\mathcal{W}_1(p^\sigma_T, \mathcal{N}(0, \boldsymbol{I}_D)) \leq \mathcal{W}_1(p^\sigma_T, \mathcal{N}(0, (1 - \alpha_T)\boldsymbol{I}_D)) + \mathcal{W}_1(\mathcal{N}(0, (1 - \alpha_T)\boldsymbol{I}_D), \mathcal{N}(0, \boldsymbol{I}_D))$$

$$\leq e^{-T}\left(\mathfrak{m}^Z_\sigma + \sqrt{D}\right).$$

Substituting this in the inequality (52) implies that

$$\mathcal{W}_1(\hat{p}^\sigma_\delta, p^\sigma_\delta) \leq e^{-m_I(T-\delta)-\delta}\left(\mathfrak{m}^Z_\sigma + \sqrt{D}\right) \leq e^{-m_I(T-\delta)}\left(\mathfrak{m}^Z_\sigma + \sqrt{D}\right). \qquad \square$$

**Proposition D.5.** *Consider the geometric guidance model (\*) and the dynamics (51), for the corresponding generated distribution $\tilde{p}^\sigma_t$ and $\hat{p}^\sigma_t$, we have*

$$\mathcal{W}_1\left(\tilde{p}^\sigma_{T-\delta}, \hat{p}^\sigma_\delta\right) \leq \frac{\eta\sqrt{D - d_1}}{\eta - m_I}e^{-(m_I - 1)(T-\delta)} + \frac{\eta M_2}{(m_I - 1)(\eta - 1)},$$

*where $M_2$ is the constant defined in (34).*

*Proof.* By the $m_I$-strong log-concavity of $p^\sigma_t$ (Theorem 8), we have

$$\frac{1}{2}\frac{\mathrm{d}}{\mathrm{d}t}\|\hat{\boldsymbol{X}}_t - \tilde{\boldsymbol{X}}_t\|^2 = \left\langle \hat{\boldsymbol{X}}_t - \tilde{\boldsymbol{X}}_t, \frac{\mathrm{d}}{\mathrm{d}t}\left(\hat{\boldsymbol{X}}_t - \tilde{\boldsymbol{X}}_t\right)\right\rangle$$

$$= \left\langle \hat{\boldsymbol{X}}_t - \tilde{\boldsymbol{X}}_t, \hat{\boldsymbol{X}}_t - \tilde{\boldsymbol{X}}_t + \nabla_{\boldsymbol{x}} \log p^\sigma_{T-t}(\hat{\boldsymbol{X}}_t) - \nabla_{\boldsymbol{x}} \log p^\sigma_{T-t}(\tilde{\boldsymbol{X}}_t) + \eta P_1 \tilde{\boldsymbol{X}}_t\right\rangle$$

$$\leq -(m_I - 1)\|\hat{\boldsymbol{X}}_t - \tilde{\boldsymbol{X}}_t\|^2 + \eta\|\hat{\boldsymbol{X}}_t - \tilde{\boldsymbol{X}}_t\|\|P_1\tilde{\boldsymbol{X}}_t\|$$

Note that $P_1 \tilde{\boldsymbol{X}}_t = \tilde{\boldsymbol{Y}}_t$. It follows that

$$\frac{\mathrm{d}}{\mathrm{d}t} \|\hat{\boldsymbol{X}}_t - \tilde{\boldsymbol{X}}_t\| \leq -(m_I - 1)\|\hat{\boldsymbol{X}}_t - \tilde{\boldsymbol{X}}_t\| + \eta\|\tilde{\boldsymbol{Y}}_t\|.$$

Moreover, by (35), taking the expectation on the both sides yields

$$\frac{\mathrm{d}}{\mathrm{d}t}\mathbb{E}\left[\|\hat{\boldsymbol{X}}_t - \tilde{\boldsymbol{X}}_t\|\right] \leq -(m_I - 1)\mathbb{E}\left[\|\hat{\boldsymbol{X}}_t - \tilde{\boldsymbol{X}}_t\|\right] + \eta M_\eta(t).$$

Then, Grönwall's inequality implies that

$$\mathcal{W}_1\left(\tilde{p}_{T-\delta}, \hat{p}_\delta^\sigma\right) \leq \mathbb{E}\left[\|\hat{\boldsymbol{X}}_{T-\delta} - \tilde{\boldsymbol{X}}_{T-\delta}\|\right]$$

$$\leq \eta \int_0^{T-\delta} M_\eta(s) e^{-(m_I - 1)(T-\delta-s)}\mathrm{d}s =: I(\eta),$$

when the initial coupling is chosen as $\hat{\boldsymbol{X}}_0 = \tilde{\boldsymbol{X}}_0 \sim \mathcal{N}(0, \boldsymbol{I}_D)$. For $I(\eta)$, by the definition of $M_\eta(t)$ in (35), we have

$$I(\eta) \leq \eta \int_0^{T-\delta}\left(\sqrt{D - d_1}e^{-(\eta-1)s} + \frac{M_2}{\eta - 1}\right)e^{-(m_I - 1)(T-\delta-s)}\mathrm{d}s$$

$$= \frac{\eta\sqrt{D - d_1}}{\eta - m_I}e^{-(m_I - 1)(T-\delta)}\left(1 - e^{-(\eta - m_I)(T-\delta)}\right)$$

$$+ \frac{\eta M_2}{(m_I - 1)(\eta - 1)}\left(1 - e^{-(m_I - 1)(T-\delta)}\right)$$

$$\leq \frac{\eta\sqrt{D - d_1}}{\eta - m_I}e^{-(m_I - 1)(T-\delta)} + \frac{\eta M_2}{(m_I - 1)(\eta - 1)}. \qquad \square$$

## D.5 Discretization Error

To clarify why performance degrades in practice when $\eta$ becomes too large, we analyze the discretization error of the geometric guidance model $(*)$. In practice, ODEs are typically solved using the Euler method, while SDEs are solved using the Euler–Maruyama (EM) scheme. Since our model is formulated as a deterministic ODE in $(*)$, we focus on the Euler approximation; the analysis for the corresponding SDE and the EM scheme is analogous.

More specifically, we partition the interval $[0, T - \delta]$ into $N$ subintervals with step size $h = (T - \delta)/N$, and define $t_k = kh$ for $k = 0, 1, \ldots, N$. The Euler scheme then constructs the sequence $\left\{\boldsymbol{X}_k^h\right\}_{k=0}^N$ via

$$\boldsymbol{X}_{k+1}^h = \boldsymbol{X}_k^h + h\left(\boldsymbol{X}_k^h + \nabla_{\boldsymbol{x}}\log p_{T-t_k}^\sigma(\boldsymbol{X}_k^h) - \eta P_1\boldsymbol{X}_k^h\right), \quad \boldsymbol{X}_0^h \sim \mathcal{N}(\boldsymbol{0}, \boldsymbol{I}_D).$$

Let $\boldsymbol{X}_k^h \sim \tilde{p}_k^h$. Our goal is to bound the Wasserstein error $\mathcal{W}_1(\tilde{p}_{T-\delta}, \tilde{p}_N^h)$. Under the Lipschitz continuity of $\nabla_{\boldsymbol{x}}\log p_t^\sigma$, standard results yield $\mathcal{W}_1(\tilde{p}_{T-\delta}, \tilde{p}_N^h) \leq \mathcal{O}(he^\eta)$ (Griffiths & Higham, 2010, Theorem 2.4). Because of Theorem 8, we not only have the $L_S$-smoothness

$$\left\|\nabla_{\boldsymbol{x}}^2\log p_t^\sigma(\boldsymbol{x})\right\|_{\mathrm{op}} \leq L_S, \quad L_S = \sup_{t\in[\delta,T]}L_t,$$

but also the $m_I$-strong log-concavity

$$-\nabla_{\boldsymbol{x}}^2\log p_t^\sigma(\boldsymbol{x}) \succeq m_I\boldsymbol{I}_D, \quad m_I = \inf_{t\in[\delta,T]}m_t.$$

The additional strong log-concavity yields the improved bound

$$\mathcal{W}_1(\tilde{p}_{T-\delta}, \tilde{p}_N^h) \leq \mathcal{O}(h\eta^2).$$

**Theorem D.6.** *Assume that $t \mapsto \nabla_{\boldsymbol{x}} \log p_t^\sigma(\boldsymbol{x})$ is $C^1$ for each $\boldsymbol{x}$, and there exist $A, B \geq 0$ such that*

$$\|\partial_t \nabla_{\boldsymbol{x}} \log p_t^\sigma(\boldsymbol{x})\| \leq A + B\|x\|.$$

*For $\eta > 2$, if $h(\eta - 1) < 1$, we have*

$$\mathcal{W}_1(\tilde{p}_{T-\delta}, \tilde{p}_N^h) \leq \mathcal{O}(h\eta^2).$$

*Proof.* Let

$$b(t, \boldsymbol{x}) = \boldsymbol{x} + \nabla_{\boldsymbol{x}} \log p_{T-t}^\sigma(\boldsymbol{x}) - \eta P_1 \boldsymbol{x}.$$

Let $\Phi_k(\boldsymbol{x}) =: \boldsymbol{x}_{t_{k+1}}$, where $\boldsymbol{x}_t$ is the solution of ODE

$$\frac{\mathrm{d}\boldsymbol{x}_t}{\mathrm{d}t} = b(t, \boldsymbol{x}_t), \quad t \in [t_k, t_{k+1}], \tag{53}$$

with initial value $\boldsymbol{x}_{t_k} = \boldsymbol{x}$. By $(*)$, we can see

$$\tilde{\boldsymbol{X}}_{t_{k+1}} = \Phi_k(\tilde{\boldsymbol{X}}_{t_k})$$

Moreover, define the Euler one-step map

$$\Psi_k(\boldsymbol{x}) = \boldsymbol{x} + hb(t_k, \boldsymbol{x}),$$

so that Euler scheme is

$$\boldsymbol{X}_{k+1}^h = \Psi_k(\boldsymbol{X}_k^h).$$

Therefore,

$$\begin{aligned}
\boldsymbol{e}_{k+1} := \tilde{\boldsymbol{X}}_{t_{k+1}} - \boldsymbol{X}_{k+1}^h &= \Phi_k(\tilde{\boldsymbol{X}}_{t_k}) - \Psi_k(\boldsymbol{X}_k^h) \\
&= \left(\Phi_k(\tilde{\boldsymbol{X}}_{t_k}) - \Phi_k(\boldsymbol{X}_k^h)\right) + \left(\Phi_k(\boldsymbol{X}_k^h) - \Psi_k(\boldsymbol{X}_k^h)\right).
\end{aligned}$$

Next, we analyze these two terms respectively.

(i) By the $m_I$-strong log-concavity, we have

$$\left\langle \nabla_{\boldsymbol{x}} \log p_{T-t}^\sigma(\boldsymbol{x}) - \nabla_{\boldsymbol{x}} \log p_{T-t}^\sigma(\boldsymbol{y}), \boldsymbol{x} - \boldsymbol{y} \right\rangle \leq -m_I\|\boldsymbol{x} - \boldsymbol{y}\|^2.$$

Moreover, since $P_1$ is an orthogonal projection,

$$\langle (\boldsymbol{I}_D - \eta P_1)(\boldsymbol{x} - \boldsymbol{y}), \boldsymbol{x} - \boldsymbol{y} \rangle = \|\boldsymbol{x} - \boldsymbol{y}\|^2 - \eta \|P_1(\boldsymbol{x} - \boldsymbol{y})\|^2 \leq \|\boldsymbol{x} - \boldsymbol{y}\|^2.$$

Therefore,

$$\langle b(t, \boldsymbol{x}) - b(t, \boldsymbol{y}), \boldsymbol{x} - \boldsymbol{y} \rangle \leq -(m_I - 1)\|\boldsymbol{x} - \boldsymbol{y}\|^2$$

Let $\boldsymbol{x}_t, \boldsymbol{y}_t$ be the solution of (53) with the initial value $\boldsymbol{x}_{t_k} = \boldsymbol{x}$ and $\boldsymbol{y}_{t_k} = \boldsymbol{y}$. So we have

$$\frac{\mathrm{d}}{\mathrm{d}t}\|\boldsymbol{x}_t - \boldsymbol{y}_t\|^2 = 2\langle b(t, \boldsymbol{x}_t) - b(t, \boldsymbol{y}_t), \boldsymbol{x}_t - \boldsymbol{y}_t \rangle \leq -2(m_I - 1)\|\boldsymbol{x}_t - \boldsymbol{y}_t\|^2.$$

Then by the Grönwall's Inequality (Lemma H.11),

$$\left\|\boldsymbol{x}_{t_{k+1}} - \boldsymbol{y}_{t_{k+1}}\right\| \leq e^{-(m_I - 1)h}\|\boldsymbol{x} - \boldsymbol{y}\|,$$

which implies that

$$\|\Phi_k(\boldsymbol{x}) - \Phi_k(\boldsymbol{y})\| \leq e^{-(m_I - 1)h}\|\boldsymbol{x} - \boldsymbol{y}\|. \tag{54}$$

(ii) Fix $\boldsymbol{x} \in \mathbb{R}^D$ and let $\boldsymbol{x}_t$ be the solution of (53) with the initial value $\boldsymbol{x}_{t_k} = \boldsymbol{x}$. Note that by definition

$$\Phi_k(\boldsymbol{x}) = \boldsymbol{x} + \int_{t_k}^{t_{k+1}} b(t, \boldsymbol{x}_t) \mathrm{d}t.$$

Therefore,

$$\begin{aligned}
\Phi_k(\boldsymbol{x}) - \Psi_k(\boldsymbol{x}) &= \int_{t_k}^{t_{k+1}} \left( b(t, \boldsymbol{x}_t) - b(t_k, \boldsymbol{x}) \right) \mathrm{d}t \\
&= \int_{t_k}^{t_{k+1}} \left( b(t, \boldsymbol{x}_t) - b(t, \boldsymbol{x}) \right) \mathrm{d}t + \int_{t_k}^{t_{k+1}} \left( b(t, \boldsymbol{x}) - b(t_k, \boldsymbol{x}) \right) \mathrm{d}t.
\end{aligned}$$

For above two terms, we analyze them respectively.

(a) Since $\|\boldsymbol{I}_D - \eta P_1\|_{\mathrm{op}} = \eta - 1$ ($\eta > 2$) and $\nabla_{\boldsymbol{x}} \log p_t$ is $L_S$-Lipschitz continuous, $b(t, \cdot)$ is $K(\eta)$-Lipschitz continuous for $K(\eta) = L_S + \eta - 1$. So

$$\int_{t_k}^{t_{k+1}} \|b(t, \boldsymbol{x}_t) - b(t, \boldsymbol{x})\| \mathrm{d}s \leq K(\eta) \int_{t_k}^{t_{k+1}} \|\boldsymbol{x}_t - \boldsymbol{x}\| \mathrm{d}s$$

Note that

$$\|\boldsymbol{x}_t - \boldsymbol{x}\| = \left\| \int_{t_k}^t b(s, \boldsymbol{x}_s) \mathrm{d}s \right\| \leq \int_{t_k}^t \|b(s, \boldsymbol{x}_s)\| \mathrm{d}s \leq (t - t_k) \sup_{s \in [t_k, t_{k+1}]} \|b(s, \boldsymbol{x}_s)\|.$$

Therefore,

$$\int_{t_k}^{t_{k+1}} \|b(t, \boldsymbol{x}_t) - b(t, \boldsymbol{x})\| \mathrm{d}s \leq \frac{h^2}{2} K(\eta) \sup_{s \in [t_k, t_{k+1}]} \|b(s, \boldsymbol{x}_s)\|. \tag{55}$$

For the right hand side, using same notation as (31), let

$$C = \sup_{t \in [\delta, T]} \|\nabla_{\boldsymbol{x}} \log p_t^\sigma(\boldsymbol{0})\| < \infty.$$

It implies that $\|b(t, \boldsymbol{0})\| \leq C$ and so

$$\|b(s, \boldsymbol{x}_s)\| \leq \|b(s, \boldsymbol{x}_s) - b(s, \boldsymbol{0})\| + \|b(t, \boldsymbol{0})\| \leq C + K(\eta) \|\boldsymbol{x}_s\|.$$

Let $S = \sup_{s \in [t_k, t_{k+1}]} \|\boldsymbol{x}_s\| < \infty$. Using the similar idea as in the proof of Theorem 10 in Appendix D.2,

$$S = \sup_{s \in [t_k, t_{k+1}]} \|\boldsymbol{x}_s\| \leq C_1 \|\boldsymbol{x}\| + C_2$$

where $C_1 = \exp\left((1 + L_S)h\right)$ and $C_2 = C(\exp((1 + L_S)h) - 1)/(1 + L_S)$ as shown in (33) and they are independent of $\eta$. So

$$\sup_{s \in [t_k, t_{k+1}]} \|b(s, \boldsymbol{x}_s)\| \leq C + K(\eta) \sup_{s \in [t_k, t_{k+1}]} \|\boldsymbol{x}_s\| \leq C_1 K(\eta) \|\boldsymbol{x}\| + C_2 K(\eta) + C. \tag{56}$$

Combining (55) and (56), we have

$$\int_{t_k}^{t_{k+1}} \|b(t, \boldsymbol{x}_t) - b(t, \boldsymbol{x})\| \mathrm{d}t \leq \frac{h^2}{2} \left( C_1 K(\eta)^2 \|\boldsymbol{x}\| + C_2 K(\eta)^2 + C K(\eta) \right). \tag{57}$$

(b) Since $b(t, \boldsymbol{x}) - b(t_k, \boldsymbol{x}) = \nabla_{\boldsymbol{x}} \log p_{T-t}^\sigma(\boldsymbol{x}) - \nabla_{\boldsymbol{x}} \log p_{T-t_k}^\sigma(\boldsymbol{x})$,

$$\|b(t, \boldsymbol{x}) - b(t_k, \boldsymbol{x})\| \leq \int_{t_k}^t \|\partial_t \nabla_{\boldsymbol{x}} \log p_{T-s}^\sigma(\boldsymbol{x})\| \mathrm{d}s \leq (t - t_k)(A + B\|\boldsymbol{x}\|).$$

It implies that

$$\int_{t_k}^{t_{k+1}} \|b(t, \boldsymbol{x}) - b(t_k, \boldsymbol{x})\| \mathrm{d}t \leq \frac{h^2}{2}(A + B\|\boldsymbol{x}\|). \tag{58}$$

Therefore, combining (57) and (58),

$$\|\Phi_k(\boldsymbol{x}) - \Psi_k(\boldsymbol{x})\| \le \int_{t_k}^{t_{k+1}} \|b(t, \boldsymbol{x}_t) - b(t, \boldsymbol{x})\|\mathrm{d}t + \int_{t_k}^{t_{k+1}} \|b(t, \boldsymbol{x}) - b(t_k, \boldsymbol{x})\|\mathrm{d}t$$
$$\le \frac{h^2}{2}\left(A + C_2 K(\eta)^2 + CK(\eta) + (B + C_1 K(\eta)^2)\|\boldsymbol{x}\|\right). \tag{59}$$

Combining (54) and (59), by setting $\boldsymbol{x} = \boldsymbol{X}_k^h$ and $\boldsymbol{y} = \tilde{\boldsymbol{X}}_{t_k}$,

$$\|\boldsymbol{e}_{k+1}\| \le \left\|\Phi_k(\tilde{\boldsymbol{X}}_{t_k}) - \Phi_k(\boldsymbol{X}_k^h)\right\| + \left\|\Phi_k(\boldsymbol{X}_k^h) - \Psi_k(\boldsymbol{X}_k^h)\right\|$$
$$\le e^{-(m_I-1)h}\|\boldsymbol{e}_k\| + \frac{h^2}{2}\left(A + C_2 K(\eta)^2 + CK(\eta) + (B + C_1 K(\eta)^2)\|\boldsymbol{X}_k^h\|\right).$$

Let $a_k = \mathbb{E}\left[\|\boldsymbol{e}_k\|\right]$. By the following Lemma D.7, because $h(\eta - 1) < 1$, $\mathbb{E}\left[\|\boldsymbol{X}_k^h\|\right] \le M_e$. Taking the expectation of above inequality, we have

$$a_{k+1} \le e^{-(m_I-1)h}a_k + \frac{h^2}{2}\left(A + C_2 K(\eta)^2 + CK(\eta) + (B + C_1 K(\eta)^2)M_e\right).$$

Therefore, by coupling $\tilde{\boldsymbol{X}}_0 = \boldsymbol{X}_0^h$, i.e., $a_0 = 0$, we have

$$a_N = \mathbb{E}\left[\left\|\tilde{\boldsymbol{X}}_{T-\delta} - \boldsymbol{X}_N^h\right\|\right] \le \frac{h^2}{2}\left(A + C_2 K(\eta)^2 + CK(\eta) + (B + C_1 K(\eta)^2)M_e\right)\sum_{k=0}^{N-1} e^{-(m_I-1)hk}$$
$$\le \frac{h}{2}\left(A + C_2 K(\eta)^2 + CK(\eta) + (B + C_1 K(\eta)^2)M_e\right)\frac{e^{(m_I-1)h}}{m_I - 1}.$$

It follows that as $h \to 0$ and $\eta \to \infty$,

$$\mathcal{W}_1(\tilde{p}_{T-\delta}, \tilde{p}_N^h) \le \mathbb{E}\left[\left\|\tilde{\boldsymbol{X}}_{T-\delta} - \boldsymbol{X}_N^h\right\|\right] \le \mathcal{O}(h\eta^2). \qquad \square$$

**Lemma D.7.** *For $\eta > 2$, if $h(\eta - 1) < 1$, then*

$$\sup_k \mathbb{E}\left[\left\|\boldsymbol{X}_k^h\right\|\right] \le M_e,$$

*where $M_e$ is independent of $\eta$.*

*Proof.* First, by construction,

$$\boldsymbol{X}_{k+1}^h = \left(\boldsymbol{I}_D + h(\boldsymbol{I}_D - \eta P_1)\right)\boldsymbol{X}_k^h + h\nabla_{\boldsymbol{x}}\log p_{T-t_k}^\sigma(\boldsymbol{X}_k^h).$$

Let $M_h = \boldsymbol{I}_D + h(\boldsymbol{I}_D - \eta P_1)$. Then because $P_1$ is an orthogonal projection, there are only two eigenvalues of $M_h$: for $\boldsymbol{x} \in \ker P_1$, $M_h\boldsymbol{x} = (1 + h)\boldsymbol{x}$, and for $\boldsymbol{x} \in \operatorname{Im} P_1$, $M_h\boldsymbol{x} = (1 + h(1 - \eta))\boldsymbol{x}$. Because $h(\eta - 1) < 1$, $1 + h(1 - \eta) \in [0, 1]$. So

$$\|M_h\|_{\mathrm{op}} = 1 + h.$$

Similarly, as shown in (32),

$$\left\|\nabla_{\boldsymbol{x}}\log p_{T-t_k}^\sigma(\boldsymbol{X}_k^h)\right\| \le L_S\left\|\boldsymbol{X}_k^h\right\| + C.$$

Therefore,

$$\left\|\boldsymbol{X}_{k+1}^h\right\| \le \|M_h\|_{\mathrm{op}}\left\|\boldsymbol{X}_k^h\right\| + h\left\|\nabla_{\boldsymbol{x}}\log p_{T-t_k}^\sigma(\boldsymbol{X}_k^h)\right\|$$
$$\le (1 + h(1 + L_S))\left\|\boldsymbol{X}_k^h\right\| + Ch.$$

Taking expectations on the both sides, we have

$$\mathbb{E}\left[\left\|\boldsymbol{X}_k^h\right\|\right] \le (1 + h(1 + L_S))^k \mathbb{E}\left[\left\|\boldsymbol{X}_0^h\right\|\right] + Ch\sum_{j=0}^{k-1}(1 + h(1 + L_S))^j$$

$$\leq e^{(1+L_S)t_k} \left( \mathbb{E}\left[\left\|\boldsymbol{X}_0^h\right\|\right] + Ct_k \right).$$

Because $\boldsymbol{X}_0^h \sim \mathcal{N}(\boldsymbol{0}, \boldsymbol{I}_D)$, $\mathbb{E}\left[\left\|\boldsymbol{X}_0^h\right\|\right] \leq \sqrt{\mathbb{E}\left[\left\|\boldsymbol{X}_0^h\right\|^2\right]} = \sqrt{D}$. Therefore, if let

$$M_e := e^{(1+L_S)(T-\delta)} \left( \sqrt{D} + C(T-\delta) \right),$$

which is independent of $\eta$, then

$$\mathbb{E}\left[\left\|\boldsymbol{X}_k^h\right\|\right] \leq M_e. \qquad \square$$

## E  Analysis for Assumptions

### E.1  More Details for Orthogonality Assumption

For Assumption I, consider the case where $A_1^\top A_2 \neq \boldsymbol{O}$, i.e., $\mathcal{M}_1$ is not orthogonal to $\mathcal{M}_2$. In this case, $A = (A_1, A_2) \notin \mathcal{O}^{D \times d}$, meaning that $A^\top A \neq \boldsymbol{I}_d$ and $AA^\top$ is no longer an orthogonal projection. We claim that this relaxation does not affect our analysis regarding the guidance scale $\eta$. Based on our results, it is necessary to examine its influence from three perspective: the smoothness and concavity of $\log p_t^\sigma$ (Section 5.1), the estimation of the target manifold $\mathcal{M}_1$ (Section 5.2), and the distance between generated and target distributions (Section 5.3).

(a) Smoothness and Convexity: First, the results on the strong log-concavity of the latent density in Theorem 7 are independent of the orthogonality of $A$. Therefore, to analyze the smoothness and concavity of $\log p_t^\sigma$, it suffices to revisit the proof of Theorem 8. Note that

$$\boldsymbol{X}_t^\sigma = \sqrt{\alpha_t} A \boldsymbol{Z}_\sigma + \sqrt{1-\alpha_t}\boldsymbol{\xi} \sim p_t^\sigma.$$

By Proposition 3, Corollary 4, and the $m_0^Z$-strong log-concavity of the latent density $p_\sigma^Z$ (Theorem 7), we obtain the following bounds:

$$\left\|\nabla_{\boldsymbol{x}}^2 \log p_t^\sigma(\boldsymbol{x})\right\|_{\mathrm{op}} \leq L_t^A, \quad L_t^A := \frac{\alpha_t(\Lambda_A + \lambda_A) + (1-\alpha_t)m_0^Z}{(1-\alpha_t)(\alpha_t \lambda_A + m_0^z(1-\alpha_t))},$$

and

$$-\nabla_{\boldsymbol{x}}^2 \log p_t^\sigma(\boldsymbol{x}) \succeq m_t^Z \boldsymbol{I}_D, \quad m_t^Z := \frac{(1-\alpha_t)m_0^z - \alpha_t(\Lambda_A - \lambda_A)}{(1-\alpha_t)(\alpha_t \lambda_A + m_0^z(1-\alpha_t))},$$

where

$$\Lambda_A = \|A\|_{\mathrm{op}}^2 = \lambda_{\max}(A^\top A), \quad \lambda_A = \lambda_{\min}(A^\top A).$$

Because $A = (A_1, A_2)$ and $A_i$ are orthogonal,

$$A^\top A = \begin{pmatrix} \boldsymbol{I}_{d_1} & C \\ C^\top & \boldsymbol{I}_{d_2} \end{pmatrix} = \boldsymbol{I}_d + \begin{pmatrix} \boldsymbol{O} & C \\ C^\top & \boldsymbol{O} \end{pmatrix}, \quad C := A_1^\top A_2.$$

Let $\sigma_{\max}(C)$ be the maximal singular value of $C$. Then, we have

$$1 - \sigma_{\max}(C) \leq \lambda_A \leq \Lambda_A \leq 1 + \sigma_{\max}(C).$$

Moreover, because $\|C\|_{\mathrm{op}} \leq \|A_1\|_{\mathrm{op}}\|A_2\|_{\mathrm{op}} = 1$, $\sigma_{\max}(C) \leq 1$, which implies that $0 \leq \lambda_A \leq \Lambda_A \leq 2$.

For smoothness, it is clear that $0 < L_t^A < \infty$, so the non-orthogonality of $A$ does not affect the $L$-smoothness of $\log p_t^\sigma$, except that the constant changes from $L_t$ to $L_t^A$. However, for strong log-concavity, it requires $m_t^A > 1$ (Corollary 9), which holds if

$$t > \frac{1}{2}\log\frac{m_0^z - \Lambda_A}{m_0^z - \lambda_A}, \tag{60}$$

under the condition $m_0^Z > 2 \geq \Lambda_A$. This requires a modification of Assumption III, $M \leq 2\sqrt{m-2}$, for the same reason discussed in the proof of Corollary 9.

(b) Estimating Target Manifold: Since Theorem 10 depends only on the $L$-smoothness of $\log p_t^\sigma$, and the geometric guidance model $(*)$ does not involve $A$, the result of Theorem 10 remains valid even when $A$ is not orthogonal.

(c) Distance to Target Distribution: Because the condition $m_t^A > 1$ requires inequality (60), one can set

$$\delta > \frac{1}{2} \log \frac{m_0^z - \Lambda_A}{m_0^z - \lambda_A},$$

and consider the geometric guidance model $(*)$ on interval $[0, T - \delta]$. With this adjustment, the results in Theorem 11 still hold, up to changes in certain constants. For instance, the non-orthogonality of $A$ changes the bound on $\mathcal{W}_1((Q_1)\#\mathbb{P}_X, \mathbb{P}_{X|Y}(\cdot \mid Y = 1))$, specifically the constant $\tilde{C}$ in Theorem 11.

### E.2 More details of Assumption III

In the following, we demonstrate a family of distributions that satisfy both Assumption II and Assumption III. Consider the density function $p_i^Z$ of the distribution $\mathbb{P}_i^Z$, given by the form:

$$p_i^Z(\boldsymbol{z}) = e^{-V_i(\boldsymbol{z})} \chi_{K_i}(\boldsymbol{z}),$$

where $K_i \subset \mathbb{R}^{d_i}$ is a convex and compact set, and

$$\nabla_{\boldsymbol{z}}^2 V_i(\boldsymbol{z}) \succeq m \boldsymbol{I}_{d_i}.$$

In other words, $p_i^Z$ belongs to the class of strongly log-concave densities supported on convex and compact subsets of $\mathbb{R}^{d_i}$.

First, for such $p_i^Z$, strong log-concavity on a convex set does not perfectly align with Assumption II, which induces a question of whether this property can substitute for Assumption II in deriving the strong log-concavity of the mixture latent density $p_\sigma^Z$ defined in Equation (11).

In the proof of Theorem 7, the strong log-concavity of $p_\sigma^Z$ is inherited from that of the component densities $p_{i,\sigma}^Z$ defined in Equation (12), which are shown to be strongly log-concave via Corollary 4, under Assumption II. In other words, the key question is whether strong log-concavity on a convex set suffices to replace the strong log-concavity condition in Proposition 3, and thereby still allow us to deduce the conclusion of Corollary 4.

**Proposition E.1.** *Let $\boldsymbol{Z}$ be a random variable on $\mathbb{R}^k$ with the density function $p^Z$ given by*

$$p^Z(\boldsymbol{z}) = e^{-V(\boldsymbol{z})} \chi_K(\boldsymbol{z}),$$

*where $K$ is a convex set. Let $B \in \mathbb{R}^{n \times k}$. Assume there are $m_0, \Lambda > 0$ such that*

$$\nabla_{\boldsymbol{z}}^2 V(\boldsymbol{z}) \succeq m_0 \boldsymbol{I}_k, \quad \|B\|_{\mathrm{op}}^2 \leq \Lambda,$$

*and $\lambda := \lambda_{\min}(B^\top B) \geq 0$. For $\alpha \in \mathbb{R}$ and $\beta > 0$, let*

$$\boldsymbol{X} = \alpha B \boldsymbol{Z} + \beta \boldsymbol{\xi}, \quad \boldsymbol{\xi} \sim \mathcal{N}(\boldsymbol{0}, \boldsymbol{I}_n)$$

*with the density function $p_X$ on $\mathbb{R}^n$. We have*

$$\nabla_{\boldsymbol{x}}^2 \log p_X(\boldsymbol{x}) \preceq \left( \frac{\alpha^2 \Lambda}{\beta^2(\alpha^2 \lambda + m_0 \beta^2)} - \frac{1}{\beta^2} \right) \boldsymbol{I}_n.$$

*Proof.* By the same calculation, we have

$$\nabla_{\boldsymbol{x}}^2 \log p_X(\boldsymbol{x}) = \frac{\alpha^2}{\beta^4} B \operatorname{Cov}_{\mu_x}(\boldsymbol{Z}) B^\top - \frac{1}{\beta^2} \boldsymbol{I}_n,$$

Note in this case,

$$d\mu_x(\boldsymbol{z}) = \frac{e^{-U_x(\boldsymbol{z})}\chi_K(\boldsymbol{z})d\boldsymbol{z}}{\int_K e^{-U}(\boldsymbol{y})d\boldsymbol{y}},$$

where

$$U_x(\boldsymbol{z}) = \frac{1}{2\sigma^2}\|\boldsymbol{x} - B\boldsymbol{z}\|^2 + V(\boldsymbol{z}).$$

It follows that

$$\nabla_{\boldsymbol{z}}^2 U_x(\boldsymbol{z}) = \frac{\alpha^2}{\beta^2}B^\top B + \nabla_{\boldsymbol{z}}^2 V(\boldsymbol{z}) \succeq m\boldsymbol{I}_k, \ m := \frac{\alpha^2\lambda}{\beta^2} + m_0.$$

Instead of applying Lemma H.6, by using the Brascamp–Lieb Inequality on a convex set (Bobkov & Ledoux, 2000, Proposition 2.1), we still have

$$\mathrm{Var}_{\mu_x}(f) \leq \frac{1}{m}\mathbb{E}_{\mu_x}\left[\|\nabla f\|^2\right],$$

for any $C^1$ function $f\colon \mathbb{R}^k \to \mathbb{R}$, which also indicates that

$$\|\mathrm{Cov}_{\mu_x}(\boldsymbol{Z})\|_{\mathrm{op}} \leq \frac{1}{m}.$$

Then, the following proof is as same as the proof in Proposition 3 and in Corollary 4 so that we have the same result

$$\nabla_{\boldsymbol{x}}^2 \log p_X(\boldsymbol{x}) \preceq \left(\frac{\alpha^2\Lambda}{\beta^2(\alpha^2\lambda + m_0\beta^2)} - \frac{1}{\beta^2}\right)\boldsymbol{I}_n. \qquad \square$$

Therefore, Proposition E.1 shows that, in our settings, Assumption II can be replaced Assumption II′:

**Assumption II′.** *For $i = 1, 2$, $\mathbb{P}_i^Z$ admits the density function $p_i^Z$ that has the form $p_i^Z(\boldsymbol{z}) = e^{-V_i(\boldsymbol{z})}\chi_{K_i}(\boldsymbol{z})$ such that $K_i \subset \mathbb{R}^{d_i}$ is a convex and compact set, and*

$$\nabla_{\boldsymbol{z}}^2 V_i(\boldsymbol{z}) \succeq m\boldsymbol{I}_{d_i}.$$

Next, we verify if $p_i^Z$ in such class can satisfy Assumption III.

**Proposition E.2.** *For $i = 1, 2$, let $p_i^Z$ satisfy Assumption II′, and let $p_{i,\sigma}^Z$ defined by Equation (12). Fix a $\sigma > 0$, we have*

$$\sup_{\boldsymbol{x}}\|\nabla_{\boldsymbol{x}}\log p_{1,\sigma}^Z(\boldsymbol{x}) - \nabla_{\boldsymbol{x}}\log p_{2,\sigma}^Z(\boldsymbol{x})\| \leq \frac{\sqrt{|K_1|^2 + |K_2|^2}}{\sigma^2},$$

*where $|K_i| = \sup\{\|\boldsymbol{z}\|\colon \boldsymbol{z} \in K_i\}$.*

*Proof.* First, by the definition (12),

$$p_{1,\sigma}^Z(\boldsymbol{z}) = (2\pi\sigma^2)^{-\frac{d}{2}}\int_{K_1}\exp\left(-\frac{1}{2\sigma^2}\|\boldsymbol{z} - (\boldsymbol{z}_1, 0)^\top\|^2\right)p_1^Z(\boldsymbol{z}_1)d\boldsymbol{z}_1,$$

Therefore,

$$\nabla_{\boldsymbol{z}}\log p_{1,\sigma}^Z(\boldsymbol{z}) = \frac{\nabla_{\boldsymbol{z}}p_{1,\sigma}^Z(\boldsymbol{z})}{p_{1,\sigma}^Z(\boldsymbol{z})} = \frac{-\frac{1}{\sigma^2}\int_{K_1}(\boldsymbol{z} - (\boldsymbol{z}_1, 0)^\top)\exp\left(-\frac{1}{2\sigma^2}\|\boldsymbol{z} - (\boldsymbol{z}_1, 0)^\top\|^2\right)p_1^Z(\boldsymbol{z}_1)d\boldsymbol{z}_1}{\int_{K_1}\exp\left(-\frac{1}{2\sigma^2}\|\boldsymbol{z} - (\boldsymbol{z}_1, 0)^\top\|^2\right)p_1^Z(\boldsymbol{z}_1)d\boldsymbol{z}_1}.$$

It follows that

$$\nabla_{\boldsymbol{z}}\log p_{1,\sigma}^Z(\boldsymbol{z}) = \frac{1}{\sigma^2}\left((m_1(\boldsymbol{z}), 0)^\top - \boldsymbol{z}\right),$$

where

$$m_1(\boldsymbol{z}) = \frac{\int_{K_1}\boldsymbol{z}_1\exp\left(-\frac{1}{2\sigma^2}\|\boldsymbol{z} - (\boldsymbol{z}_1, 0)^\top\|^2\right)p_1^Z(\boldsymbol{z}_1)d\boldsymbol{z}_1}{\int_{K_1}\exp\left(-\frac{1}{2\sigma^2}\|\boldsymbol{z} - (\boldsymbol{z}_1, 0)^\top\|^2\right)p_1^Z(\boldsymbol{z}_1)d\boldsymbol{z}_1}.$$

Similarly,

$$\nabla_{\boldsymbol{z}} \log p_{2,\sigma}^Z(\boldsymbol{z}) = \frac{1}{\sigma^2}\left((0, m_2(\boldsymbol{z}))^\top - \boldsymbol{z}\right),$$

for

$$m_2(\boldsymbol{z}) = \frac{\int_{K_2} \boldsymbol{z}_2 \exp\left(-\frac{1}{2\sigma^2}\left\|\boldsymbol{z} - (\boldsymbol{z}_2, 0)^\top\right\|^2\right) p_2^Z(\boldsymbol{z}_2)\mathrm{d}\boldsymbol{z}_2}{\int_{K_2} \exp\left(-\frac{1}{2\sigma^2}\left\|\boldsymbol{z} - (\boldsymbol{z}_2, 0)^\top\right\|^2\right) p_2^Z(\boldsymbol{z}_2)\mathrm{d}\boldsymbol{z}_2}.$$

Therefore,

$$\left\|\nabla_{\boldsymbol{z}} \log p_{1,\sigma}^Z(\boldsymbol{z}) - \nabla_{\boldsymbol{z}} \log p_{2,\sigma}^Z(\boldsymbol{z})\right\| = \frac{1}{\sigma^2}\left\|(m_1(\boldsymbol{z}), m_2(\boldsymbol{z}))^\top\right\| = \frac{1}{\sigma^2}\sqrt{\|m_1(\boldsymbol{z})\|^2 + \|m_2(\boldsymbol{z})\|^2}.$$

Note that

$$m_i(\boldsymbol{z}) = \mathbb{E}_{\boldsymbol{Z} \sim \mu_z^i}[\boldsymbol{Z}], \quad \mathrm{d}\mu_z^i(\boldsymbol{z}_i) := \frac{\exp\left(-\frac{1}{2\sigma^2}\left\|\boldsymbol{z} - (\boldsymbol{z}_i, 0)^\top\right\|^2\right) p_i^Z(\boldsymbol{z}_i)\mathrm{d}\boldsymbol{z}_i}{\int_{K_i} \exp\left(-\frac{1}{2\sigma^2}\left\|\boldsymbol{z} - (\boldsymbol{z}_i, 0)^\top\right\|^2\right) p_i^Z(\boldsymbol{z}_i)\mathrm{d}\boldsymbol{z}_i}.$$

By the convexity of $K_i$ and Lemma H.10, $m_i(\boldsymbol{z}) \in K_i$. Then, the boundedness of $K_i$ implies

$$\sup_{\boldsymbol{x}}\left\|\nabla_{\boldsymbol{x}} \log p_{1,\sigma}^Z(\boldsymbol{x}) - \nabla_{\boldsymbol{x}} \log p_{2,\sigma}^Z(\boldsymbol{x})\right\| \leq \frac{\sqrt{|K_1|^2 + |K_2|^2}}{\sigma^2}. \qquad \square$$

**Sufficient conditions for Assumption III.** If $p_i^Z$ belongs to the class of distributions

$$\left\{e^{-V(\boldsymbol{z})}\chi_K(\boldsymbol{z}) \colon \nabla^2 V \succeq m\boldsymbol{I}, \ K \text{ is compact and convex.}\right\}, \tag{61}$$

then $p_{i,\sigma}^Z$ given Equation (12) is strongly log-concave by Proposition E.1. Moreover, if we choose $\sigma$ such that

$$M \leq \frac{\sqrt{|K_1|^2 + |K_2|^2}}{\sigma^2} \leq 2\sqrt{m-1} \quad \Leftrightarrow \quad \sigma^2 \geq \sqrt{\frac{|K_1|^2 + |K_2|^2}{4(m-1)}}, \tag{62}$$

Proposition E.2 shows that Assumption III is satisfied. Then the mixture latent distribution $p_\sigma^Z$ given by Equation (11) is $m_0^z$-strongly log-concave provided by Theorem 7, which further implies that $p_t^\sigma$ in the geometric guidance model ($*$) satisfies:

$$\left\|\nabla_{\boldsymbol{x}}^2 \log p_t^\sigma(\boldsymbol{x})\right\|_{\mathrm{op}} \leq L_t, \quad -\nabla_{\boldsymbol{x}}^2 \log p_t^\sigma(\boldsymbol{x}) \succeq m_t \boldsymbol{I}_D.$$

# F Lipschitz Continuity of Score Function

If we only focus on the Lipschitz continuity of the score function $\nabla_{\boldsymbol{x}} \log p_t$, where $p_t$ is obtained by a DDPM initialized from a distribution whose latent distribution admits a smooth density function $p^Z$, then the conditions in Proposition 3 can be relaxed. We consider two cases below.

The first case aligns with the setting considered in De Bortoli (2022), where $\mathrm{supp}\, p^Z$ is assumed to be compact. We provide an alternative proof for this case, motivated by the argument used in the proof of Proposition 3.

**Proposition F.1.** *Let $\boldsymbol{Z}$ be a random variable on $\mathbb{R}^k$ with the density function $p^Z$, and let $\phi \colon \mathbb{R}^k \to \mathbb{R}^n$ be continuous. Assume $\mathrm{supp}\, p^Z$ is compact. For $\alpha \in \mathbb{R}$ and $\beta > 0$, let*

$$\boldsymbol{X} = \alpha\phi(\boldsymbol{Z}) + \beta\boldsymbol{\xi}, \quad \boldsymbol{\xi} \sim \mathcal{N}(\boldsymbol{0}, \boldsymbol{I}_n),$$

*with the density function $p_X$ on $\mathbb{R}^n$. We have*

$$\left\|\nabla_{\boldsymbol{x}}^2 \log p_X(\boldsymbol{x})\right\|_{\mathrm{op}} \leq \frac{1}{\beta^2} + \frac{\alpha^2 R^2}{\beta^4},$$

*for some constant $R > 0$.*

*Proof.* By the similar calculation as in the proof of Proposition 3,

$$\nabla_{\boldsymbol{x}}^2 \log p_X(\boldsymbol{x}) = \frac{\alpha^2}{\beta^4} \operatorname{Cov}_{\mu_x}(\phi(\boldsymbol{Z})) - \frac{1}{\beta^2} \boldsymbol{I}_n,$$

which follows that

$$\left\| \nabla_{\boldsymbol{x}}^2 \log p_X(\boldsymbol{x}) \right\|_{\mathrm{op}} \leq \frac{1}{\beta^2} + \frac{\alpha^2}{\beta^4} \| \operatorname{Cov}_{\mu_x}(\phi(\boldsymbol{Z})) \|_{\mathrm{op}}.$$

To bound the second term, first by the definition of $\mu_x$, $\operatorname{supp} \mu_x = \operatorname{supp} p^Z$. Because $\operatorname{supp} p^Z$ is compact and $\phi$ is continuous, $\phi(\operatorname{supp} \mu_x)$ is compact, which means that there exists a $R > 0$ such that

$$\sup \{ \| \phi(\boldsymbol{z}) \| : \boldsymbol{z} \in \operatorname{supp} \mu_x \} \leq R.$$

Then, we obtain that for any $\boldsymbol{u} \in \mathbb{R}^n$,

$$\boldsymbol{u}^\top \operatorname{Cov}_{\mu_x}(\phi(\boldsymbol{Z}))\boldsymbol{u} = \operatorname{Var}_{\mu_x} \left( \boldsymbol{u}^\top \phi(\boldsymbol{Z}) \right) \leq \operatorname{Var}_{\mu_x} \left( \| \boldsymbol{u} \| \| \phi(\boldsymbol{Z}) \| \right) \leq R^2 \| \boldsymbol{u} \|^2,$$

which indicates that

$$\| \operatorname{Cov}_{\mu_x}(\phi(\boldsymbol{Z})) \|_{\mathrm{op}} \leq R^2.$$

Therefore, we have

$$\left\| \nabla_{\boldsymbol{x}}^2 \log p_X(\boldsymbol{x}) \right\|_{\mathrm{op}} \leq \frac{1}{\beta^2} + \frac{\alpha^2 R^2}{\beta^4}. \qquad \square$$

*Remark* F.1. This proposition shows that when the latent density function has compact support, no additional conditions—such as log-concavity or $L$-smoothness—are required for the latent distribution. Moreover, under the compactness assumption, the results of Proposition 3 can be extended to the nonlinear case, as shown in De Bortoli (2022).

Next, in the non-compact case, Proposition 3 requires the strong log-concavity of the latent density $p^Z$, as it is used to establish not only the $L$-smoothness but also the concavity of $\log p_X$ (see Corollary 4). However, if we are only interested in the $L$-Lipschitz continuity of the score function, the assumption of concavity can be relaxed to the $L_0$-smoothness of $\log p^Z$, i.e., $\left\| \nabla_{\boldsymbol{z}}^2 \log p^Z(\boldsymbol{z}) \right\| \leq L_0$, or even to the weaker condition $\nabla_{\boldsymbol{z}}^2 \log p^Z(\boldsymbol{z}) \preceq L_0 \boldsymbol{I}_k$; see Proposition F.2 below.

**Proposition F.2.** *Let $\boldsymbol{Z}$ be a random variable on $\mathbb{R}^k$ with the density function $p^Z$ and $B \in \mathbb{R}^{n \times k}$. Assume there are $L_0, \Lambda > 0$ such that*

$$\nabla_{\boldsymbol{z}}^2 \log p^Z(\boldsymbol{z}) \preceq L_0 \boldsymbol{I}_k, \quad \|B\|_{\mathrm{op}}^2 \leq \Lambda,$$

*and $\lambda := \lambda_{\min}(B^\top B) > 0$, the minimum of all eigenvalues of $B^\top B$. For $\alpha \in \mathbb{R}$ and $\beta > 0$, let*

$$\boldsymbol{X} = \alpha B \boldsymbol{Z} + \beta \boldsymbol{\xi}, \quad \boldsymbol{\xi} \sim \mathcal{N}(\boldsymbol{0}, \boldsymbol{I}_n),$$

*with the density function $p_X$ on $\mathbb{R}^n$. If $\alpha^2 \lambda - L_0 \beta^2 > 0$, we have*

$$\left\| \nabla_{\boldsymbol{x}}^2 \log p_X(\boldsymbol{x}) \right\|_{\mathrm{op}} \leq \frac{1}{\beta^2} + \frac{\alpha^2 \Lambda}{\beta^2 (\alpha^2 \lambda - L_0 \beta^2)}$$

*and*

$$\nabla_{\boldsymbol{x}}^2 \log p_X(\boldsymbol{x}) \preceq \left( \frac{\alpha^2 \Lambda}{\beta^2 (\alpha^2 \lambda - L_0 \beta^2)} - \frac{1}{\beta^2} \right) \boldsymbol{I}_n.$$

*Proof.* The main difference of this proof to the proof in Proposition 3 is how to bound $\| \operatorname{Cov}_{\mu_x}(\boldsymbol{Z}) \|_{\mathrm{op}}$.

Note that

$$\nabla_{\boldsymbol{z}}^2 U_x(\boldsymbol{z}) = \frac{\alpha^2}{\beta^2} B^\top B + \nabla_{\boldsymbol{z}}^2 V(\boldsymbol{z}) \succeq \left( \frac{\alpha^2 \lambda}{\beta^2} - L_0 \right) I_k,$$

because $-\nabla_{\boldsymbol{z}}^2 \log p^Z(\boldsymbol{z}) = \nabla_{\boldsymbol{z}}^2 V(\boldsymbol{z}) \succeq -L_0 I_k$. When $\alpha^2 \lambda - L_0 \beta^2 > 0$, we similarly obtain

$$\| \operatorname{Cov}_{\mu_x}(\boldsymbol{Z}) \|_{\mathrm{op}} \leq \frac{\beta^2}{\alpha^2 \lambda - L_0 \beta^2}.$$

Therefore,

$$\left\|\nabla_{\boldsymbol{x}}^2 \log p_X(\boldsymbol{x})\right\|_{\mathrm{op}} \leq \frac{1}{\beta^2} + \frac{\alpha^2 \Lambda}{\beta^2(\alpha^2\lambda - L_0\beta^2)}.$$

On the other hand,

$$B \operatorname{Cov}_{\mu_x}(\boldsymbol{Z})B^\top \preceq \frac{\Lambda\beta^2}{\alpha^2\lambda - L_0\beta^2}\boldsymbol{I}_n,$$

which follows that

$$\nabla_{\boldsymbol{x}}^2 \log p_X(\boldsymbol{x}) \preceq \left(\frac{\alpha^2\Lambda}{\beta^2(\alpha^2\lambda - L_0\beta^2)} - \frac{1}{\beta^2}\right)\boldsymbol{I}_n. \qquad \square$$

## G  More Details for Nonlinear Extension

### G.1  Omitted Proofs in Section 6

*Proof of Lemma 12.* The proof consists of the following three steps. First, let $\boldsymbol{z} \in \mathbb{R}^d$ be arbitrary.

(i) Local construction: Because $\phi\colon \mathbb{R}^d \to \mathcal{M} \subset \mathbb{R}^D$ is an isometry, the columns of $J\phi(\boldsymbol{z})$ form an orthonormal basis for the tangent space $T_{\phi(\boldsymbol{z})}\mathcal{M}$. These vectors can be extended to an orthonormal basis of $\mathbb{R}^D$ by adjoining

$$\{n_1(\boldsymbol{z}), n_2(\boldsymbol{z}), \ldots, n_{D-d}(\boldsymbol{z})\},$$

where each $n_i$ is a smooth normal vector fields along $\mathcal{M}$. For such $n_i$, one can define the Fermi coordinates map as

$$F\colon \mathbb{R}^d \times \mathbb{R}^{D-d} \longrightarrow \mathbb{R}^D, \quad F(\boldsymbol{z}, \boldsymbol{v}) = \phi(\boldsymbol{z}) + \sum_{i=1}^{D-d} v_i n_i(\boldsymbol{z}).$$

Then, by the Tubular Neighborhood Theorem (Theorem I.3), there exists a $\varepsilon\colon \mathbb{R}^d \to (0, \infty)$ such that for the set

$$V = \left\{(\boldsymbol{z}, \boldsymbol{v}) \in \mathcal{M} \times \mathbb{R}^{D-d}\colon \|\boldsymbol{v}\| < \varepsilon(\boldsymbol{z})\right\},$$

$F\colon V \to U = F(V)$ is a diffeomorphism, where the open set $U \subset \mathbb{R}^D$ is a neighborhood of $\mathcal{M}$, called a tubular neighborhood. Let $\pi\colon \mathbb{R}^d \times \mathbb{R}^{D-d} \to \mathbb{R}^d$ be the projection, i.e., $\pi(\boldsymbol{z}, \boldsymbol{v}) = \boldsymbol{z}$. Then, one can construct

$$\tilde{\phi}^*\colon U \longrightarrow \mathbb{R}^d, \quad \tilde{\phi}^*(\boldsymbol{x}) = \pi(F^{-1}(\boldsymbol{x}))$$

(ii) Check conditions: First, because $\mathcal{M} \subset U$, and $F$ is diffeomorphic from $V$ to $U$ with $F(\boldsymbol{z}, 0) = \phi(\boldsymbol{z})$,

$$\tilde{\phi}^*(\phi(\boldsymbol{z})) = \pi(F^{-1}(\phi(\boldsymbol{z}))) = \pi(\boldsymbol{z}, 0) = \boldsymbol{z}, \quad \forall\, \boldsymbol{z} \in \mathbb{R}^d.$$

For the derivative condition, by the definition of $F$, we have

$$JF(\boldsymbol{z}, 0) = (J_{\boldsymbol{z}}F(\boldsymbol{z}, 0), J_{\boldsymbol{v}}F(\boldsymbol{z}, 0)) = (J\phi(\boldsymbol{z}), \boldsymbol{n}(\boldsymbol{z})),$$

where $\boldsymbol{n} = (n_1(\boldsymbol{z}), \ldots, n_{D-d}(\boldsymbol{z}))$. By $J\phi^\top J\phi = \boldsymbol{I}_d$, $JF(\boldsymbol{z}, 0)$ is orthogonal, which follows that

$$J(F^{-1})(F(\boldsymbol{z}, 0)) = JF(\boldsymbol{z}, 0)^{-1} = JF(\boldsymbol{z}, 0)^\top = \begin{pmatrix} J\phi(\boldsymbol{z})^\top \\ \boldsymbol{n}(\boldsymbol{z})^\top \end{pmatrix}.$$

On the other hand, $F^{-1}$ can be written as $F^{-1}(\boldsymbol{x}) = (F_1(\boldsymbol{x}), F_2(\boldsymbol{x}))$, where $F_1 = \pi \circ F^{-1} = \tilde{\phi}^*$ on $U$. It implies that

$$J(F^{-1})(F(\boldsymbol{z}, 0)) = \begin{pmatrix} J\tilde{\phi}^*(\phi(\boldsymbol{z})) \\ JF_2(\phi(\boldsymbol{z})) \end{pmatrix}.$$

Therefore, $J\tilde{\phi}^*(\phi(\boldsymbol{z})) = J\phi(\boldsymbol{z})^\top$.

(iii) Global construction: By the Urysohn Lemma (Munkres, 2018), there exists a smooth function $\chi\colon \mathbb{R}^D \to [0,1]$ such that $\chi|_{\tilde{U}} \equiv 1$ and $\chi|_{\mathbb{R}^D \setminus U} \equiv 0$, where $\tilde{U} \subset U$ is a open neighborhood of $\mathcal{M}$. Let $h\colon \mathbb{R}^D \to \mathbb{R}^d$ be any smooth function—for instance, a constant function $h \equiv \boldsymbol{c}$. Define

$$\phi^*(\boldsymbol{x}) = \chi(\boldsymbol{x})\tilde{\phi}^*(\boldsymbol{x}) + (1 - \chi(\boldsymbol{x}))h(\boldsymbol{x}),$$

then the desired identities hold:

$$\phi^* \circ \phi = \mathrm{id}_{\mathbb{R}^d}, \quad J\phi^*(\phi(\boldsymbol{z})) = J\phi(\boldsymbol{z})^\top. \qquad \square$$

*Proof of Theorem 13.* Let $\phi\colon \mathbb{R}^d \to \mathbb{R}^D$ be the isometry for defining $\mathcal{M} = \mathrm{Im}\,\phi$. Because $\mathrm{supp}\,\mathbb{P}_X \subset \mathcal{M}$, there exists a $\mathbb{P}^Z$ defined on $\mathbb{R}^d$ such that $\boldsymbol{X} = \phi(\boldsymbol{Z}) \sim \mathbb{P}_X$ when $\boldsymbol{Z} \sim \mathbb{P}_Z$. Let $t$ be fixed in $(0, T]$. By (2),

$$\boldsymbol{X}_t = \sqrt{\alpha_t}\phi(\boldsymbol{Z}) + \sqrt{1 - \alpha_t}\boldsymbol{\xi}.$$

Define $F^t\colon \mathbb{R}^D \to \mathbb{R}^D$ by

$$F^t(\boldsymbol{x}) := \sqrt{\alpha_t}\phi \circ \phi^* \left(\frac{\boldsymbol{x}}{\sqrt{\alpha_t}}\right), \quad \alpha_t = e^{-2t},$$

where $\phi^*$ is defined in Lemma 12. Then we have

$$F^t(\boldsymbol{X}_t) = \sqrt{\alpha_t}\phi \circ \phi^* \left(\boldsymbol{X}_0 + \sqrt{\frac{1 - \alpha_t}{\alpha_t}}\boldsymbol{\xi}\right), \quad \boldsymbol{X}_0 := \phi(\boldsymbol{Z}).$$

Now consider the Taylor expansion of $\varphi := \phi \circ \phi^*$ at $\boldsymbol{X}_0 = \phi(\boldsymbol{Z})$, with integral remainder. We obtain

$$F^t(\boldsymbol{X}_t) = F^t(\boldsymbol{X}_0) + \sqrt{1 - \alpha_t}J\varphi(\boldsymbol{X}_0)\boldsymbol{\xi} + R(\boldsymbol{\xi}),$$

where $R(\boldsymbol{\xi})$ denotes the remainder term.

Next, we analyze the three terms on the right-hand side one by one. For the first term, because $\boldsymbol{X}_0 = \phi(\boldsymbol{Z}) \in \mathcal{M}$, $\boldsymbol{Z} = \phi^*(\boldsymbol{X}_0)$ by the definition of $\phi^*$; see the proof of Lemma 12. It implies that

$$F^t(\boldsymbol{X}_0) = \sqrt{\alpha_t}\phi \circ \phi^*(\boldsymbol{X}_0) = \sqrt{\alpha_t}\boldsymbol{X}_0.$$

For the second term, by Lemma 12,

$$J\varphi(\boldsymbol{X}_0) = J\phi(\boldsymbol{Z})J\phi^*(\phi(\boldsymbol{Z})) = J\phi(\boldsymbol{Z})J\phi(\boldsymbol{Z})^\top.$$

Moreover, because $J\phi^\top J\phi = \boldsymbol{I}_d$, $P := J\varphi(\boldsymbol{X}_0)$ is an orthogonal projection with rank $d$. For the third term,

$$R(\boldsymbol{\xi}) = \frac{1 - \alpha_t}{\sqrt{\alpha_t}} \int_0^1 (1 - s)D^2\varphi\left(\boldsymbol{X}_0 + s\sqrt{(1 - \alpha_t)/\alpha_t}\boldsymbol{\xi}\right)[\boldsymbol{\xi}, \boldsymbol{\xi}]\mathrm{d}s.$$

By the proof of Lemma 12, $\phi^* \equiv \boldsymbol{c}$ on $\mathbb{R}^D \setminus U$ for a tubular neighborhood $U$ of $\mathcal{M}$, which means $J\phi^* = 0$ and $D^2\phi^* = 0$ on $\mathbb{R}^D \setminus U$. It follows that

$$D^2\varphi(\boldsymbol{x})[\boldsymbol{u}, \boldsymbol{v}] = D^2\phi(\phi^*(\boldsymbol{x}))\left[J\phi^*(\boldsymbol{x})\boldsymbol{u}, J\phi^*(\boldsymbol{x})\boldsymbol{v}\right] + J\phi(\phi^*(\boldsymbol{x}))(D^2\phi^*(\boldsymbol{x})[\boldsymbol{u}, \boldsymbol{v}]) = 0$$

for $\boldsymbol{x} \in \mathbb{R}^D \setminus U$. For a chosen $\delta$, we can choose a tubular neighborhood $U$ sufficiently thin such that $\boldsymbol{X}_0 + s\sqrt{(1 - \alpha_t)/\alpha_t}\boldsymbol{\xi} \notin U$ for $s > \delta$. Therefore, we have

$$R(\boldsymbol{\xi}) = \frac{1 - \alpha_t}{\sqrt{\alpha_t}} \int_0^\delta (1 - s)D^2\varphi\left(\boldsymbol{X}_0 + s\sqrt{(1 - \alpha_t)/\alpha_t}\boldsymbol{\xi}\right)[\boldsymbol{\xi}, \boldsymbol{\xi}]\mathrm{d}s.$$

Assume $D^2\varphi$ is bounded on $U$. Then, for any small $\varepsilon' > 0$, one can choose $\delta$ sufficiently small such that $\|R(\boldsymbol{\xi})\| \le \varepsilon'$.

Combining these analyses, we obtain

$$\sqrt{1-\alpha_t}\|(\boldsymbol{I}_D - P)\boldsymbol{\xi}\| - \varepsilon' \leq \|\boldsymbol{X}_t - F^t(\boldsymbol{X}_t)\| \leq \sqrt{1-\alpha_t}\|(\boldsymbol{I}_D - P)\boldsymbol{\xi}\| + \varepsilon'. \tag{63}$$

Let $f^t(\boldsymbol{x}) := \|\boldsymbol{x} - F^t(\boldsymbol{x})\|$. Similarly as the proof of Proposition 1, by the Laurent-Massart bound (Lemma H.1), (63) implies that

$$\mathbb{P}\left(r(t)\sqrt{1-2\sqrt{\varepsilon}} - \varepsilon' \leq f^t(\boldsymbol{X}_t) \leq r(t)\sqrt{1+2\sqrt{\varepsilon}+2\varepsilon} + \varepsilon'\right) \geq 1 - 2e^{-2(D-d)\varepsilon},$$

where $r(t) = \sqrt{(D-d)(1-\alpha_t)}$. Because $d \ll D$, one can choose small $\varepsilon$ such that $\delta = e^{-2(D-d)\varepsilon}$ is also small enough. As a result, $\mathbb{P}(f^t(\boldsymbol{X}_t) \approx r(t)) \geq 1 - \delta$, i.e., $\boldsymbol{X}_t$ concentrates on $\mathcal{M}^t = (f^t)^{-1}(r(t))$ with high probability. $\qquad\square$

### G.2 More Results of Experiments

**Comparison of FID.** Table 2 serves as a complement to Table 1.

Table 2: Comparison of FID on CIFAR-10

|  | Airplane | Bird | Cat | Deer | Dog | Overall |
|---|---|---|---|---|---|---|
| CGM ($\eta = 1$) | 17.95 | 21.69 | 20.34 | 19.24 | 23.62 | 4.07 |
| GeGM ($\eta = 50$) | 18.98 | 18.39 | 17.35 | 17.38 | 18.45 | 5.15 |

**FID v.s. guidance scale on CIFAR-10.** By sampling with the nonlinear GeGM (16), Figure 3 shows how the FID varies with the guidance scale $\eta$ across all classes from CIFAR-10, which is consistent with the result of Theorem 11.

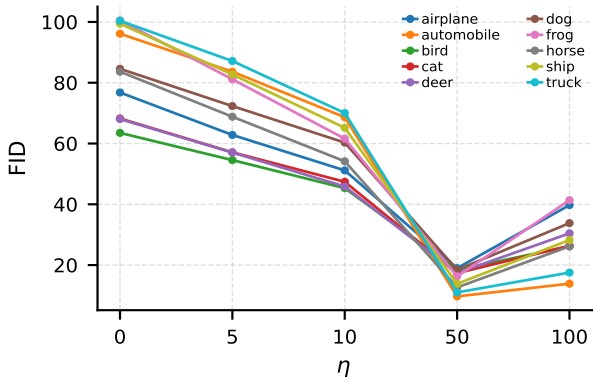

Figure 3: FID v.s. guidance scale $\eta$ of GeGM on all classes of CIFAR-10

## H  Technical Lemmas

**Lemma H.1** (Laurent & Massart (2000))**.** *Let $X$ be a $\chi^2$-random variable with $n$ degrees of freedom, i.e., $X = \sum_{i=1}^n \xi_i^2$ with $\xi_i \overset{i.i.d.}{\sim} \mathcal{N}(0,1)$. Then, for any $\alpha > 0$, we have*

$$\mathbb{P}(X - n \geq 2\sqrt{n\alpha} + 2\alpha) \leq e^{-\alpha},$$
$$\mathbb{P}(X - n \leq -2\sqrt{n\alpha}) \leq e^{-\alpha}.$$

**Lemma H.2.** *Let $\mathbb{R}^n = V \oplus V^\perp$ be an orthogonal decomposition of $\mathbb{R}^n$, where $V$ is a linear subspace and $V^\perp$ is its orthogonal complement. Let $\boldsymbol{X}, \boldsymbol{Y} \in \mathbb{R}^n$ be random variables such that $\boldsymbol{X} \in V$, $\boldsymbol{Y} \in V^\perp$, and $\boldsymbol{X}$ independent of $\boldsymbol{Y}$. Suppose that $\boldsymbol{X}$ and $\boldsymbol{Y}$ admit densities $p_X$ and $p_Y$ on $V$ and $V^\perp$, respectively, with respect to the canonical volume measures on $V$ and $V^\perp$. Then the density function of $\boldsymbol{Z} = \boldsymbol{X} + \boldsymbol{Y}$ is given by*

$$p_Z(\boldsymbol{z}) = p_X(Q\boldsymbol{x})p_Y(Q^\perp\boldsymbol{x}),$$

*where $Q$ is the orthogonal projection onto $V$, and $Q^\perp = \boldsymbol{I}_n - Q$ is the orthogonal projection onto $V^\perp$.*

*Proof.* Let $m_V$ and $m_{V^\perp}$ be the canonical volume measure on $V$ and $V^\perp$, respectively. Define $\Phi \colon V \times V^\perp \to \mathbb{R}^n$ by $\phi(\boldsymbol{x}, \boldsymbol{y}) = \boldsymbol{x} + \boldsymbol{y}$. Clearly, $\Phi$ is an orthogonal linear map, which indicates $|\det J\Phi| = 1$, so

$$\Phi_\# (m_V \otimes m_{V^\perp}) = m_n,$$

where $m_n$ is the Lebesgue measure on $\mathbb{R}^n$.

Let $\mathbb{P}_X$ and $\mathbb{P}_Y$ be the distributions of $\boldsymbol{X}$ and $\boldsymbol{Y}$, respectively. Then $\mathrm{d}\mathbb{P}_X = p_X \mathrm{d}m_V$ and $\mathrm{d}\mathbb{P}_Y = p_Y \mathrm{d}m_{V^\perp}$. By the independence of $\boldsymbol{X}$ and $\boldsymbol{Y}$, we have

$$\mathrm{d} (\mathbb{P}_X \otimes \mathbb{P}_Y) = p_X(\boldsymbol{x})p_Y(\boldsymbol{y})\mathrm{d} (m_V(\boldsymbol{x}) \otimes m_{V^\perp}(\boldsymbol{y})) .$$

Since $\boldsymbol{Z} = \boldsymbol{X} + \boldsymbol{Y} = \Phi(\boldsymbol{X}, \boldsymbol{Y})$, it follows that $\boldsymbol{Z} \sim \mathbb{P}_Z = \Phi_\# (\mathbb{P}_X \otimes \mathbb{P}_Y)$, and thus

$$\begin{aligned}
\mathbb{P}_Z(U) &= \int_{\mathbb{R}^n} \chi_U(\boldsymbol{z})\mathrm{d}\mathbb{P}_Z(\boldsymbol{z}) \\
&= \int_{V \times V^\perp} \chi_U(\boldsymbol{x} + \boldsymbol{y})\mathrm{d} (\mathbb{P}_X \otimes \mathbb{P}_Y) \\
&= \int_{V \times V^\perp} \chi_U(\boldsymbol{x} + \boldsymbol{y})p_X(\boldsymbol{x})p_Y(\boldsymbol{y})\mathrm{d} (m_V(\boldsymbol{x}) \otimes m_{V^\perp}(\boldsymbol{y})) \\
&= \int_{\mathbb{R}^n} \chi_U(\boldsymbol{z})p_X(Q\boldsymbol{z})p_Y(Q^\perp\boldsymbol{z})\mathrm{d}\Phi_\# (m_V(\boldsymbol{x}) \otimes m_{V^\perp}(\boldsymbol{y})) \\
&= \int_{\mathbb{R}^n} \chi_U(\boldsymbol{z})p_X(Q\boldsymbol{z})p_Y(Q^\perp\boldsymbol{z})\mathrm{d}m_n(\boldsymbol{z}).
\end{aligned}$$

Therefore, we have

$$p_Z(\boldsymbol{z}) = p_X(Q\boldsymbol{x})p_Y(Q^\perp\boldsymbol{x}). \qquad \square$$

**Lemma H.3.** *Let $(\boldsymbol{W}_t)_{t \geq 0}$ be a standard Brownian motion on $\mathbb{R}^m$ and $A \in \mathcal{O}^{m \times n}$. Let*

$$\boldsymbol{B}_t := A^\top \boldsymbol{W}_t.$$

*Then $(\boldsymbol{B}_t)_{t \geq 0}$ is a standard Brownian motion on $\mathbb{R}^n$.*

*Proof.* The path continuity of $t \mapsto \boldsymbol{B}_t = A^\top \boldsymbol{W}_t$ follows directly from the path continuity of $t \mapsto \boldsymbol{W}_t$, as does the independence of increments. The initial condition $\boldsymbol{B}_0 = A^\top \boldsymbol{W}_0 = 0$ is immediate. Moreover, since $A \in \mathcal{O}^{m \times n}$, we have,

$$\boldsymbol{B}_t - \boldsymbol{B}_s = A^\top (\boldsymbol{B}_t - \boldsymbol{B}_s) \sim \mathcal{N}(0, (t - s)\boldsymbol{I}_m), \quad \forall\, t > s. \qquad \square$$

**Lemma H.4** (Jost (2008)). *For a function $g \colon \mathbb{R}^n \to \mathbb{R}^m$, if $g \colon \mathbb{R}^n \to \operatorname{Im} g$ is a diffeomorphism, that is, both $g$ and its inverse $g^{-1} \colon \operatorname{Im} g \to \mathbb{R}^n$ are continuously differentiable, then $g_\# p_X$, the density function of $g_\# \mathbb{P}_X$ on $\operatorname{Im} g$ with respect to the canonical volume measure on $\operatorname{Im} g$, satisfies*

$$g_\# p_X(\boldsymbol{y}) = p_X(\boldsymbol{x}) \left|\det \left(Jg(\boldsymbol{x})Jg(\boldsymbol{x})^\top\right)\right|^{\frac{1}{2}}, \quad \boldsymbol{x} = g^{-1}(\boldsymbol{y}).$$

*Moreover, when $g(\boldsymbol{x}) = A\boldsymbol{x}$ for an $A \in \mathcal{O}^{m \times n}$, $A_\# p_X(\boldsymbol{y}) = p_X(A^\top\boldsymbol{y})$.*

*Remark* H.1. This result is essentially a general form of the change-of-variables formula, which has been widely used in the context of generative models on manifolds (see, e.g., Loaiza-Ganem et al. (2024)). To rigorously justify this result, some basic knowledge of Riemannian geometry is required. Since $g \colon \mathbb{R}^n \to \operatorname{Im} g$ is a diffeomorphism, the image $\operatorname{Im} g \subset \mathbb{R}^n$ is a submanifold. When $\operatorname{Im} g$ is equipped with the canonical Riemannian structure induced from the ambient Euclidean space $\mathbb{R}^n$, the canonical volume measure on $\operatorname{Im} g$ coincides with the Riemannian volume measure. Therefore, the relevant results from Jost (2008, Section 1.4) can be applied to establish the desired formula rigorously.

**Lemma H.5.** *Let $\boldsymbol{X} \sim \mathcal{N}(\boldsymbol{0}, \boldsymbol{I}_n)$ with large $n$. Then, with high probability, $\boldsymbol{X}$ is approximately uniformly distributed on the sphere $\mathbb{S}^{n-1}(\sqrt{n})$, i.e., $\boldsymbol{X} \sim \operatorname{Unif}(\mathbb{S}^{n-1}(\sqrt{n}))$.*

*Proof.* First, consider $\boldsymbol{Y} := \frac{\boldsymbol{X}}{\|\boldsymbol{X}\|}$. We first show that $\boldsymbol{Y} \sim \operatorname{Unif}(\mathbb{S}^{n-1})$. Note that $\mathbb{S}^{n-1}$ is a compact homogeneous space:

$$\mathbb{S}^{n-1} \cong \operatorname{SO}(n)/\operatorname{SO}(n-1),$$

where $\operatorname{SO}(n) \subset \mathbb{R}^{n \times n}$ denotes the special orthogonal group. Consider the natural action of $\operatorname{SO}(n)$ on $\mathbb{S}^{n-1}$ given by $R \colon \mathbb{S}^{n-1} \to \mathbb{S}^{n-1}$, $\boldsymbol{x} \mapsto R\boldsymbol{x}$ for all $R \in \operatorname{SO}(n)$. Then by the existence and uniqueness of Haar measure (Folland, 2016, Theorem 2.49), $\operatorname{Unif}(\mathbb{S}^{n-1})$ is the unique rotation-invariant probability measure on $\mathbb{S}^{n-1}$. Therefore, it is sufficient to prove that the distribution of $\boldsymbol{Y}$ is rotation-invariant, i.e., $\boldsymbol{Y} \overset{\mathrm{d}}{=} R\boldsymbol{Y}$ for all $R \in \operatorname{SO}(n)$.

Since $\boldsymbol{X} \sim \mathcal{N}(\boldsymbol{0}, \boldsymbol{I}_n)$ and $R \in \operatorname{SO}(n)$, we have $R\boldsymbol{X} \sim \mathcal{N}(\boldsymbol{0}, \boldsymbol{I}_n)$ and $\|R\boldsymbol{X}\| = \|\boldsymbol{X}\|$. Hence,

$$\boldsymbol{Y} = \frac{\boldsymbol{X}}{\|\boldsymbol{X}\|} \overset{\mathrm{d}}{=} \frac{R\boldsymbol{X}}{\|R\boldsymbol{X}\|} = R\boldsymbol{Y},$$

which implies that $\boldsymbol{Y} \sim \operatorname{Unif}(\mathbb{S}^{n-1})$. Similarly, by the uniqueness of the invariant measure,

$$\sqrt{n}\boldsymbol{Y} = \frac{\sqrt{n}}{\|\boldsymbol{X}\|}\boldsymbol{X} \sim \operatorname{Unif}(\mathbb{S}^{n-1}(\sqrt{n})).$$

Moreover, as shown in the proof in Proposition 1, the Laurent-Massart Bound implies that $\|\boldsymbol{X}\| \approx \sqrt{n}$ with high probability when $n$ is large. Therefore,

$$\boldsymbol{X} \approx \frac{\sqrt{n}}{\|\boldsymbol{X}\|}\boldsymbol{X} \sim \operatorname{Unif}(\mathbb{S}^{n-1}(\sqrt{n})). \qquad \square$$

**Lemma H.6** (Corollary 4.8.2 of Bakry et al. (2013)). *Let $U \colon \mathbb{R}^n \to \mathbb{R}$ be $C^2$ such that $\nabla^2 U \succeq \rho \boldsymbol{I}_n$ for some $\rho > 0$. Then the probability measure*

$$\mathrm{d}\mu(\boldsymbol{x}) = \frac{e^{-U(\boldsymbol{x})}}{\int e^{-U(\boldsymbol{y})}\mathrm{d}\boldsymbol{y}}\mathrm{d}\boldsymbol{x}$$

*on $\mathbb{R}^n$ satisfies the Poincaré Inequality with the constant $1/\rho$.*

**Lemma H.7.** *Let $\mu, \nu \in \mathcal{P}(\mathbb{R}^n)$ be two probability measures, and let $f \colon \mathbb{R}^n \to \mathbb{R}^m$ be measurable. Then*

$$f_\#(w_1\mu + w_2\nu) = w_1 f_\#\mu + w_2 f_\#\nu,$$

*for any $w_1, w_2 \in [0, 1]$ with $w_1 + w_2 = 1$.*

*Proof.* For any $A \in \mathcal{B}(\mathbb{R}^m)$,

$$\begin{aligned}
f_\#(w_1\mu + w_2\nu)(A) &= (w_1\mu + w_2\nu)\left(f^{-1}(A)\right) \\
&= w_1\mu\left(f^{-1}(A)\right) + w_2\nu\left(f^{-1}(A)\right) \\
&= w_1 f_\#\mu(A) + w_2 f_\#\nu(A). \qquad \square
\end{aligned}$$

**Lemma H.8.** *Let $\mu$ be a probability measure on $\mathbb{R}^n$. Then, we have*

$$\mathbb{E}_{\boldsymbol{X} \sim \mu} [\|\boldsymbol{X}\|] \leq \sqrt{\mathbb{E}_{\boldsymbol{X} \sim \mu} \left[ \|\boldsymbol{X}\|^2 \right]}.$$

*In particular, if $\mu = \mathcal{N}(\boldsymbol{0}, \boldsymbol{I}_n)$,*

$$\mathbb{E} [\|\boldsymbol{X}\|] \leq \sqrt{n}.$$

*Proof.* Because $\mu$ is a probability measure, by the Hölder's Inequality,

$$\int_{\mathbb{R}^n} \|\boldsymbol{x}\| \cdot 1 \mathrm{d}\mu(\boldsymbol{x}) \leq \left( \int_{\mathbb{R}^n} \|\boldsymbol{x}\|^2 \mathrm{d}\mu(\boldsymbol{x}) \right)^{\frac{1}{2}} \left( \int_{\mathbb{R}^n} 1 \mathrm{d}\mu(\boldsymbol{x}) \right)^{\frac{1}{2}},$$

that is, $\mathbb{E}_{\boldsymbol{X} \sim \mu} [\|\boldsymbol{X}\|] \leq \sqrt{\mathbb{E}_{\boldsymbol{X} \sim \mu}[\|\boldsymbol{X}\|^2]}$. In particular, if $\mu = \mathcal{N}(\boldsymbol{0}, \boldsymbol{I}_n)$, $\mathbb{E}[\|\boldsymbol{X}\|^2] = n$. □

**Lemma H.9.** *Let $\mu_1, \mu_2, \nu_1, \nu_2$ be probability measures on $\mathbb{R}^n$, and let*

$$\mu = w\mu_1 + (1 - w)\mu_2, \quad \nu = w\nu_1 + (1 - w)\nu_2, \quad w \in [0, 1].$$

*Then, we have*

$$\mathcal{W}_1(\mu, \nu) \leq w\mathcal{W}_1(\mu_1, \nu_1) + (1 - w)\mathcal{W}_1(\mu_2, \nu_2).$$

*Proof.* By the existence of optimal coupling on $\mathbb{R}^n$ (Chewi et al., 2024), there is a $\gamma_i \in \Gamma(\mu_i, \nu_i)$ for $i = 1, 2$ such that

$$\mathcal{W}_1(\mu_i, \nu_i) = \int_{\mathbb{R}^n \times \mathbb{R}^n} \|\boldsymbol{x} - \boldsymbol{y}\| \mathrm{d}\gamma_i(\boldsymbol{x}, \boldsymbol{y}).$$

Let

$$\pi = w\gamma_1 + (1 - w)\gamma_2.$$

Clearly, $\pi$ is a probability measure on $\mathbb{R}^n \times \mathbb{R}^n$. Moreover, by definition,

$$\pi(A \times \mathbb{R}^n) = w\gamma_1(A \times \mathbb{R}^n) + (1 - w)\gamma_2(A \times \mathbb{R}^n) = w\mu_1(A) + (1 - w)\mu_2(A) = \mu(A),$$
$$\pi(\mathbb{R}^n \times B) = w\gamma_1(\mathbb{R}^n \times B) + (1 - w)\gamma_2(\mathbb{R}^n \times B) = w\nu_1(B) + (1 - w)\nu_2(B) = \nu(B),$$

which means $\pi \in \Gamma(\mu, \nu)$. Therefore,

$$\begin{aligned}
\mathcal{W}_1(\mu, \nu) &\leq \int_{\mathbb{R}^n \times \mathbb{R}^n} \|\boldsymbol{x} - \boldsymbol{y}\| \mathrm{d}\pi(\boldsymbol{x}, \boldsymbol{y}) \\
&= w \int_{\mathbb{R}^n \times \mathbb{R}^n} \|\boldsymbol{x} - \boldsymbol{y}\| \mathrm{d}\gamma_1(\boldsymbol{x}, \boldsymbol{y}) + (1 - w) \int_{\mathbb{R}^n \times \mathbb{R}^n} \|\boldsymbol{x} - \boldsymbol{y}\| \mathrm{d}\gamma_2(\boldsymbol{x}, \boldsymbol{y}) \\
&= w\mathcal{W}_1(\mu_1, \nu_1) + (1 - w)\mathcal{W}_1(\mu_2, \nu_2). \quad\quad \square
\end{aligned}$$

**Lemma H.10.** *Let $\mu \in \mathbb{R}^n$ be a probability measure such that its support $K$ is closed and convex. Then*

$$\mathbb{E}_{\boldsymbol{X} \sim \mu}[\boldsymbol{X}] \in K.$$

*Proof.* Suppose that $\boldsymbol{m} = \mathbb{E}_{\boldsymbol{X} \sim \mu}[\boldsymbol{X}] \notin K$. By the convexity and closedness of $K$, the strong separation theorem (Rockafellar, 1997) implies that there are $\boldsymbol{u} \in \mathbb{R}^n \backslash \{0\}$ and $c \in \mathbb{R}$ such that $\langle \boldsymbol{u}, \boldsymbol{m} \rangle > c$ and

$$\langle \boldsymbol{u}, \boldsymbol{x} \rangle \leq c, \ \forall \ \boldsymbol{x} \in K.$$

Let $\boldsymbol{X} \sim \mu$. $\boldsymbol{X} \in K$ for almost everywhere and so

$$\langle \boldsymbol{u}, \boldsymbol{X} \rangle \leq c, \quad a.e..$$

Then taking the expectation on the both sides, we have

$$\langle \boldsymbol{u}, \boldsymbol{m} \rangle \leq c,$$

which induces a contradiction. □

**Lemma H.11** (Grönwall's Inequality)**.** *If* $u\colon [0, T] \to \mathbb{R}$ *satisfies the linear ODE inequality as*

$$\frac{\mathrm{d}}{\mathrm{d}t} u(t) \le a(t)u(t) + b(t),$$

*then*

$$u(t) \le u(0)e^{\int_0^t a(r)\mathrm{d}r} + \int_0^t b(s)e^{\int_s^t a(r)\mathrm{d}r}\mathrm{d}s.$$

*Proof.* Let $\Phi(t) = \exp\left(-\int_0^t a(s)\mathrm{d}s\right)$. Then, $\Phi'(t) = -a(t)\Phi(t)$ and

$$\Phi(t)\frac{\mathrm{d}}{\mathrm{d}t}u(t) \le \Phi(t)a(t)u(t) + \Phi(t)b(t) \;\Rightarrow\; \frac{\mathrm{d}}{\mathrm{d}t}\left(\Phi(t)u(t)\right) \le \Phi(t)b(t).$$

By integrating on the both sides of above inequality, we have

$$u(t) \le u(0)e^{\int_0^t a(r)\mathrm{d}r} + \int_0^t b(s)e^{\int_s^t a(r)\mathrm{d}r}\mathrm{d}s. \qquad \square$$

**Lemma H.12.** *If a* $C^1$ *function* $f\colon \mathbb{R}^n \to \mathbb{R}$ *is* $\rho$*-strongly convex, then it satisfies* $\rho$*-Polyak–Łojasiewicz (PL) inequality:*

$$\|\nabla_{\boldsymbol{x}} f(\boldsymbol{x})\|^2 \ge 2\rho\left(f(\boldsymbol{x}) - f(\boldsymbol{x}_*)\right),$$

*where* $\boldsymbol{x}_*$ *is the unique minimizer of* $f$*.*

*Proof.* Because $f$ is $\rho$-strongly convex,

$$f(\boldsymbol{y}) \ge f(\boldsymbol{x}) + \langle \nabla_{\boldsymbol{x}} f(\boldsymbol{x}), \boldsymbol{y} - \boldsymbol{x}\rangle + \frac{\rho}{2}\|\boldsymbol{y} - \boldsymbol{x}\|^2.$$

Minimizing the both sides with respect to $\boldsymbol{y}$, we obtain

$$f(\boldsymbol{x}_*) \ge f(\boldsymbol{x}) - \frac{1}{2\rho}\|\nabla_{\boldsymbol{x}} f(\boldsymbol{x})\|^2,$$

which is precisely the $\rho$-PL inequality. $\qquad \square$

**Lemma H.13.** *Let* $f\colon \mathbb{R}^k \to \mathbb{R}^n$ *be* $L$*-Lipschitz continuous. For two probability measures* $\mu, \nu \in \mathcal{P}(\mathbb{R}^n)$*,*

$$\mathcal{W}_1(f_{\#}\mu, f_{\#}\nu) \le L\mathcal{W}_1(\mu, \nu).$$

*Proof.* Let $(\boldsymbol{X}, \boldsymbol{Y})$ be an optimal coupling for $(\mu, \nu)$, that is, $\boldsymbol{X} \sim \mu$, $\boldsymbol{Y} \sim \nu$, and $\mathcal{W}_1 = \mathbb{E}[\|\boldsymbol{X} - \boldsymbol{Y}\|]$. Besides, $f(\boldsymbol{X}) \sim f_{\#}\mu$ and $f(\boldsymbol{Y}) \sim f_{\#}\nu$. Then, by the Lipschitz continuity of $f$,

$$\begin{aligned}
\mathcal{W}_1(f_{\#}\mu, f_{\#}\nu) &\le \mathbb{E}\left[\|f(\boldsymbol{X}) - f(\boldsymbol{Y})\|\right]\\
&\le L\mathbb{E}\left[\|\boldsymbol{X} - \boldsymbol{Y}\|\right]\\
&= L\mathcal{W}_1(\mu, \nu). \qquad \square
\end{aligned}$$

## I   Preliminaries for Manifold

We provide only the minimal background on smooth manifolds necessary for this work. For a comprehensive treatment, we refer the reader to Lee (2012).

**Definition I.1.** A subset $\mathcal{M} \subset \mathbb{R}^n$ is called a $m$-dimensional (embedded) (sub)manifold of $\mathbb{R}^n$ if there are a family open sets $\{U_\alpha\}_{\alpha \in \Gamma}$ in $\mathbb{R}^n$, a family of open sets $\{V_\alpha\}_{\alpha \in \Gamma}$ in $\mathbb{R}^m$, and a family of smooth ($C^\infty$) maps $\{\phi_\alpha\}_{\alpha \in \Gamma}$ such that

$$\mathcal{M} \subset \bigcup_{\alpha \in \Gamma} U_\alpha, \text{ and } \phi_\alpha\colon V_\alpha \to U_\alpha \cap \mathcal{M}$$

is a diffeomorphism, i.e., $\phi_\alpha^{-1}\colon U_\alpha \cap \mathcal{M} \to V_\alpha$ is also smooth.

Each pair $(\phi_\alpha, V_\alpha)$ is called a chart, and $\{(\phi_\alpha, V_\alpha)\}_{\alpha \in \Gamma}$ is called an atlas of $\mathcal{M}$. In general, a single chart cannot cover the entire manifold $\mathcal{M}$. However, if $\mathcal{M}$ is closed, then there exists a chart $\phi\colon V \to \mathcal{M}$ that can almost cover $\mathcal{M}$, in the sense that the volume measure of the set $\mathcal{M} \setminus \phi(V)$ is zero; see Lee (2019) for more details.

**Definition I.2.** Let $\mathcal{M} \subset \mathbb{R}^n$ be a $m$-dimensional manifold. For any $\boldsymbol{x} \in \mathcal{M}$, the tangent space, denoted $T_{\boldsymbol{x}}\mathcal{M}$, is a vector space defined as

$$T_{\boldsymbol{x}}\mathcal{M} := \{\gamma'(0)\colon \exists\, \varepsilon > 0,\ \gamma\colon (-\varepsilon, \varepsilon) \to \mathcal{M} \text{ smooth}, \gamma(0) = \boldsymbol{x}\}.$$

**Lemma I.1.** *Let $\mathcal{M} \subset \mathbb{R}^n$ be a smooth submanifold. If a $C^1$ function $g\colon \mathbb{R}^n \to \mathbb{R}$ is constant on $\mathcal{M}$, then for any $\boldsymbol{x} \in \mathcal{M}$, $\nabla g(\boldsymbol{x})$ is normal to $\mathcal{M}$; that is, $\nabla g(\boldsymbol{x}) \perp T_{\boldsymbol{x}}\mathcal{M}$.*

*Proof.* For any $\boldsymbol{v} \in T_{\boldsymbol{x}}\mathcal{M}$, let $\gamma\colon [0,1] \to \mathcal{M}$ be a smooth curve such that $\gamma(0) = \boldsymbol{x}$ and $\gamma'(0) = \boldsymbol{v}$. Then, because $g(\gamma(t)) \equiv \boldsymbol{c}$,

$$0 = \frac{\mathrm{d}}{\mathrm{d}t}\bigg|_{t=0} g(\gamma(t)) = \langle \nabla g(\gamma(0)), \gamma'(0) \rangle = \langle \nabla g(\boldsymbol{x}), \boldsymbol{v} \rangle$$

Therefore, $\nabla g(\boldsymbol{x}) \perp T_{\boldsymbol{x}}\mathcal{M}$. $\qquad\square$

**Theorem I.2** (Constant Rank Theorem (Lee, 2012)). *Let $f\colon \mathbb{R}^n \to \mathbb{R}^r$ be a smooth map and $\boldsymbol{c} \in \mathbb{R}^r$. Let*

$$\mathcal{M} := \{\boldsymbol{x} \in \mathbb{R}^n\colon f(\boldsymbol{x}) = \boldsymbol{c}\}.$$

*If $\operatorname{rank} JF(\boldsymbol{x}) = r$ for any $\boldsymbol{x} \in \mathcal{M}$, then $\mathcal{M}$ is a $(n-r)$-dimensional manifold.*

**Theorem I.3** (Tubular Neighborhood Theorem (Lee, 2012)). *Let $\mathcal{M} \subset \mathbb{R}^D$ be a $d$-dimensional submanifold. There is a smooth $\varepsilon\colon \mathcal{M} \to (0, \infty)$ such that for*

$$V := \left\{(\boldsymbol{z}, \boldsymbol{v}) \in \mathcal{M} \times \mathbb{R}^{D-d}\colon \|\boldsymbol{v}\| < \varepsilon(\boldsymbol{z})\right\},$$

*$F\colon V \to U = F(V)$ is a diffeomorphism and $U \subset \mathbb{R}^D$ is a neighborhood of $\mathcal{M}$.*

*Remark* I.1. For a given tubular neighborhood $V$ of $\mathcal{M}$, we also call $U = F(V) \subset \mathbb{R}^D$ is its tubular neighborhood in $\mathbb{R}^D$. Moreover, we can define the corresponding orthogonal projection $\pi\colon U \to \mathcal{M}$ as

$$\pi(\boldsymbol{x}) = \pi_1(F^{-1}(\boldsymbol{x})),$$

where $\pi_1\colon V \to \mathcal{M}$ is $\pi_1(\boldsymbol{z}, \boldsymbol{v}) = \boldsymbol{z}$.

