# OpenReview forum: "Understanding Guidance Scale in Diffusion Models from a Geometric Perspective"
_TMLR — Accepted by TMLR_

### Review · Reviewer_xZ9X · 2025-12-02

**Summary Of Contributions:**

### Summary

The paper provides a geometric viewpoint on conditional diffusion guidance by modeling class-conditional data as lying near low-dimensional manifolds. Under this interpretation, classifier and classifier-free guidance correspond to pushing samples in the normal direction of the target class manifold to offer an intuitive explanation of how the guidance scale $η$ influences sample alignment and fidelity. The authors introduce a geometric guidance ODE that replaces probabilistic guidance with an explicit normal-direction projection toward the target manifold. In the linear case, this reduces to the analytically tractable form $-η(I - AA^T)x$.

Two key theoretical results are presented:

1. Theorem 10: Increasing η contracts samples exponentially toward the target manifold:  $E[\text{dist}(X, M_y)] \leq e^{-\eta} + \frac{1}{\eta}$

2. Theorem 11: The Wasserstein-1 distance between the geometric guidance model (GeGM) and the true conditional distribution satisfies
   $W_1 \leq O(e^{-T} + \sqrt{\delta} + \sigma + \frac{1}{\eta}) + \tilde{C}$
   implying that larger η does not worsen the theoretical error bound.

### Strengths

1. The geometric interpretation provides a clear and intuitive explanation of the role of guidance in diffusion models.

2. The theoretical analysis is rigorous and shows that increasing η moves samples closer to the target manifold without worsening the distributional error.

3. The proposed geometric ODE gives a principled alternative to probabilistic guidance.

### Weaknesses

1. As shown in Theorem 11, the bound includes a non-zero constant $C̃$, implying that the nonlinear geometric guidance model cannot exactly match the true conditional distribution.
2. Transition from probabilistic to geometric guidance lacks mathematical rigor. The claim that $\nabla_x \log p_t(y|x) \approx -\eta P_1 x$ relies on informal arguments about approximate constancy without explicit error bounds.

3. Experiments are limited to CIFAR-10 and do not provide direct comparisons to standard probabilistic guidance, making it difficult to assess relative effectiveness.

**Audience:**

Yes

**Audience Explanation:**

TMLR focuses on machine learning theory, and this work touches several active directions:
- **Diffusion Theory:** Extends prior 1D or GMM-based analyses with a more general geometric framework.
- **Generative Modeling:** Offers theoretical insight into conditional diffusion guidance.
- **Manifold Geometry:** Provides a geometric interpretation linking score fields and class manifolds.
- **Analytical Tools:** The mollification construction is technically interesting and potentially reusable.

These concepts are of great interest to researchers studying diffusion model theory, conditional guidance, or geometric generative modeling.

**Broader Impact Concerns:**

The broader impact of this work centers on both positive and negative implications of improving theoretical understanding of conditional diffusion models. On the positive side, stronger guidance mechanisms may enable better synthetic data for addressing class imbalance or improving data efficiency. However, the same improvements could also facilitate the generation of sensitive or undesirable content if misused. Additionally, the reliance on linear manifold assumptions may not hold for highly structured datasets, potentially leading to biased or unpredictable behavior in real-world scenarios.

**Claims And Evidence:**

Yes

**Claims Explanation:**

The submission provides strong theoretical support for its core linear-manifold claims, including score decomposition for manifold recovery (as discussed in Theorem 2) and the stability of mollification under Wasserstein and log-concavity arguments.
Overall, while the foundational linear theory is solid, the paper still contains specific gaps, particularly in justifying geometric guidance rigorously and extending theory to the nonlinear case.

### Identified Gaps in the Submission

1) **Geometric Guidance Justification**
The claim that $\nabla_x \log p_t(y=1 \mid x)$ is “almost normal’’ to $M_1^t$ (implying $-\eta P_1 x$) is only supported by Lemma I.1, which assumes exact manifold constancy. The paper does not quantify $\|\log p_t(y \mid x) - c\|$ on $M_1^t$, so the accuracy of this approximation is unclear.

2) **Assumption III Validation**
Assumption III ($M < 2\sqrt{m-1}$) is validated only for a synthetic log-concave family, with no empirical check on CIFAR-10 or method to estimate $M$ from data. Thus, it is uncertain whether real image distributions satisfy the conditions required for the theoretical guarantees.

3) **Limited Empirical Validation of Nonlinear Theory**
The nonlinear guidance method (Section 6) is evaluated only on CIFAR-10, which is relatively simple and low-dimensional. There is no experimentation on more complex or highly nonlinear datasets (e.g., ImageNet subsets, CelebA-HQ, or natural image manifolds with nontrivial curvature), making it difficult to assess whether the nonlinear formulation reflects the intended geometric principles or generalizes beyond a single dataset.

**Requested Changes:**

**Formalize the Geometric Guidance Derivation**
   Provide a quantitative approximation guarantee such as
   $\|\nabla_x \log p_t(y=1 \mid x) + \eta P_1 x\| \le \epsilon(t)$
   where $\epsilon(t)$ depends on manifold curvature, classifier smoothness, and deviation of $\log p_t(y=1\mid x)$ from constancy on $M_1^t$.

**Characterize and Validate Assumption III**
   - Give sufficient conditions for when $M < 2\sqrt{m-1}$ holds.
   - Provide concrete examples (e.g., Gaussian mixtures, truncated Gaussians).
   - Propose a practical estimator for $M$ using observed data.
   - Empirically verify the assumption on CIFAR-10.

**Quantify the Error Floor in Theorem 11**
   - Report empirical values of $\tilde{C}$ on CIFAR-10.
   - Compare $\tilde{C}$ to $e^{-T}$, $\delta^{1/2}$, $\sigma$, and $\eta^{-1}$ to evaluate practical relevance.
   - Discuss whether the error floor is fundamental to geometric guidance.

**Provide Theory for the Nonlinear Case**
   Either introduce convergence or stability guarantees for the nonlinear flow, or clearly state that it is intended as an empirical contribution only.

**Expanded and Fair Experiments:**
  Ensure comparable guidance strength between CGM and GeGM.
  Add ablations on $\eta, \sigma, \delta, \kappa, M$.
  Critically, evaluate nonlinear guidance on harder datasets (e.g., ImageNet subsets, CelebA-HQ).

---

> ### Author Response · Authors · 2025-12-08
>
> We appreciate the reviewer’s careful reading and constructive suggestions.
>
> **Scope and positioning.** Our paper’s main goal is to provide theoretical analysis in the linear setting. Section 5 develops guarantees under a linear manifold hypothesis; the assumptions (e.g., Assumptions II–III) and the results in Theorems 10 and 11 are stated only in this regime. We acknowledge that Section 6 is an initial, constructive step: it introduces a principled way to build a nonlinear geometric guidance term, but we cannot claim that the linear-theory guarantees transfer directly. Accordingly:
>
> - It is not meaningful to test Assumption III (estimating $M$ ) or the constant $\tilde{C}$ from Theorem 11 on nonlinear CIFAR-10 experiments, since these quantities are defined and justified under the linear manifold hypothesis.
>
> - Theoretically, Section 6 provides a fundamental construction of a nonlinear geometric guidance term adapted to the nonlinear setting. Establishing analogues of Theorems 10–11 in this setting will require different assumptions and is left for future work (as discussed in Limitations, Section 7).
>
> - Experimentally, Section 6.3 has two goals: (i) demonstrate that the nonlinear geometric guidance model can generate target conditional samples, which partially supports Theorem 13 and our construction intuition of geometric guidance term; and (ii) examine how performance changes with guidance scale in our model. The observed trends are consistent with the spirit of Theorem 11 in nonlinear regimes, but are not derived from it.
>
> **Dataset choice.** We use CIFAR-10 for two reasons: (i) it is nonlinear enough to test the construction beyond the linear case; (ii) it is not as complex as ImageNet, so our first realization of $F_{y, \theta}^t$ is feasible. Given that the paper's goal is theory, we acknowledge the nonlinear experiments are foundational. Building robust nonlinear geometric guidance (e.g., beyond simple $\\|\nabla_x F_{y, \theta}^t(x)\\|$ control, which is time-consuming and unstable on larger datasets) is important future work.
>
> Below we address the specific points.
>
> ### R1. Formalize the Geometric Guidance Derivation
>
> We sincerely thank the reviewer for highlighting this point, which prompted us to sharpen our intuition. To formalize the idea that $\nabla_{x} \log p_t(y=1 \mid x)$ is "almost normal" to $M_1^t$, We have obtained, under mild smoothness of the classifier, that
> $$
> \\|\nabla_x \log p_t(y=1 \mid x)+\eta_t P_1 x\\| \leq \beta_t, \quad \forall x \in M_1^t,
> $$
> for some scalar $\eta_t>0$. Moreover, if the classifier is confident on $M_1^t$,
> $$
> p_t(y=1 \mid x)>1-\varepsilon_t, \quad \forall x \in M_1^t,
> $$
> then $\beta_t=\mathcal{O}\left(\varepsilon_t\right)$. Intuitively, on $M_1^t$ the probabilistic guidance is almost parallel to the geometric normal, with a residual controlled by classifier confidence.
>
> We will add this result with a rigorous and detailed analysis in the appendix.
>
> ### R2. Characterize and Validate Assumption III
>
> **Sufficient conditions.** We have discussed Assumption III in Appendix E.2, and we apologize for not stating the results explicitly. In the revision, we will make it explicit that if
> $$
> p^Z_i(z) \in \\{e^{-V(z)}\chi_K(z) \colon \nabla^2 V \succeq mI,~K \text{ is compact and convex.}\\},
> $$
> then $p^Z_{i,\sigma}$ (Eq. (11)) is strongly log-concave (Proposition E.1). Moreover, choosing $\sigma$ so that
> $$
> M \leq \frac{\sqrt{|K_1|^2 + |K_2|^2}}{\sigma^2} \leq 2\sqrt{m-1} \quad\Longleftrightarrow\quad \sigma^2 \geq \sqrt{\frac{|K_1|^2 + |K_2|^2}{4(m-1)}},
> $$
> ensures Assumption III holds (Proposition E.2). Consequently, the mixture latent distribution $p^Z_\sigma$ (Eq. (10)) is $m_0^z$-strongly log-concave (Theorem 7), which further implies for the geometric guidance model ($*$) that the smoothed density $p^\sigma_t$ satisfies:
> $$
> \\|\nabla^2_{x}\log p_t^\sigma(x)\\| \leq L_t,\quad -\nabla^2_{x}\log p_t^\sigma(x) \succeq m_tI_D.
> $$
> We will add these statements explicitly at the end of Appendix E.2.
>
> **Concrete examples.** The above applies, for example, when each $p^Z_i(z)$ is a lower-dimensional Gaussian truncated to a compact, convex set $K$; with an appropriate choice of $\sigma$, Assumption III follows.
>
> **On estimating $M$ and validating III on CIFAR-10.** As noted at the beginning, because Assumptions II–III belong to the linear-manifold theory, testing $M$ or validating Assumption III on nonlinear CIFAR-10 would not be meaningful. Establishing nonlinear analogues of Theorems 10–11 requires new assumptions, which we leave to future work.

---

> ### Author Response · Authors · 2025-12-08
>
> ### R3. Quantify the Error Floor in Theorem 11
>
> **Empirical values on CIFAR-10.** Because the analysis in Theorem 11 is derived under the linear manifold hypothesis, there is no canonical $\tilde{C}$ in the nonlinear setting. As an empirical indication of the residual gap between the target and generated distributions, the FID reported in Fig. 2 (Sec. 6.3) and Fig. 3 (Appendix G.2) partially reflects this discrepancy.
>
> **Comparison of parameters.** Here $T$ is the final diffusion time, and $\delta,\sigma$ are hand-chosen parameters used to avoid analytical irregularities. As shown in the proof of Theorem 11, $\tilde{C} = \tilde{C}_1 + \tilde{C}_2$:
> - $\tilde{C}_1$ relates to first moments of latent density and is independent of $T,\delta,\sigma$ (see Eq. (36), Lemma D.3).
> - $\tilde{C}_2$ arises from score bounds (Eq. (38), Proposition D.5) and hence depends subtly on $T,\delta,\sigma$ (Appendix D.2).
>
> The guidance scale $\eta$ is hand-chosen and independent of above terms. Since Theorem 11 is a linear result, for nonlinear CIFAR-10 we do not compare $\tilde{C}$ to $T,\delta,\sigma$; instead, we report the empirical effect of $\eta$ (Fig. 2; Fig. 3).
>
> **Is $\tilde{C}$ fundamental?** At present, we believe the error floor is inherent to the geometric guidance model. Because of the analytical simplicity of the geometric guidance, it cannot provide as much information as the probability guidance term did. For example, although we show
> $$
> \\|\nabla_{x} \log p_t(y=1 \mid x)+\eta_t P_1 x\\| \leq \beta_t,
> $$
> when $p_t(y=1 \mid x) > 1 -\varepsilon_t$ for all $x \in M^t_1$, indicating that probabilistic guidance $\nabla_{x} \log p_t(y=1 \mid x)$ is “almost parallel” to geometric guidance $P_1 x$, the norm of the probabilistic guidance carries information that the geometric term cannot capture. This is a trade-off made for the sake of analytical tractability.
>
>
> ### R4. Provide Theory for the Nonlinear Case
>
> As discussed in Limitations (Sec. 7), the main theoretical results on guidance scale in Sec. 5 do not directly apply to the nonlinear construction in Sec. 6. The nonlinear extension provides a fundamental approach to construct geometric guidance; proving results analogous to Theorems 10 and 11 for the nonlinear model under appropriate assumptions is an important direction for future work.
>
> ### R5. Expanded and Fair Experiments
>
> **Comparable guidance strength (CGM vs. GeGM).** The norm of $\nabla_{x} \log p_t(y=1 \mid x)$ and $P_1x$ are not directly comparable—indeed $\\|\nabla_x \log p_t(y=1 \mid x)+\eta_t P_1 x\\| \leq \beta_t$ reflects this mismatch. In practice, it is reasonable to choose different guidance strengths for CGM and GeGM. Our results (Table 1; Fig. 2 in Sec. 6.3; Table 2 and Fig. 3 in Appendix G.2) show GeGM performs best around $\eta \approx 50$, whereas CGM typically uses $\eta < 10$.
>
> **Ablations.** The parameters $\sigma,\delta,M$ appear in the linear-manifold analysis (with $\sigma,\delta$ used as hand-chosen parameters to manage analytical irregularities) and are not applicable to the nonlinear CIFAR-10 setting. We therefore keep CIFAR-10 ablations focused on $\eta$ (already reported in Fig. 2 and Fig. 3).
>
> **Evaluation on harder datasets.** We acknowledge that our nonlinear experiments are not exhaustive and that the construction is foundational. Developing a robust nonlinear geometric guidance model will require additional work. For example, our current training approach for $F^t_{y,\theta}$ which relies on constraining $\\|\nabla_x F^t_{y,\theta}(x)\\|$, can be time-consuming and unstable on more complex datasets such as ImageNet. Designing scalable training strategies and a well-adapted geometric guidance model for such datasets is a future topic.
>
> ### Broader Impact
>
> We appreciate the concern. Our work is theoretical—not a deployable system—and we will add a brief note that any practical use should include content-safety filtering and bias audits, given potential dual-use risks. We also clarify that our linear assumptions are idealizations and may not hold in practice; potential benefits (e.g., class balancing, data efficiency) are noted alongside these limits.

---

> ### Comment · Reviewer_xZ9X · 2025-12-21
>
> I thank the authors for their thoughtful response and clarifications. While the linear-manifold theory is clearly well developed, a few points would benefit from further explanation. I am not an expert in this area, and my comments are intended as constructive suggestions rather than critical remarks.
>
> ## Few Questions
> 1. The arguments rely on the classifier being confident on the data manifold, but it is not discussed how often this condition holds during the diffusion process, especially in early or ambiguous stages of sampling.
>
> 2. Authors have explained why guidance strengths are not directly comparable, but it would still be helpful to explain (whether theoretically or empirically) whether geometric guidance offers practical advantages over standard classifier-free guidance when both are tuned for similar output quality.
>
> 3. After the response, it would still be helpful to more explicitly state either in the introduction or in the abstract whether the paper is intended as:
>    - a) a theoretical explanation of existing probabilistic guidance,
>    - b) a principled but simplified alternative guidance method, or
>    - c) primarily a conceptual framework for understanding guidance behavior.

---

> ### Author Response · Authors · 2025-12-21
>
> We thank the reviewer for the quick follow-up and the constructive suggestions. We respond to the three points below.
>
> 1. We agree that at early stages of the reverse process (small $t$ ), $\nabla_x \log p_{T-t}(y=1 \mid x)$ may not be well-approximated by geometric guidance, since $p_{T-t}(y=1 \mid x)$ can be less confident due to class mixing. Importantly, this does not affect our main linear-theory guarantee: the strong log-concavity of $p_t^\sigma$ stabilizes the reverse dynamics with respect to the initial condition. Concretely, in the proof we may replace the initial condition $\tilde{X}_0 \sim \mathcal{N}(0, I)$ by an arbitrary distribution $q$ (e.g., the distribution before class mixing); accordingly, in Proposition D.4 (p.36) we take $\hat{X}_0 \sim q$. Then Equation (52) on p. 37 still yields an initial-error term of order $\mathcal{O}(e^{-T})$. We thank the reviewer for raising this point and will add a short discussion at the end of Appendix C.3.
>
> 2. At present, our focus is primarily theoretical: the geometric surrogate is analytically tractable, which allows us to study the role of the guidance scale $\eta$ in a controlled setting. For practical advantages over CFG/CG, especially in nonlinear settings, we currently lack corresponding theory and our nonlinear construction is still relatively naive; thus we do not claim empirical superiority. A careful comparison at matched output quality, together with improved nonlinear analysis and training of geometric guidance, is an important direction for our future work.
>
> 3. Thank you for this suggestion. Our intent is closest to (c): a conceptual/theoretical framework for understanding guidance behavior, particularly the guidance scale, using a principled surrogate that approximates probabilistic guidance under our assumptions. We will state this more explicitly in the introduction (and adjust wording to avoid over-interpreting the results as a direct theory of standard CFG/classifier guidance).

---

### Review · Reviewer_vsZJ · 2025-12-17

**Summary Of Contributions:**

The paper studies the problem of controlling diffusing models to generate conditional distributions, where, importantly, a guidance term appears. The authors propose a new linear geometric guidance term, assuming a linear manifold hypothesis, which requires certain regularity conditions. To ensure these conditions are fulfilled, the authors use mollification technique to construct a surrogate score function. Finally, the method is extended to nonlinear settings.

Overall, the paper is well written and really easy to follow. Although the paper is sometimes math-dense, the authors did a good job of introducing each term with enough details. Moreover, if more information is needed, the appendix is well structured and includes substantial extra details about the results.

However, I think the experiment section felt short; it lacks details and discussion about the results, and it is not convincing.

**Audience:**

Yes

**Audience Explanation:**

Yes, studying the problem of controlling diffusing models to generate conditional distributions is of interest to the community.

**Claims And Evidence:**

Yes

**Claims Explanation:**

The claims made in the paper are generally supported by theoretical results

**Requested Changes:**

I am not an expert in this topic, so the following points should be viewed more as suggestions to improve the work rather than critical adjustments:

1. I think it would be helpful to add a discussion about the choice of $\sigma$ in the mollification; there is no analysis about the impact of this hyperparameter in the method.
2. I am wondering if there is any scenario where considering the linear geometric term would be useful? If so, I think it would be interesting to do an experiment to showcase this, since most of the paper focuses on the linear manifold assumption.
3. In my opinion, since it's a novel assumption, we authors should provide more intuition about assumption 3. In particular, it is important to discuss how restrictive this assumption is and to at least give a simple example in the main text when this assumption is fulfilled.
4. I think more details in the experiments are necessary. For instance, it is not clear which dataset is used in the first experiment; I suppose it's the CIFAR-10, but there is no information about it in the main paper. Moreover, in Table 1, why is the guidance scale selected as $\eta=5$ and $\eta=50$?.  Figure 2: Is there any intuition why the FID increases for $\eta=100$? Overall, I think the experiment section needs more details and discussion of the results.
5. Why do the authors set $\kappa=1$ for eq (16)? Could give more intuition about it?
6. One minor comment, I think the proofs in the main text are not necessary. They make the paper longer and are not particularly insightful. Therefore, I would suggest moving them to the appendix.

---

> ### Author Response · Authors · 2025-12-20
>
> We thank the reviewer for the positive assessment of the writing and the constructive suggestions on how to improve clarity and presentation. We address each point in turn and will incorporate the suggested clarifications in the revision.
>
> **Choice of $\sigma$:** The primary role of $\sigma$ is to guarantee Assumption III so that the surrogate score function satisfies the desired regularity properties (and $\sigma$ also helps ensure existence/smoothness of the density after mollification). We discussed this in Appendix E.2 but did not state it explicitly. In the revision, we will make explicit that if each $p^Z_i$ belongs to $\\{e^{-V(z)}\chi_K(z) \colon \nabla^2 V \succeq mI,~K \text{ is compact and convex.}\\}$, then choosing $\sigma^2 \geq c$ for some constant $c$ ensures Assumption III. We will add this statement at the end of Appendix E.2.
>
> Regarding impact: in Theorem 10, $\sigma$ only affects constants through score Lipschitz bounds (Appendix D.2, Equation (25)); in Theorem 11, beyond the explicit $\mathcal{O}(\sigma)$ term, $\tilde{C}$ is also affected by $\sigma$ in a more subtle way. In the proof, $\tilde{C}=\tilde{C}_1+\tilde{C}_2$, where $\tilde{C}_1$ is independent of $\sigma$ (Equation (36), Lemma D.3), while $\tilde{C}_2$ comes from score bounds and therefore depends on $\sigma$ through those bounds (Equation (38), Proposition D.5; see Appendix D.2).
>
> **Usefulness of the linear geometric term:** We agree with the reviewer that for many real-world datasets it can be unrealistic to ignore curvature entirely and assume a global linear manifold. However, our intention is that the linear setting serves as an analytically tractable base case that motivates the nonlinear extension. In particular, the linear analysis provides conceptual and technical guidance for constructing the nonlinear term in Section 6. We acknowledge that Section 6 is only a first step in that direction: it provides a fundamental construction of the nonlinear geometric guidance term, while proving nonlinear analogues of Theorems 10–11 under appropriate assumptions is left for future work (as discussed in Limitations, Section 7).
>
> **More about Assumption III:** Thank you for pointing this out—we agree that Assumption III needs more intuition in the main text. While we discussed it in Appendix E.2, we did not state the most direct sufficient conditions explicitly.
>
> In the revision, we will state explicitly that if
> $$
> p^Z_i(z) \in \\{e^{-V(z)}\chi_K(z) \colon \nabla^2 V \succeq mI,~K \text{ is compact and convex.}\\},
> $$
> then $p^Z_{i,\sigma}$ (Eq. (11)) is strongly log-concave (Proposition E.1). Moreover, choosing $\sigma$ so that
> $$
> M \leq \frac{\sqrt{|K_1|^2 + |K_2|^2}}{\sigma^2} \leq 2\sqrt{m-1} \quad\Longleftrightarrow\quad \sigma^2 \geq \sqrt{\frac{|K_1|^2 + |K_2|^2}{4(m-1)}},
> $$
> ensures Assumption III (Proposition E.2). Consequently, the mixture latent distribution $p^Z_\sigma$ (Eq. (10)) is $m_0^z$-strongly log-concave (Theorem 7), which further implies for the geometric guidance model ($*$) that the smoothed density $p^\sigma_t$ satisfies the desired regular properties (Theorem 8). We will add these statements explicitly at the end of Appendix E.2.
>
> Moreover, the above applies, for example, when each $p^Z_i(z)$ is a lower-dimensional Gaussian truncated to a compact, convex set $K$; with an appropriate choice of $\sigma$, Assumption III follows. We will add this example in Remark 4 under Assumption III.

---

> > ### Author Response · Authors · 2025-12-20
> >
> > **More experimental discussions:** We choose CIFAR-10 as the dataset for two reasons. First, it is not too simple, which means that it has the nonlinear structure so that we can use it to test the nonlinear model. Second, it is not too complicated, like ImageNet, which means that our first construction of nonlinear geometric guidance term $F^t_{y,\theta}$ can be well-approximated. Constructing a robust nonlinear geometric guidance model still requires more effort in the future.
> >
> > For $\eta$, as shown in Figure 2 , GeGM achieves its best performance around $\eta=50$, while CGM achieves its best performance around $\eta \approx 1$, as many previous works have shown. Therefore, we choose these values for comparison in Table 1 (Section 6.3) and Table 2 (Appendix G.2). The reason they differ is that the norm of $P_1 x$ is not comparable to the norm of $\nabla_x \log p_t(y=1 \mid x)$. More specifically, thanks to Reviewer xZ9X and bkAk, we clarify our original intuition that $\nabla_x \log p_t(y=1 \mid x)$ is "almost parallel" to $P_1 x$ :
> > $$
> > \\|\nabla_x \log p_t(y=1 \mid x)+\eta_t P_1 x\\| \leq \beta_t, \quad \forall x \in M_1^t,
> > $$
> > for some scalar $\eta_t>0$. Moreover, $\beta_t=\mathcal{O}(\varepsilon_t)$ when the classifier confidence satisfies $p_t(y=1 \mid x)> 1-\varepsilon_t$ for all $x \in M_1^t$. This indicates that while the directions can align, the norm of the probabilistic guidance contains more information which the geometric guidance cannot provide; therefore, their norms are not directly comparable.
> >
> > When $\eta$ becomes too large, the increase of FID cannot be directly explained by Theorem 11, since that result is derived under the linear manifold hypothesis and for continuous-time dynamics. We thank the reviewer for pointing out this issue. In the original version, we only hypothesized that the degradation might be caused by discretization. To clarify this point, we now analyze the additional error introduced by the Euler discretization and show that the discretization error is bounded by $\mathcal{O}(h \eta^2)$, where $h$ is the step size. Therefore, under the linear assumptions, when $\eta$ is too large, the discretization error can grow rapidly and degrade performance, which is consistent with experimental observations: as noted below Theorem 11, Wu et al. (2024, Figure 3) reported that large guidance scales can harm modality on a simple synthetic dataset, and that this issue can be mitigated by reducing the discretization step size. We will add this result with rigorous analysis in the appendix in the revised version.
> >
> > However, for nonlinear settings such as CIFAR-10, we currently lack a corresponding theoretical analysis, so we cannot provide a formal explanation; although our intuition is that large $\eta$ similarly amplifies discretization error, it may be also induced by other factors practically (e.g., the training method for the nonlinear guidance). Extending the discretization analysis to the nonlinear case is an important direction for future work.
> >
> > Overall, since our main goal is theory under the linear manifold hypothesis, we acknowledge that our experiments are not exhaustive and that the nonlinear construction is still foundational. Section 6.3 has two purposes: (i) to demonstrate that the nonlinear GeGM can generate target-conditioned samples, partially supporting Theorem 13 and our construction philosophy; and (ii) to study how performance varies with the guidance scale. While Theorem 11 is a linear, continuous-time result and does not directly apply, the observed trends are qualitatively consistent and motivate further theory in the nonlinear setting. Extending the method to more complex datasets (e.g., ImageNet) will likely require improved training strategies for $F_{y, \theta}^t$, since directly constraining $\\|\nabla_x F_{y, \theta}^t(x)\\|$ can be time-consuming and unstable; we view this as important future work.
> >
> > **Choice of $\kappa$:** The parameter $\kappa$ controls the trade-off between driving $F_{y, \theta}^t(x)$ to zero and encouraging large gradients $\nabla F_{y, \theta}^t(x)$. Mathematically, $\kappa$ sets a length scale: after rescaling, it can be absorbed into the parametrization, so taking $\kappa=1$ is without loss of generality.
> >
> > Practically, if $\kappa$ is too small, $\\|\nabla F^t_{y,\theta}(X)\\|$ tends to be small, so it does not provide sufficient guidance. Conversely, if $\kappa$ is too large, $F^t_{y,\theta}(x)$ may deviate from $0$ on $M^t_1$, in which case $\nabla F^t_{y,\theta}(X)$ is no longer meaningful for characterizing $M^t_1$. We therefore normalize inputs and outputs so that the two terms in the loss have comparable norm, and under this normalization $\kappa = 1$ is a natural, balanced choice.
> >
> > **Proofs:** Thank you for the suggestion. Our intent was to provide intuition via short proofs/sketches of some main theorem. In the revision, we will consider moving these proofs to the appendix.

---

### Review · Reviewer_bkAk · 2025-12-18

**Summary Of Contributions:**

Classifier guidance and classifier-free guidance are central techniques in diffusion-based generative modeling. In practice, the conditional score term $\nabla_x \log p_t(y\mid x)$ is not analytically tractable and must be approximated (e.g., via an auxiliary classifier or an implicit CFG construction). Despite its widespread use, the role of the guidance scale $\eta$ is strong empirically but remains theoretically under-explained. Motivated by the fact that practitioners routinely tune $\eta$, this paper aims to provide a geometrically interpretable perspective on guidance and to derive bounds that clarify the effect of $\eta$.


**Geometric reinterpretation of probabilistic guidance**

To pursue this goal, the paper adopts a linear manifold hypothesis. Specifically, it assumes that the data for each class live in a linear subspace spanned by the columns of $A_i$ for the $i$-th class. Under this hypothesis, the paper argues that the conditional guidance term $\nabla_x \log p_t(y\mid x)$ acts (approximately) in the normal direction of the noisy class manifold, and can therefore be replaced by a simple geometric vector field $-\eta P_i x$ with $P_i = I - A_iA_i^\top$.

Using this replacement, the authors define a “geometric guidance model” and prove that as $\eta$ increases, samples become closer to the target class manifold (Theorem 10). They also provide a coarse $W_1$ upper bound to the target conditional distribution containing an $O(\eta^{-1})$ term (Theorem 11).

Note that the proposed geometric-guidance ODE analysis requires regularity of the score field $\nabla_x \log p_t(\cdot)$ (existence as a function, Lipschitz-type control, etc.). In particular, for a purely manifold-supported distribution, densities can be singular; and in multimodal settings, scores can behave poorly. In this regard, the authors explicitly introduce mollification to “ensure existence of density,” avoid non-smoothness issues, and obtain strong log-concavity-type properties needed for the analysis.

For real data, they propose learning implicit functions $F_\theta^t$ so that noisy manifolds are approximated as level sets $M_t = (F_\theta^t)^{-1}(0)$, and then using $\nabla_x F_\theta^t$ as the geometric guidance signal.

**Nonlinear extension via learned level sets.**

Beyond the linear subspace setting, the paper assumes a union-of-manifolds hypothesis: each class $y$ lies on a nonlinear manifold $M_y \subset \mathbb{R}^D$. At a fixed diffusion time $t$, the class-conditional noisy distribution is argued to concentrate near a $(D-1)$-dimensional hypersurface that can be represented as a level set $M_y^t = ( x : f_y^t(x)=r(t) )$. Since $\nabla f_y^t(x)$ is normal to this surface, the paper proposes replacing probabilistic guidance by a normal-direction term and obtains a nonlinear geometric-guidance ODE with $-\eta \nabla_x f_y^{T-t}(x)$. Because $f_y^t$ is unknown, they learn an implicit function $F_{y,\theta}^t$ from forward-noised class-$y$ samples such that $M_y^t \approx (F_{y,\theta}^t)^{-1}(0)$, and then use $\nabla_x F_{y,\theta}^t$ as the guidance signal.

**Audience:**

Yes

**Audience Explanation:**

Some TMLR readers would still be interested, but the appeal is likely narrower and more conditional than in a “direct CFG theory” paper.

On the positive side, guidance (classifier/CFG) and the tuning of (\eta) are ubiquitous in practice, and the paper offers a clean, geometry-driven lens (“guidance as normal-direction manifold contraction”) with formal $\eta$-dependent statements—so researchers who care about *analyzable surrogate models* for conditional sampling, manifold hypotheses, or Wasserstein-style stability bounds may find the framework and proofs informative.

However, the interest may be substantially reduced for a large portion of TMLR’s applied diffusion audience because the main theorems are not about the actual guidance mechanisms people use: the paper replaces $\nabla_x\log p_t(y\mid x)$ with a hand-crafted proxy (projector-based regularizer), introduces mollification (changing the target), and still yields a bound with a non-vanishing constant floor. In addition, its optimistic framing of large (\eta) can be seen as incomplete or potentially at odds with other analyses emphasizing failure modes under score error. In that sense, readers seeking operational insight comparable to more practice-facing work (e.g., Autoguidance-style explanations/interventions) may find the contribution less direct, whereas theory-oriented readers may still value it as a principled but idealized proxy study of guidance scaling.

**Claims And Evidence:**

Yes

**Claims Explanation:**

**Strengths (well-supported):**

- The paper’s *internal* theoretical claims about the geometric guidance surrogate are supported by formal statements and proofs under explicit assumptions: e.g., Theorem 10 gives an $\eta$-dependent convergence rate of the off-manifold component, and Theorem 11 gives a corresponding Wasserstein bound.
- The need for a surrogate / mollification is explained clearly (lack of density for manifold-supported distributions; multi-modality causing irregularities), and the mollified distribution’s closeness is quantified.

**Concerns (claims may be overstated relative to evidence):**

- The abstract and narrative sometimes read as if the theory explains practical guidance scale in real conditional diffusion, but the core proofs are for a modified dynamics where $\nabla_x \log p_t(y\mid x)$ is replaced by $-\eta P_i x$. This is acknowledged as a replacement made “to avoid the difficulty” of analyzing $\nabla_x \log p_t(y\mid x)$, but it weakens the directness of the conclusions for classifier/CFG guidance.
- Theorem 11 includes an additive constant $\tilde{C}$ in the $W_1$ bound, meaning the bound does not go to zero as $\eta\to\infty$; this limits how strongly one can interpret the result as “increasing $\eta$ improves generation performance”.
- The well-posedness results with mollification rely on assumptions that are both strong and somewhat hard to validate for realistic diffusion latents.
- The paper suggests that large-$\eta$ degradation is likely due to discretization stiffness rather than the continuous-time formulation; this is presented as a hypothesis and supported by citing prior empirical observations, but it remains not fully tested within the paper via a targeted numerical study (e.g., step-size sweeps at fixed $\eta$).
- The nonlinear extension is conceptually appealing but still largely heuristic: it introduces an additional learned object $F_\theta^t$ and uses $\nabla F$ for guidance, but there is limited analysis connecting level-set learning error to sampling error.

Overall: the evidence convincingly supports the paper’s claims about the proposed geometric surrogate model; it is less convincing if interpreted as a theory of standard classifier/CFG guidance without additional bridging arguments or experiments.

**Requested Changes:**

- Tighten/clarify claims and scope.
    - Rephrase statements like “increasing (\eta) improves generation performance” to explicitly mean “improves manifold alignment under the geometric surrogate,” and discuss the role/interpretation of the constant $\tilde{C}$ in Theorem 11.
- Stronger bridge to real guidance (CFG / classifier guidance).
    - Add a section that more carefully maps the geometric proxy $-\eta P_i x$ to properties of $\nabla_x \log p_t(y\mid x)$ beyond the “approximately constant on manifold” heuristic (e.g., conditions under which the proxy is a controlled approximation).
- Empirical diagnostics of the “large-(\eta) degradation is numerical” hypothesis.
    - Provide an explicit experiment varying discretization step size at fixed large $\eta$, showing whether quality/diversity degradation is mitigated as hypothesized.

---

> ### Author Response · Authors · 2025-12-20
>
> We thank the reviewer for the thorough and thoughtful feedback, and we will revise the paper to clarify scope and strengthen the bridge to practical guidance where possible. Below we address the specific points.
>
> **Bridge to CFG / Classifier Guidance.** Our intuition that probabilistic guidance $\nabla_x \log p_t(y=1 \mid x)$ "almost parallel" to geometric guidance $P_1x$ was not rigorous. We sincerely thank the reviewer for highlighting this point, which prompted us to sharpen the argument. To formalize this intuition, we have obtained, under mild smoothness of the classifier, that
> $$
> \\|\nabla_x \log p_t(y=1 \mid x)+\eta_t P_1 x\\| \leq \beta_t, \quad \forall x \in M_1^t,
> $$
> for some scalar $\eta_t>0$. Moreover, if the classifier is confident on $M_1^t$,
> $$
> p_t(y=1 \mid x)>1-\varepsilon_t, \quad \forall x \in M_1^t,
> $$
> then $\beta_t=\mathcal{O}(\varepsilon_t)$. Intuitively, on $M_1^t$ the probabilistic guidance is almost parallel to the geometric guidance, with a residual controlled by classifier confidence.
>
> We will add this result with a rigorous and detailed analysis in the appendix.
>
> **Degradation of Performance as $\eta \rightarrow \infty$.** Our discussion below Theorem 11 about the hypothesis that performance degradation is due to discretization was not sufficiently clear or persuasive. We apologize for this and thank the reviewer for pointing it out. To clarify, we further analyzed the additional error induced by using Euler discretization to approximate the continuous dynamics. Under the linear manifold hypothesis and the existing results in Section 5, we show that the Euler discretization introduces an additional discretization error term in the $\mathcal{W}_1$, which is bounded by $\mathcal{O}(h\eta^2)$, where $h$ is the discretized step size. Therefore, as $\eta \rightarrow \infty$, this discretization error can become large and lead to performance degradation. This conclusion is consistent with the empirical observations, for example, the result reported by Wu et al. (2024, Figure 3), as discussed below Theorem 11. We will add this discretization analysis to the appendix in the revised version.
>
> For nonlinear settings such as CIFAR-10, we currently lack a corresponding theory, so we cannot provide a formal explanation. While our intuition is that large $\eta$ similarly amplifies discretization error, the degradation may also involve other practical factors (e.g., the training method for the learned nonlinear guidance). Extending discretization analysis to the nonlinear case is an important direction for future work.
>
> **More Discussions on $\tilde{C}$.** For the error floor $\tilde{C}$ in Theorem 11, we first clarify its decomposition and dependencies. As shown in the proof of Theorem 11, $\tilde{C} = \tilde{C}_1 + \tilde{C}_2$:
> - $\tilde{C}_1$ relates to first moments of latent density and is independent of $T,\delta,\sigma$ (see Eq. (36), Lemma D.3).
> - $\tilde{C}_2$ arises from score bounds (Eq. (38), Proposition D.5) and hence depends subtly on $T,\delta,\sigma$ (Appendix D.2).
>
> Regarding its role, at present we believe this error floor is inherent to the geometric guidance model: due to its analytical simplicity, geometric guidance cannot encode as much information as the probabilistic guidance term. For example, although we show
> $$
> \\|\nabla_{x} \log p_t(y=1 \mid x)+\eta_t P_1 x\\| \leq \beta_t,
> $$
> indicating that probabilistic guidance $\nabla_{x} \log p_t(y=1 \mid x)$ is “almost parallel” to geometric guidance $P_1 x$, the norm of the probabilistic guidance carries information that the geometric term cannot capture. This is a trade-off made for analytical tractability. We will add this discussion (and adjust any potentially overstated wording in the main text) in the revised version.

---

> > ### Author Response · Authors · 2025-12-20
> >
> > **More Discussion on Nonlinear Extension.** For the nonlinear extension, we acknowledge that Section 6 is an initial, constructive step: it introduces a principled way to build a nonlinear geometric guidance term, but we cannot claim that the linear-theory guarantees transfer directly.
> >
> > - Theoretically, Section 6 provides a fundamental construction of a nonlinear geometric guidance term adapted to the nonlinear setting. Establishing analogues of Theorems 10–11 will require different assumptions and is left for future work (as discussed in Limitations, Section 7).
> >
> > - Experimentally, Section 6.3 has two goals: (i) demonstrate that the nonlinear geometric guidance model can generate target conditional samples, which partially supports Theorem 13 and our construction intuition of geometric guidance term; and (ii) examine how performance changes with guidance scale in our model. The observed trends are consistent with the spirit of Theorem 11 in nonlinear regimes, but are not derived from it. We acknowledge the nonlinear experiments are foundational. Building robust nonlinear geometric guidance (e.g., beyond simple $\\|\nabla_x F_{y, \theta}^t(x)\\|$ control, which is time-consuming and unstable on larger datasets) is important future work.

---

### Author Response · Authors · 2025-12-20
**Updated Manuscript Uploaded: Key Revisions in Response to Reviewer Feedback**

In the revised version, we have made the following updates:

- **Appendix C.3:** We clarify our intuition that $\nabla_{x} \log p_t(y=1 \mid x)$ is "almost normal" to $M_1^t$ (thanks to Reviewer xZ9X and Reviewer bkAk). We also point this out in the main text at the end of p.7.

- **Appendix D.5:** We analyze the Euler discretization error to clarify our original hypothesis relating large $\eta$ to performance degradation (thanks to Reviewer vsZJ and Reviewer bkAk). We also rephrased the discussion at the end of Section 5.3 (p.13).

- **Assumption III clarification:** We make the sufficient conditions for Assumption III more explicit at the end of Appendix E.3 (p.45) and add a concrete truncated-Gaussian example in Remark 4 (p.10) (thanks to Reviewer xZ9X and Reviewer vsZJ).

- **Error floor $\tilde{C}$:** We add further discussion of the error floor $\tilde{C}$ in Remark D.1 (p.35) (thanks to all reviewers).

---

### Decision · Action_Editor_5At7 · 2026-01-26

**Recommendation:** Accept with minor revision

**Additional Comments:**

Before the final camera-ready upload, the authors should ensure that the following points from the discussion are integrated into the main text rather than just the appendix:
- Briefly explain the trade-off in Remark D.1 regarding why the Wasserstein bound does not vanish as $\eta \to \infty$.
- Ensure the truncated Gaussian example (Remark 4) is clearly visible to help readers grasp the regularity conditions.
- Ensure the distinction between the "proven" linear theory and the "principled construction" of the nonlinear model is clear to avoid overstating the scope of the theorems.

**Audience:**

Yes

**Audience Explanation:**

The guidance scale is one of the most widely used yet least understood hyperparameters in diffusion-based generative models. This work provides a clean, geometric view of guidance as a manifold-contracting force. This is of interest for researchers working on the theory of diffusion models, manifold hypothesis, and Wasserstein stability. Even though the nonlinear extension remains signficantly heuristic, the construction of level-set-based guidance is a novel contribution that could set the ground for future theoretical work.

**Claims And Evidence:**

Yes

**Claims Explanation:**

The paper provides a rigorous theoretical analysis of a geometric surrogate for diffusion guidance.  While the primary theorems are derived under a simplified linear manifold assumption and a modified guidance term, the authors have successfully bridged these theoretical results to practical settings in the revision. Specifically, they have provided: (i) A formal argument showing that the probabilistic guidance term is nearly parallel to the geometric normal vector under high classifier confidence; (ii) A new analysis explaining why extremely high guidance scales lead to performance degradation (a phenomenon previously observed empirically but now linked to Euler discretization error); (iii) A well-justified mollification technique to handle multi-modal data and ensure well-posed dynamics.

---

> ### Author Response · Authors · 2026-02-03
>
> Dear AE,
>
> Thank you for your suggestions. We have revised the paper based on your feedback and uploaded the camera-ready version. The main updates are:
>
> - Wasserstein bound: We add Remark 7 at the end of Theorem 11 to briefly discuss the inevitability of the error floor in the geometric guidance model, and we refer the reader to Remark D.1 for a more detailed discussion.
>
> - Assumption III: We clarified the truncated-Gaussian example in Remark 4 and pointed out that a detailed discussion of sufficient conditions ensuring Assumption III is provided in Appendix E.2.
>
> - Nonlinear extension:
>
> 	. In Abstract, we replace "... and support our theoretical findings." with "... and are consistent with our theories.".
>
> 	. In Section 1, in the introduction of the nonlinear extension, we tightened the scope of the statements by replacing "... empirically verify our theoretical findings ..." and "... validate our theoretical findings ..." with "... providing empirical evidence consistent with ..." and "... illustrate guidance-scale effects beyond the linear setting", respectively.
>
> 	. In Section 6.3, in the discussion of Performance v.s. Guidance scale, we replace "... this trend empirically supports Theorem 11 ..." with "... this trend is consistent with Theorem 11 ...".
>
> 	. At the end of Section 6.3, we add Remark 9 to emphasize that the observed trends are just consistent with the spirit of Theorem 11, and that establishing nonlinear analogues of Theorems 10–11 requires additional work.
>
> 	. In Section 7, we replace "... provided additional support for our theoretical findings." with "... provided additional evidence consistent with our theoretical findings.".
>
> We appreciate you and the reviewers for the detailed feedback and insightful comments, which have been invaluable in improving both the clarity and rigor of this work.
>
> Best regards,
>
> Authors